# Your Policy Regularizer is Secretly an Adversary

**Rob Brekelmans**                                                    *brekelma@usc.edu*
*University of Southern California*
*Information Sciences Institute*

**Tim Genewein**                                                    *timgen@deepmind.com*
**Jordi Grau-Moya**
**Grégoire Delétang**
**Markus Kunesch**
**Shane Legg**
**Pedro Ortega**                                                    *pedro.ortega@gmail.com*
*DeepMind*

**Reviewed on OpenReview:** *https: // openreview. net/ forum? id=XXXX*

## Abstract

Policy regularization methods such as maximum entropy regularization are widely used in reinforcement learning to improve the robustness of a learned policy. In this paper, we unify and extend recent work showing that this robustness arises from hedging against worst-case perturbations of the reward function, which are chosen from a limited set by an implicit adversary. Using convex duality, we characterize the robust set of adversarial reward perturbations under KL- and $\alpha$-divergence regularization, which includes Shannon and Tsallis entropy regularization as special cases. Importantly, generalization guarantees can be given within this robust set. We provide detailed discussion of the worst-case reward perturbations, and present intuitive empirical examples to illustrate this robustness and its relationship with generalization. Finally, we discuss how our analysis complements previous results on adversarial reward robustness and path consistency optimality conditions.

## 1 Introduction

Regularization plays a crucial role in various settings across reinforcement learning (RL), such as trust-region methods (Peters et al., 2010; Schulman et al., 2015; 2017; Bas-Serrano et al., 2021), offline learning (Levine et al., 2020; Nachum et al., 2019a;b; Nachum & Dai, 2020), multi-task learning (Teh et al., 2017; Igl et al., 2020), and soft $Q$-learning or actor-critic methods (Fox et al., 2016; Nachum et al., 2017; Haarnoja et al., 2017; 2018; Grau-Moya et al., 2018). Various justifications have been given for policy regularization, such as improved optimization (Ahmed et al., 2019), connections with probabilistic inference (Levine, 2018; Kappen et al., 2012; Rawlik et al., 2013; Wang et al., 2021), and robustness to perturbations in the environmental rewards or dynamics (Derman et al., 2021; Eysenbach & Levine, 2021; Husain et al., 2021).

In this work, we use convex duality to analyze the reward robustness which naturally arises from policy regularization in RL. In particular, we interpret regularized reward maximization as a two-player game between the agent and an imagined adversary that modifies the reward function. For a policy $\pi(a|s)$ regularized with a convex function $\Omega(\pi) = \mathbb{E}_\pi[\dot{\Omega}(\pi)]$ and regularization strength $1/\beta$, we investigate statements of the form

$$\max_{\pi(a|s)} (1-\gamma)\mathbb{E}_{\tau(\pi)}\left[\sum_{t=0}^{\infty} \gamma^t\left(r(a_t, s_t) - \frac{1}{\beta}\dot{\Omega}\big(\pi(a_t|s_t)\big)\right)\right] = \max_{\pi(a|s)} \min_{r'(a,s)\in\mathcal{R}_\pi} (1-\gamma)\mathbb{E}_{\tau(\pi)}\left[\sum_{t=0}^{\infty} \gamma^t r'(a_t, s_t)\right], \quad (1)$$

where $r'(a, s)$ indicates a modified reward function chosen from an appropriate robust set $\mathcal{R}_\pi$ (see Fig. 1-2). Eq. (1) suggests that an agent may translate uncertainty in its estimate of the reward function into

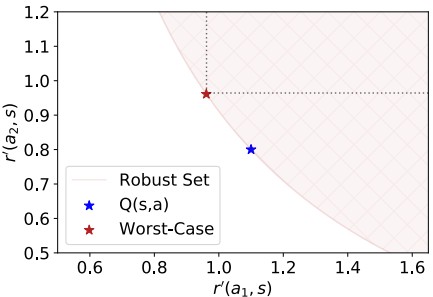

Figure 1: **Robust set** $\mathcal{R}_\pi$ (red region) of perturbed reward functions to which a stochastic policy generalizes, in the sense of Eq. (2). Red star indicates the worst-case perturbed reward $r'_{\pi_*} = r - \Delta r_{\pi_*}$ (Prop. 2) chosen by the adversary. The robust set also characterizes the set of reward perturbations $\Delta r(a, s)$ that are feasible for the adversary, which differs based on the choice of regularization function, regularization strength $\beta$, and reference distribution $\pi_0$ (see Sec. 4.1 and Fig. 2). We show the robust set for the optimal single-step policy with value estimates $Q(a, s) = r(a, s)$ and KL divergence regularization to a uniform $\pi_0$, with $\beta = 1$. Our robust set is larger and has a qualitatively different shape compared to the robust set of Derman et al. (2021) (dotted lines, see Sec. 5.2).

regularization of a learned policy, which is particularly relevant in applications such as inverse RL (Ng et al., 2000; Arora & Doshi, 2021) or learning from human preferences (Christiano et al., 2017).

This reward robustness further implies that regularized policies achieve a form of 'zero-shot' generalization to new environments where the reward is adversarially chosen. In particular, for any given $\pi(a|s)$ and a modified reward $r' \in \mathcal{R}_\pi$ within the corresponding robust set, we obtain the following performance guarantee

$$\mathbb{E}_{\tau(\pi)}\left[\sum_{t=0}^{\infty} \gamma^t r'(a_t, s_t)\right] \geq \mathbb{E}_{\tau(\pi)}\left[\sum_{t=0}^{\infty} \gamma^t \left(r(a_t, s_t) - \frac{1}{\beta}\dot{\Omega}(\pi_t)\right)\right]. \tag{2}$$

Eq. (2) states that the expected modified reward under $\pi(a|s)$, with $r' \in \mathcal{R}_\pi$ as in Fig. 1, will be greater than the value of the regularized objective with the original, unmodified reward. It is in this particular sense that we make claims about robustness and zero-shot generalization throughout the paper.

Our analysis unifies recent work exploring similar interpretations (Ortega & Lee, 2014; Husain et al., 2021; Eysenbach & Levine, 2021; Derman et al., 2021) as summarized in Sec. 5 and Table 1. Our contributions include

- A thorough analysis of the robustness associated with KL and $\alpha$-divergence policy regularization, which includes popular Shannon entropy regularization as a special case. Our derivations for the $\alpha$-divergence generalize the Tsallis entropy RL framework of Lee et al. (2019).

- We derive the worst-case reward perturbations $\Delta r_\pi = r - r'_\pi$ corresponding to any stochastic policy $\pi$ and a fixed regularization scheme (Prop. 2).

- For the optimal regularized policy in a given environment, we show that the corresponding worst-case reward perturbations match the advantage function for *any* $\alpha$-divergence. We relate this finding to the path consistency optimality condition, which has been used to construct learning objectives in (Nachum et al., 2017; Chow et al., 2018), and a game-theoretic indifference condition, which occurs at a Nash equilibrium between the agent and adversary (Ortega & Lee, 2014).

- We visualize the set $\mathcal{R}_\pi$ of adversarially perturbed rewards against which a regularized policy is robust in Fig. 1-2, with details in Prop. 1. Our use of divergence instead of entropy regularization to analyze the robust set clarifies several unexpected conclusions from previous work. In particular, similar plots in Eysenbach & Levine (2021) suggest that MaxEnt RL is not robust to the reward function of the training environment, and that increased regularization strength may hurt robustness. Our analysis in Sec. 5.1 and App. F.4 establishes the expected, opposite results.

- We perform experiments for a sequential grid-world task in Sec. 4 where, in contrast to previous work, we explicitly visualize the reward robustness and adversarial strategies resulting from our theory. We use the path consistency or indifference conditions to certify optimality of the policy.

| | Ortega & Lee (2014) | Eysenbach & Levine (2021) | Husain et al. (2021) | Derman et al. (2021) | Ours |
|---|---|---|---|---|---|
| Multi-Step Analysis | ✗ | ✓ | ✓ | ✓ | ✓ |
| Worst-Case $\Delta r(a,s)$ | policy form | policy form | value form | policy (via dual LP Eq. (11)) | policy & value forms |
| Robust Set | ✗ | ✓ (see our App. F.4) | ✗ | ✓(flexible specification) | ✓ |
| Divergence Used | KL ($\alpha = 1$) | Shannon entropy (Sec. 5.1) | any convex $\Omega$ | derived from robust set | any convex $\Omega$, $\alpha$-Div examples |
| $\mu(a,s)$ or $\pi(a\vert s)$ Reg.? | $\pi(a\vert s)$ | $\pi(a\vert s)$ | Both | $\pi(a\vert s)$ | Both |
| Indifference | ✓ | ✗ | ✗ | ✗ | ✓ |
| Path Consistency | ✗ | ✗ | ✗ | ✗ | ✓ |

Table 1: Comparison to related work.

## 2 Preliminaries

In this section, we review linear programming (LP) formulations of discounted Markov Decision Processes (MDP) and extensions to convex policy regularization.

**Notation** For a finite set $\mathcal{X}$, let $\mathbb{R}^{\mathcal{X}}$ denote the space of real-valued functions over $\mathcal{X}$, with $\mathbb{R}_+^{\mathcal{X}}$ indicating restriction to non-negative functions. We let $\Delta^{\vert\mathcal{X}\vert}$ denote the probability simplex with dimension equal to the cardinality of $\mathcal{X}$. For $\mu, q \in \mathbb{R}^{\mathcal{X}}$, $\langle \mu, q \rangle = \sum_{x \in \mathcal{X}} \mu(x) q(x)$ indicates the inner product in Euclidean space.

### 2.1 Convex Conjugate Function

We begin by reviewing the convex conjugate function, also known as the Legendre-Fenchel transform, which will play a crucial role throughout our paper. For a convex function $\Omega(\mu)$ which, in our context, has domain $\mu \in \mathbb{R}_+^{\mathcal{X}}$, the conjugate function $\Omega^*$ is defined via the optimization

$$\Omega^*(\Delta r) = \sup_{\mu \in \mathbb{R}_+^{\mathcal{X}}} \langle \mu, \Delta r \rangle - \Omega(\mu), \tag{3}$$

where $\Delta r \in \mathbb{R}^{\mathcal{X}}$. The conjugate operation is an involution for proper, lower semi-continuous, convex $\Omega$ (Boyd & Vandenberghe, 2004), so that $(\Omega^*)^* = \Omega$ and $\Omega^*$ is also convex. We can thus represent $\Omega(\mu)$ via a conjugate optimization

$$\Omega(\mu) = \sup_{\Delta r \in \mathbb{R}^{\mathcal{X}}} \langle \mu, \Delta r \rangle - \Omega^*(\Delta r). \tag{4}$$

Differentiating with respect to the optimization variable in Eq. (3) or (4) suggests the optimality conditions

$$\mu_{\Delta r} = \nabla \Omega^*(\Delta r) \qquad \Delta r_\mu = \nabla \Omega(\mu). \tag{5}$$

Note that the above conditions also imply relationships of the form $\mu_{\Delta r} = (\nabla \Omega)^{-1}(\Delta r)$. This dual correspondence between values of $\mu$ and $\Delta r$ will form the basis of our adversarial interpretation in Sec. 3.

### 2.2 Divergence Functions

We are interested in the conjugate duality associated with policy regularization, which is often expressed using a statistical divergence $\Omega(\mu)$ over a joint density $\mu(a,s) = \mu(s)\pi(a\vert s)$ (see Sec. 2.3). In particular, we consider the family of $\alpha$-divergences (Amari, 2016; Cichocki & Amari, 2010), which includes both the forward and reverse KL divergences as special cases. In the following, we consider extended divergences that accept unnormalized density functions as input (Zhu & Rohwer, 1995) so that we may analyze function space dualities and evaluate Lagrangian relaxations without projection onto the probability simplex.

**KL Divergence** The 'forward' KL divergence to a reference policy $\pi_0(a\vert s)$ is commonly used for policy regularization in RL. Extending the input domain to unnormalized measures, we write the divergence as

$$\Omega_{\pi_0}(\mu) = \mathbb{E}_{\mu(s)}\Big[ D_{KL}[\pi : \pi_0] \Big] = \sum_{s \in \mathcal{S}} \mu(s) \sum_{a \in \mathcal{A}} \left( \pi(a\vert s) \log \frac{\pi(a\vert s)}{\pi_0(a\vert s)} - \pi(a\vert s) + \pi_0(a\vert s) \right). \tag{6}$$

Using a uniform reference $\pi_0(a\vert s) = 1 \; \forall \, (a,s)$, we recover the Shannon entropy up to an additive constant.

**$\alpha$-Divergence** The $\alpha$-divergence $\mathbb{E}_{\mu(s)}\big[ D_\alpha[\pi_0 : \pi] \big]$ over possibly unnormalized measures is defined as

$$\Omega_{\pi_0}^{(\alpha)}(\mu) = \frac{1}{\alpha(1-\alpha)} \sum_{s \in \mathcal{S}} \mu(s) \left( (1-\alpha) \sum_{a \in \mathcal{A}} \pi_0(a\vert s) + \alpha \sum_{a \in \mathcal{A}} \pi(a\vert s) - \sum_{a \in \mathcal{A}} \pi_0(a\vert s)^{1-\alpha} \pi(a\vert s)^\alpha \right) \tag{7}$$

| Divergence | Conjugate | Conjugate Expression | Optimizing Argument ($\pi_{\Delta_r}$ or $\mu_{\Delta_r}$) |
|---|---|---|---|
| $\frac{1}{\beta}D_{\mathrm{KL}}[\pi : \pi_0]$ | $\frac{1}{\beta}\Omega^*_{\pi_0,\beta}(\Delta r)$ | $\frac{1}{\beta}\sum\limits_{a}\pi_0(a\|s)\exp\left\{\beta\cdot\Delta r(a,s)\right\} - \frac{1}{\beta}$ | $\pi_0(a\|s)\exp\left\{\beta\cdot\Delta r(a,s)\right\}$ |
| $\frac{1}{\beta}D_{\mathrm{KL}}[\mu : \mu_0]$ | $\frac{1}{\beta}\Omega^*_{\mu_0,\beta}(\Delta r)$ | $\frac{1}{\beta}\sum\limits_{a,s}\mu_0(a,s)\exp\left\{\beta\cdot\Delta r(a,s)\right\} - \frac{1}{\beta}$ | $\mu_0(a,s)\exp\left\{\beta\cdot\Delta r(a,s)\right\}$ |
| $\frac{1}{\beta}D_\alpha[\pi_0 : \pi]$ | $\frac{1}{\beta}\Omega^{*(\alpha)}_{\pi_0,\beta}(\Delta r)$ | $\frac{1}{\beta}\frac{1}{\alpha}\sum\limits_{a}\pi_0(a\|s)\exp_\alpha\left\{\beta\cdot\left(\Delta r(a,s)-\psi_{\Delta_r}(s;\beta)\right)\right\}^\alpha - \frac{1}{\beta}\frac{1}{\alpha} + \psi_{\Delta_r}(s;\beta)$ | $\pi_0(a\|s)\exp_\alpha\left\{\beta\cdot\left(\Delta r(a,s)-\psi_{\Delta_r}(s;\beta)\right)\right\}$ |
| $\frac{1}{\beta}D_\alpha[\mu_0 : \mu]$ | $\frac{1}{\beta}\Omega^{*(\alpha)}_{\mu_0,\beta}(\Delta r)$ | $\frac{1}{\beta}\frac{1}{\alpha}\sum\limits_{a,s}\mu_0(a,s)\exp_\alpha\left\{\beta\cdot\Delta r(a,s)\right\}^\alpha - \frac{1}{\beta}\frac{1}{\alpha}$ | $\mu_0(a,s)\exp_\alpha\left\{\beta\cdot\Delta r(a,s)\right\}$ |

Table 2: Conjugate Function expressions for KL and $\alpha$-divergence regularization of either the policy $\pi(a|s)$ or occupancy $\mu(a,s)$. See App. B.1-B.4 for derivations. The final column shows the optimizing argument in the definition of the conjugate function $\frac{1}{\beta}\Omega^*(\Delta r)$, for example $\mu_{\Delta_r} := \arg\max_\mu\langle\mu,\Delta r\rangle - \frac{1}{\beta}\Omega_{\mu_0}(\mu)$. Note that each conjugate expression for $\pi(a|s)$ regularization also contains an outer expectation over $\mu(s)$.

Taking the limiting behavior, we recover the 'forward' KL divergence $D_{KL}[\pi : \pi_0]$ as $\alpha \to 1$ or the 'reverse' KL divergence $D_{KL}[\pi_0 : \pi]$ as $\alpha \to 0$.

To provide intuition for the $\alpha$-divergence, we define the deformed $\alpha$-logarithm as in Lee et al. (2019), which matches Tsallis's $q$-logarithm (Tsallis, 2009) for $\alpha = 2 - q$. Its inverse is the $\alpha$-exponential, with

$$\log_\alpha(u) = \frac{1}{\alpha - 1}\left(u^{\alpha-1} - 1\right), \qquad \exp_\alpha(u) = \left[1 + (\alpha - 1)u\right]^{\frac{1}{\alpha-1}}_+. \tag{8}$$

where $[\cdot]_+ = \max(\cdot, 0)$ ensures fractional powers can be taken and suggests that $\exp_\alpha(u) = 0$ for $u \leq 1/(1-\alpha)$. Using the $\alpha$-logarithm, we can rewrite the $\alpha$-divergence similarly to the KL divergence in Eq. (6)

$$\Omega^{(\alpha)}_{\pi_0}(\mu) = \frac{1}{\alpha}\sum_{s\in\mathcal{S}}\mu(s)\left(\sum_{a\in\mathcal{A}}\pi(a|s)\log_\alpha\frac{\pi(a|s)}{\pi_0(a|s)} - \pi(a|s) + \pi_0(a|s)\right).$$

For a uniform reference $\pi_0$, the $\alpha$-divergence differs from the Tsallis entropy by only the $1/\alpha$ factor and an additive constant (see App. F.1).

## 2.3 Unregularized MDPs

A discounted MDP is a tuple $\{\mathcal{S}, \mathcal{A}, P, \nu_0, r, \gamma\}$ consisting of a state space $\mathcal{S}$, action space $\mathcal{A}$, transition dynamics $P(s'|s,a)$ for $s, s' \in \mathcal{S}$, $a \in \mathcal{A}$, initial state distribution $\nu_0(s) \in \Delta^{|\mathcal{S}|}$ in the probability simplex, and reward function $r(a,s) : \mathcal{S} \times \mathcal{A} \mapsto \mathbb{R}$. We also use a discount factor $\gamma \in (0,1)$ (Puterman (1994) Sec 6).

We consider an agent that seeks to maximize the expected discounted reward by acting according to a decision policy $\pi(a|s) \in \Delta^{|\mathcal{A}|}$ for each $s \in \mathcal{S}$. The expected reward is calculated over trajectories $\tau \sim \pi(\tau) := \nu_0(s_0)\prod\pi(a_t|s_t)P(s_{t+1}|s_t,a_t)$, which begin from an initial $s_0 \sim \nu_0(s)$ and evolve according to the policy $\pi(a|s)$ and MDP dynamics $P(s'|s,a)$

$$\mathcal{RL}(r) := \max_{\pi(a|s)}(1-\gamma)\,\mathbb{E}_{\tau\sim\pi(\tau)}\left[\sum_{t=0}^\infty \gamma^t\,r(s_t,a_t)\right]. \tag{9}$$

We assume that the policy is stationary and Markovian, and thus independent of both the timestep and trajectory history.

**Linear Programming Formulation** We will focus on a linear programming (LP) form for the objective in Eq. (9), which is common in the literature on convex duality. With optimization over the discounted state-action occupancy measure, $\mu(a,s) := (1-\gamma)\mathbb{E}_{\tau\sim\pi(\tau)}\left[\sum_{t=0}^\infty \gamma^t\,\mathbb{I}(a_t = a, s_t = s)\right]$, we rewrite the objective as

$$\mathcal{RL}(r) := \max_\mu\langle\mu, r\rangle \qquad \text{subject to} \quad \mu(a,s) \geq 0 \qquad \forall(a,s)\in\mathcal{A}\times\mathcal{S}, \tag{10}$$

$$\sum_a\mu(a,s) = (1-\gamma)\nu_0(s) + \gamma\sum_{a',s'}P(s|a',s')\mu(a',s') \qquad \forall s\in\mathcal{S}.$$

We refer to the constraints in the second line of Eq. (10) as the *Bellman flow constraints*, which force $\mu(a,s)$ to respect the MDP dynamics. We denote the set of feasible $\mu$ as $\mathcal{M} \subset \mathbb{R}^{\mathcal{A}\times\mathcal{S}}_+$. For normalized $\nu_0(s)$ and $P(s|a',s')$, we show in App. A.2 that $\mu(a,s) \in \mathcal{M}$ implies $\mu(a,s)$ is normalized.

It can be shown that any feasible $\mu(a, s) \in \mathcal{M}$ induces a stationary $\pi(a|s) = \mu(a, s)/\mu(s)$, where $\mu(s) := \sum_{a'} \mu(a', s)$ and $\pi(a|s) \in \Delta^{|\mathcal{A}|}$ is normalized by definition. Conversely, any stationary policy $\pi(a|s)$ induces a unique state-action visitation distribution $\mu(a, s)$ (Syed et al. (2008), Feinberg & Shwartz (2012) Sec. 6.3). Along with the definition of $\mu(a, s)$ above, this result demonstrates the equivalence of the optimizations in Eq. (9) and Eq. (10). We will proceed with the LP notation from Eq. (10) and assume $\mu(s)$ is induced by $\pi(a|s)$ whenever the two appear together in an expression.

Importantly, the flow constraints in Eq. (10) lead to a dual optimization which reflects the familiar Bellman equations (Bellman, 1957). To see this, we introduce Lagrange multipliers $V \in \mathbb{R}^{\mathcal{S}}$ for each flow constraint and $\lambda(a, s) \in \mathbb{R}_+^{\mathcal{A} \times \mathcal{S}}$ for the nonnegativity constraints. Summing over $s \in \mathcal{S}$, and eliminating $\mu(a, s)$ by setting $d/d\mu(a, s) = 0$ yields the *dual* LP

$$\mathcal{RL}^*(r) := \min_{V, \lambda} (1 - \gamma)\langle \nu_0, V \rangle \quad \text{subject to} \quad V(s) = r(a, s) + \gamma \mathbb{E}_{a,s}^{s'}[V(s')] + \lambda(a, s) \quad \forall (a, s) \in \mathcal{A} \times \mathcal{S}, \quad (11)$$

where we have used $\mathbb{E}_{a,s}^{s'}[V(s')]$ as shorthand for $\mathbb{E}_{P(s'|a,s)}[V(s')]$ and reindexed the transition tuple from $(s', a', s)$ to $(s, a, s')$ compared to Eq. (10). Note that the constraint applies for all $(a, s) \in \mathcal{A} \times \mathcal{S}$ and that $\lambda(a, s) \geq 0$. By complementary slackness, we know that $\lambda(a, s) = 0$ for $(a, s)$ such that $\mu(a, s) > 0$.

## 2.4 Regularized MDPs

We now consider regularizing the objective in Eq. (10) using a convex penalty function $\Omega(\mu)$ with coefficient $1/\beta$. We primarily focus on regularization using a conditional divergence $\Omega_{\pi_0}(\mu) := \mathbb{E}_{\mu(s)\pi(a|s)}[\dot{\Omega}(\pi)]$ between the policy and a normalized reference distribution $\pi_0(a|s)$, as in Sec. 2.2 and (Ortega & Braun, 2013; Fox et al., 2016; Haarnoja et al., 2017; 2018). We also use the notation $\Omega_{\mu_0}(\mu) = \mathbb{E}_{\mu(a,s)}[\dot{\Omega}(\mu)]$ to indicate regularization of the full state-action occupancy measure to a normalized reference $\mu_0(a, s)$, which appears, for example, in Relative Entropy Policy Search (REPS) (Peters et al., 2010; Belousov & Peters, 2019). The regularized objective $\mathcal{RL}_{\Omega,\beta}(r)$ is then defined as

$$\mathcal{RL}_{\Omega,\beta}(r) := \max_{\mu \in \mathcal{M}} \langle \mu, r \rangle - \frac{1}{\beta}\Omega_{\pi_0}(\mu) \tag{12}$$

where $\Omega_{\pi_0}(\mu)$ contains an expectation under $\mu(a, s)$ as in Eq. (6)-(7). We can also derive a dual version of the regularized LP, by first writing the Lagrangian relaxation of Eq. (12)

$$\max_\mu \min_{V, \lambda} (1 - \gamma)\langle \nu_0, V \rangle + \langle \mu, r + \gamma \mathbb{E}_{a,s}^{s'}[V] - V + \lambda \rangle - \frac{1}{\beta}\Omega_{\pi_0}(\mu). \tag{13}$$

Swapping the order of optimization under strong duality, we can recognize the maximization over $\mu(a, s)$ as a conjugate function $\frac{1}{\beta}\Omega_{\pi_0,\beta}^*$, as in Eq. (3), leading to a regularized dual optimization

$$\mathcal{RL}_{\Omega,\beta}^*(r) = \min_{V, \lambda} (1 - \gamma)\langle \nu_0, V \rangle + \frac{1}{\beta}\Omega_{\pi_0,\beta}^*\left(r + \gamma \mathbb{E}_{a,s}^{s'}[V] - V + \lambda\right) \tag{14}$$

which involves optimization over dual variables $V(s)$ only and is unconstrained, in contrast to Eq. (11). Dual objectives of this form appear in (Nachum & Dai, 2020; Belousov & Peters, 2019; Bas-Serrano et al., 2021; Neu et al., 2017). We emphasize the need to include the Lagrange multiplier $\lambda(a, s)$, with $\lambda(a, s) > 0$ when the optimal policy has $\pi_*(a|s) = 0$, since an important motivation for $\alpha$-divergence regularization is to encourage sparsity in the policy (see Eq. (8), Lee et al. (2018; 2019); Chow et al. (2018)).

**Soft Value Aggregation** In iterative algorithms such as (regularized) modified policy iteration (Puterman & Shin, 1978; Scherrer et al., 2015), it is useful to consider the *regularized Bellman optimality operator* (Geist et al., 2019). For given estimates of the state-action value $Q(a, s) := r(a, s) + \gamma \mathbb{E}_{a,s}^{s'}[V(s')]$, the operator $\mathcal{T}_{\Omega_{\pi_0,\beta}}^*$ updates $V(s)$ as

$$V(s) \leftarrow \frac{1}{\beta}\Omega_{\pi_0,\beta}^*(Q) = \max_{\pi \in \Delta^{|\mathcal{A}|}} \langle \pi, Q \rangle - \frac{1}{\beta}\Omega_{\pi_0}(\pi). \tag{15}$$

Note that this conjugate optimization is performed in each state $s \in \mathcal{S}$ and explicitly constrains each $\pi(a|s)$ to be normalized. Although we proceed with the notation of Eq. (12) and Eq. (14), our later developments are compatible with the 'soft-value aggregation' perspective above. See App. C for detailed discussion.

## 3 Adversarial Interpretation

In this section, we interpret regularization as implicitly providing robustness to adversarial perturbations of the reward function. To derive our adversarial interpretation, recall from Eq. (4) that conjugate duality yields an alternative representation of the regularizer

$$\frac{1}{\beta}\Omega_{\pi_0}^{(\alpha)}(\mu) = \max_{\Delta r \in \mathbb{R}^{\mathcal{A}\times\mathcal{S}}} \langle \mu, \Delta r \rangle - \frac{1}{\beta}\Omega_{\pi_0,\beta}^{*(\alpha)}(\Delta r). \tag{16}$$

Using this conjugate optimization to expand the regularization term in the primal objective of Eq. (12),

$$\mathcal{RL}_{\Omega,\beta}(r) = \max_{\mu \in \mathcal{M}} \min_{\Delta r \in \mathbb{R}^{\mathcal{A}\times\mathcal{S}}} \langle \mu, r - \Delta r \rangle + \frac{1}{\beta}\Omega_{\pi_0,\beta}^{*(\alpha)}(\Delta r). \tag{17}$$

We interpret Eq. (17) as a two-player minimax game between an agent and an implicit adversary, where the agent chooses an occupancy measure $\mu(a,s) \in \mathcal{M}$ or its corresponding policy $\pi(a|s)$, and the adversary chooses reward perturbations $\Delta r(a,s)$ subject to the convex conjugate $\frac{1}{\beta}\Omega_{\pi_0,\beta}^{*(\alpha)}(\Delta r)$ as a penalty function (Ortega & Lee, 2014).

To understand the limitations this penalty imposes on the adversary, we transform the optimization over $\Delta r$ in Eq. (17) to a constrained optimization in Sec. 3.1. This allows us to characterize the feasible set of reward perturbations available to the adversary or, equivalently, the set of modified rewards $r'(a,s) \in \mathcal{R}_\pi$ to which a particular stochastic policy is robust. In Sec. 3.2 and 3.4, we interpret the worst-case adversarial perturbations corresponding to an arbitrary stochastic policy and the optimal policy, respectively.

### 3.1 Robust Set of Modified Rewards

In order to link our adversarial interpretation to robustness and zero-shot generalization as in Eq. (1)-(2), we characterize the feasible set of reward perturbations in the following proposition. We state our proposition for policy regularization, and discuss differences for $\mu(a,s)$ regularization in App. D.2.

**Proposition 1.** *Assume a normalized policy $\pi(a|s)$ for the agent is given, with $\sum_a \pi(a|s) = 1 \forall s \in \mathcal{S}$. Under $\alpha$-divergence policy regularization to a normalized reference $\pi_0(a|s)$, the optimization over $\Delta r(a,s)$ in Eq. (17) can be written in the following constrained form*

$$\min_{\Delta r \in \mathcal{R}_\pi^\Delta} \langle \mu, r - \Delta r \rangle \qquad where \quad \mathcal{R}_\pi^\Delta := \left\{ \Delta r \in \mathbb{R}^{\mathcal{A}\times\mathcal{S}} \,\middle|\, \Omega_{\pi_0,\beta}^{*(\alpha)}(\Delta r) \leq 0 \right\}, \tag{18}$$

*We refer to $\mathcal{R}_\pi^\Delta \subset \mathbb{R}^{\mathcal{A}\times\mathcal{S}}$ as the feasible set of reward perturbations available to the adversary. This translates to a robust set $\mathcal{R}_\pi$ of modified rewards $r'(a,s) = r(a,s) - \Delta r(a,s)$ for the given policy. These sets depend on the $\alpha$-divergence and regularization strength $\beta$ via the conjugate function.*

*For KL divergence regularization, the constraint is*

$$\sum_{a \in \mathcal{A}} \pi_0(a|s) \exp\left\{ \beta \cdot \Delta r(a,s) \right\} \leq 1. \tag{19}$$

See App. D.1 for proof, and Table 2 for the convex conjugate function $\frac{1}{\beta}\Omega_{\pi_0,\beta}^{*(\alpha)}(\Delta r)$ associated with various regularization schemes. The proof proceeds by evaluating the conjugate function at the minimizing argument $\Delta r_\pi$ in Eq. (17) (see Sec. 3.2), with $\Omega_{\pi_0,\beta}^{*(\alpha)}(\Delta r_\pi) = 0 \,\forall \alpha$ for normalized $\pi(a|s)$ and $\pi_0(a|s)$. The constraint then follows from the fact that $\Omega_{\pi_0,\beta}^{*(\alpha)}(\Delta r_\pi)$ is convex and increasing in $\Delta r$ (Husain et al., 2021). We visualize the robust set for a two-dimensional action space in Fig. 2, with additional discussion in Sec. 4.1.

As in Eq. (2), we can provide 'zero-shot' performance guarantees using this set of modified rewards. For any perturbed reward in the robust set $r' \in \mathcal{R}_\pi$, we have $\langle \mu, r' \rangle \geq \langle \mu, r \rangle - \frac{1}{\beta}\Omega_{\pi_0}^{(\alpha)}(\mu)$, so that the policy achieves an expected modified reward which is at least as large as the regularized objective. However, notice that this form of robustness is sensitive to the exact value of the regularized objective function. Although entropy regularization and divergence regularization with a uniform reference induce the same optimal $\mu(a,s)$, we highlight crucial differences in their reward robustness interpretations in Sec. 5.1.

## 3.2 Worst-Case Perturbations: Policy Form

From the feasible set in Prop. 1, how should the adversary select its reward perturbations? In the following proposition, we use the optimality conditions in Eq. (5) to solve for the *worst-case* reward perturbations $\Delta r_\pi(a, s)$ which minimize Eq. (17) for an fixed but arbitrary stochastic policy $\pi(a|s)$.

**Proposition 2.** *For a given policy $\pi(a|s)$ or state-action occupancy $\mu(a, s)$, the worst-case adversarial reward perturbations $\Delta r_\pi$ or $\Delta r_\mu$ associated with a convex function $\Omega(\mu)$ and regularization strength $1/\beta$ are*

$$\Delta r_\pi = \nabla_\mu \frac{1}{\beta} \Omega(\mu) \,. \tag{20}$$

See App. A.1 for proof. We now provide example closed form expressions for the worst-case reward perturbations under common regularization schemes. We emphasize that the same stochastic policy $\pi(a|s)$ or joint occupancy measure $\mu(a, s)$ can be associated with different adversarial perturbations depending on the choice of $\alpha$-divergence and strength $\beta$.[1]

**KL Divergence** For KL divergence policy regularization, the worst-case reward perturbations are

$$\Delta r_\pi(a, s) = \frac{1}{\beta} \log \frac{\pi(a|s)}{\pi_0(a|s)} \,, \tag{21}$$

which corresponds to the pointwise regularization $\Delta r_\pi(a, s) = \dot{\Omega}_{\pi_0}(\pi(a|s))$ for each state-action pair, with $\Omega_{\pi_0}(\mu) = \mathbb{E}_{\mu(a,s)}[\dot{\Omega}_{\pi_0}(\pi(a|s))]$. See App. B.1. We show an analogous result in App. B.2 for state-action occupancy regularization $D_{\mathrm{KL}}[\mu : \mu_0]$, where $\Delta r_\mu(a, s) = \frac{1}{\beta} \log \frac{\mu(a,s)}{\mu_0(a,s)} = \dot{\Omega}_{\mu_0}(\mu(a, s))$.

**$\alpha$-Divergence** For KL divergence regularization, the worst-case reward perturbations had a similar expression for conditional and joint regularization. However, we observe notable differences for the $\alpha$-divergence in general. For policy regularization to a reference $\pi_0$,

$$\Delta r_\pi(a, s) = \frac{1}{\beta} \log_\alpha \frac{\pi(a|s)}{\pi_0(a|s)} + \psi_{\Delta r}(s; \beta), \tag{22}$$

where we define $\psi_{\Delta r}(s; \beta)$ as

$$\psi_{\Delta r}(s; \beta) := \frac{1}{\beta} \frac{1}{\alpha} \left( \sum_{a \in \mathcal{A}} \pi_0(a|s) - \sum_{a \in \mathcal{A}} \pi_0(a|s)^{1-\alpha} \pi(a|s)^\alpha \right). \tag{23}$$

As we discuss in App. B.3, $\psi_{\Delta r}(s; \beta)$ plays the role of a normalization constant for the optimizing argument $\pi_{\Delta r}(a|s)$ in the definition of $\frac{1}{\beta} \Omega^{*(\alpha)}_{\pi_0, \beta}(\Delta r)$ (see Eq. (3), Table 2). This term arises from differentiating $\Omega^{(\alpha)}_{\pi_0}(\mu)$ with respect to $\mu(a, s)$ instead of from an explicit constraint. Assuming the given $\pi(a|s)$ and reference $\pi_0(a|s)$ are normalized, note that $\psi_{\Delta r}(s; \beta) = \frac{1}{\beta}(1-\alpha)D_\alpha[\pi_0 : \pi]$. With normalization, we also observe that $\psi_{\Delta r}(s; \beta) = 0$ for KL divergence regularization $(\alpha = 1)$, which confirms Eq. (21) is a special case of Eq. (22).

For any given state-action occupancy measure $\mu(a, s)$ and joint $\alpha$-divergence regularization to a reference $\mu_0(a|s)$, the worst-case perturbations become

$$\Delta r_\mu(a, s) = \frac{1}{\beta} \log_\alpha \frac{\mu(a, s)}{\mu_0(a, s)}, \tag{24}$$

with detailed derivations in App. B.4. In contrast to Eq. (22), this expression lacks an explicit normalization constant, as this constraint is enforced by the Lagrange multipliers $V(s)$ and $\mu(a, s) \in \mathcal{M}$ (App. A.2).

---

[1]One exception is that a policy with $a'$ s.t. $\pi(a'|s) = 0$ can only be represented using KL regularization if $\pi_0(a'|s) = 0$.

### 3.3 Worst-Case Perturbations: Value Form

In the previous section, we analyzed the implicit adversary corresponding to *any* stochastic policy $\pi(a|s)$ for a given $\Omega, \pi_0$, and $\beta$. We now take a dual perspective, where the adversary is given access to a set of dual variables $V(s)$ across states $s \in \mathcal{S}$ and selects reward perturbations $\Delta r_V(a, s)$. We will eventually show in Sec. 3.4 that these perturbations match the policy-form perturbations at optimality.

Our starting point is Theorem 3 of Husain et al. (2021), which arises from taking the convex conjugate $(-\mathcal{RL}_{\Omega,\beta}(r))^*$ of the *entire* regularized objective $\mathcal{RL}_{\Omega,\beta}(r)$, which is concave in $\mu(a, s)$. See App. E.1.

**Theorem 1** (Husain et al. (2021)). *The optimal value of the regularized objective $\mathcal{RL}_{\Omega,\beta}(r)$ in Eq. (12), or its dual $\mathcal{RL}^*_{\Omega,\beta}(r)$ in Eq. (14), is equal to*

$$\inf_{V,\lambda} \inf_{\Delta r_V} \ (1-\gamma)\langle \nu_0, V \rangle + \frac{1}{\beta}\Omega^{*(\alpha)}_{\pi_0,\beta}\left(\Delta r_V\right) \tag{25}$$
$$\text{subject to} \quad V(s) = r(a,s) + \gamma \mathbb{E}^{s'}_{a,s}\big[V(s')\big] - \Delta r_V(a,s) + \lambda(a,s) \quad \forall (a,s) \in \mathcal{A} \times \mathcal{S}.$$

Rearranging the equality constraint to solve for $\Delta r_V(a, s)$ and substituting into the objective, this optimization recovers the regularized dual problem in Eq. (14). We can also compare Eq. (25) to the *unregularized* dual problem in Eq. (11), which does not include an adversarial cost and whose constraint $V(s) = r(a, s) + \gamma \mathbb{E}^{s'}_{a,s}\big[V(s')\big] + \lambda(a, s)$ implies an *unmodified* reward, or $\Delta r_V(a, s) = 0$. Similarly to Sec. 3.2, the adversary incorporates the effect of policy regularization via the reward perturbations $\Delta r_V(a, s)$.

### 3.4 Policy Form = Value Form at Optimality

In the following proposition, we provide a link between the policy and value forms of the adversarial reward perturbations, showing that $\Delta r_{\pi_*}(a, s) = \Delta r_{V_*}(a, s)$ for the optimal policy $\pi_*(a|s)$ and value $V_*(s)$. As in Eysenbach & Levine (2021), the uniqueness of the optimal policy implies that its robustness may be associated with an environmental reward $r(a, s)$ for a given regularized MDP.

**Proposition 3.** *For the optimal policy $\pi_*(a|s)$ and value function $V_*(s)$ corresponding to $\alpha$-divergence policy regularization with strength $\beta$, the policy and value forms of the worst-case adversarial reward perturbations match, $\Delta r_{\pi_*} = \Delta r_{V_*}$, and are related to the advantage function via*

$$\Delta r_{\pi_*}(a, s) = Q_*(a, s) - V_*(s) + \lambda_*(a, s), \tag{26}$$

*where we define $Q_*(a, s) \coloneqq r(a, s) + \gamma \mathbb{E}^{s'}_{a,s}\big[V_*(s')\big]$ and recall $\lambda_*(a, s)\pi_*(a|s) = 0$ by complementary slackness. Note that $V_*(s)$ depends on the regularization scheme via the conjugate function $\frac{1}{\beta}\Omega^{*(\alpha)}_{\pi_0,\beta}(\Delta r_V)$ in Eq. (25).*

*Proof.* See App. A.3. We consider the optimal policy in an MDP with $\alpha$-divergence policy regularization $\frac{1}{\beta}\Omega^{(\alpha)}_{\pi_0}(\mu)$, which is derived via similar derivations as Lee et al. (2019) or by eliminating $\mu(a, s)$ in Eq. (13).

$$\pi_*(a|s) = \pi_0(a|s)\exp_\alpha\left\{\beta \cdot \left(Q_*(a, s) - V_*(s) + \lambda(a, s) - \psi_{\Delta r_{\pi_*}}(s;\beta)\right)\right\}. \tag{27}$$

We prove Prop. 3 by plugging this optimal policy into the worst-case reward perturbations from Eq. (22), $\Delta r_{\pi_*}(a, s) = \frac{1}{\beta}\log_\alpha \frac{\pi_*(a|s)}{\pi_0(a|s)} + \psi_{\Delta r_{\pi_*}}(s;\beta)$. We can also use Eq. (26) to verify $\pi_*(a|s)$ is normalized, since $\psi_{\Delta r_{\pi_*}}$ ensures normalization for the policy corresponding to $\Delta r_{\pi_*}$. In App. C.3, we also show $\psi_{Q_*}(s;\beta) = V_*(s) + \psi_{\Delta r_{\pi_*}}(s;\beta)$, where $\psi_{Q_*}(s;\beta)$ is a Lagrange multiplier enforcing normalization in Eq. (15). $\square$

**Path Consistency Condition** The equivalence between $\Delta r_{\pi_*}(a, s)$ and $\Delta r_{V_*}(a, s)$ at optimality matches the *path consistency* conditions from (Nachum et al., 2017; Chow et al., 2018) and suggests generalizations to general $\alpha$-divergence regularization. Indeed, combining Eq. (22) and (26) and rearranging,

$$r(a, s) + \gamma \mathbb{E}^{s'}_{a,s}\big[V_*(s')\big] - \frac{1}{\beta}\log_\alpha \frac{\pi_*(a|s)}{\pi_0(a|s)} - \psi_{\Delta r_{\pi_*}}(s;\beta) = V_*(s) - \lambda_*(a, s) \tag{28}$$

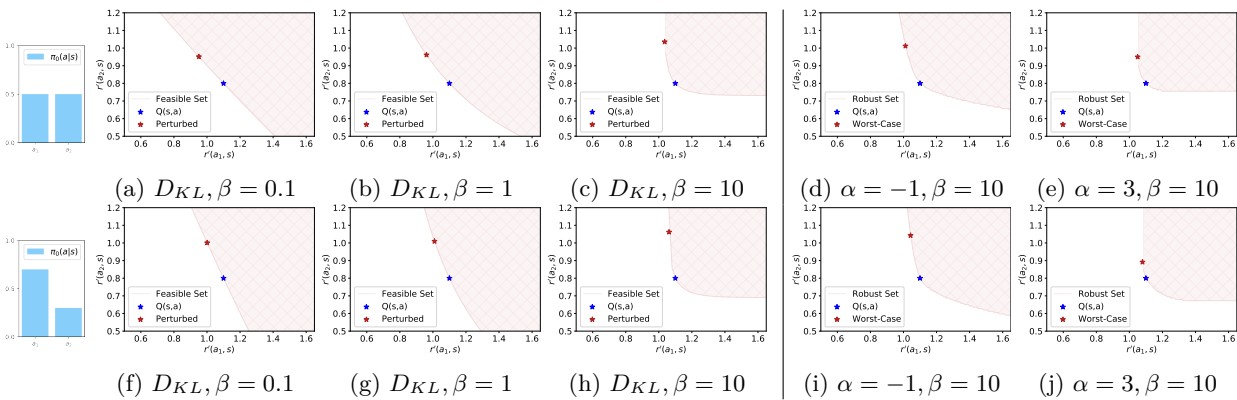

Figure 2: **Robust Set** (red region) of perturbed reward functions to which a stochastic policy generalizes, in the sense that the policy is guaranteed to achieve an expected modified reward greater than or equal to the value of the regularized objective (Eq. (2)). The robust set characterizes the perturbed rewards which are feasible for the adversary. Red stars indicate the worst-case perturbed reward $r'_{\pi_*} = r - \Delta r_{\pi_*}$ (Prop. 2). We show robust sets for the optimal $\pi_*(a|s)$ with fixed $Q(a, s) = r(a, s)$ values (blue star), where the optimal policy differs based on the regularization parameters $\alpha, \beta, \pi_0$ (see Eq. (27)). The robust set is more restricted with decreasing regularization strength (increasing $\beta$), implying decreased generalization. Importantly, the slope of the robust set boundary can be linked to the action probabilities under the policy (see Sec. 4.1).

for all $s \in \mathcal{S}$ and $a \in \mathcal{A}$. This is a natural result, since path consistency is obtained using the KKT optimality condition involving the gradient with respect to $\mu$ of the Lagrangian relaxation in Eq. (13). Similarly, we have seen in Prop. 2 that $\Delta r_\pi = \nabla_\mu \frac{1}{\beta} \Omega_{\pi_0}^{(\alpha)}(\mu)$. See App. A.4.

Path consistency conditions were previously derived for the Shannon entropy (Nachum et al., 2017) and Tsallis entropy with $\alpha = 2$ (Chow et al., 2018), but our expression in Eq. (28) provides a generalization to $\alpha$-divergences with arbitrary reference policies. We provide more detailed discussion in App. E.2.

**Indifference Condition** As Ortega & Lee (2014) discuss for the single step case, the saddle point of the minmax optimization in Eq. (17) reflects an *indifference* condition which is a well-known property of Nash equilibria in game theory (Osborne & Rubinstein, 1994). Consider $Q(a, s) = r(a, s) + \gamma \mathbb{E}_{a,s}^{s'}[V(s')]$ to be the agent's estimated payoff for each action in a particular state. For the optimal policy, value, and worst-case reward perturbations, Eq. (28) shows that the pointwise modified reward $Q_*(a, s) - \Delta r_{\pi_*}(a, s) = V_*(s)$ is equal to a constant.[2] Against the optimal strategy of the adversary, the agent becomes indifferent between the actions in its mixed strategy. The value or conjugate function $V_*(s) = \frac{1}{\beta} \Omega_{\pi_0,\beta}^*(Q_*)$ (see App. C) is known as the *certainty equivalent* (Fishburn, 1988; Ortega & Braun, 2013), which measures the total expected utility for an agent starting in state $s$, in a two-player game against an adversary defined by the regularizer $\Omega$ with strength $\beta$. We empirically confirm the indifference condition in Fig. 3 and 8.

## 4 Experiments

In this section, we visualize the robust set and worst-case reward perturbations associated with policy regularization, using intuitive examples to highlight theoretical properties of our adversarial interpretation.

### 4.1 Visualizing the Robust Set

In Fig. 2, we visualize the robust set of perturbed rewards for the optimal policy in a two-dimensional action space for the KL or $\alpha$-divergence, various $\beta$, and a uniform or non-uniform prior policy $\pi_0$. Since the optimal policy can be easily calculated in the single-step case, we consider fixed $Q_*(a, s) = r(a, s) = \{1.1, 0.8\}$ and show the robustness of the optimal $\pi_*(a|s)$, which differs based on the choice of regularization scheme using Eq. (27). We determine the feasible set of $\Delta r$ using the constraint in Prop. 1 (see App. D.3 for details), and plot the modified reward $r'_{\pi_*}(a, s) = Q_*(a, s) - \Delta r_{\pi_*}(a, s)$ for each action.

Inspecting the constraint for the adversary in Eq. (19), note that both reward increases $\Delta r(a, s) < 0$ and reward decreases $\Delta r(a, s) > 0$ contribute non-negative terms at each action, which either up- or down-weight

---

[2]This holds for actions with $\pi_*(a|s) > 0$ and $\lambda(a, s) = 0$. Note that we treat $Q(a, s)$ as the reward in the sequential case.

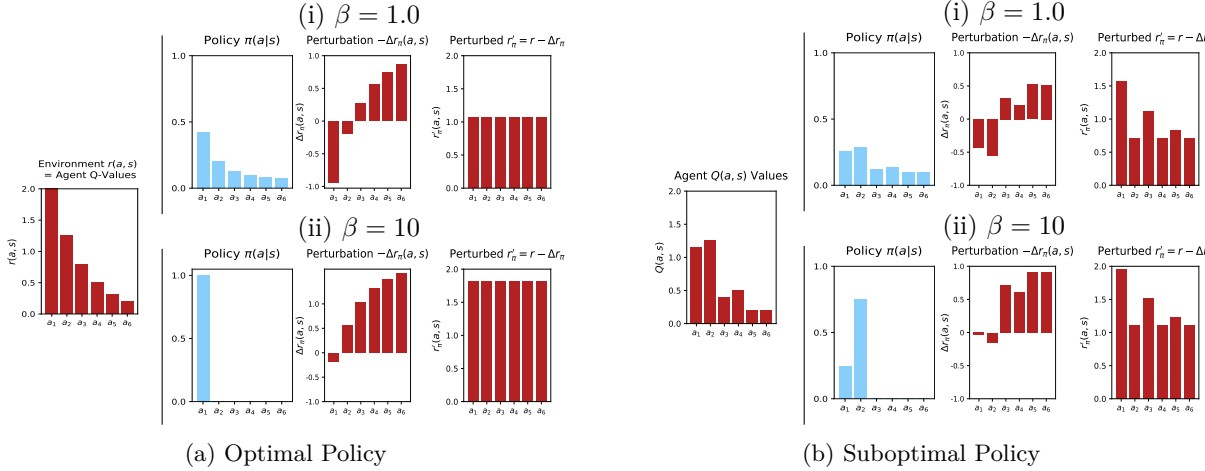

Figure 3: **Single-Step Reward Perturbations** for KL regularization to uniform reference policy $\pi_0(a|s)$. $Q$-values in left columns are used for each $\beta$ in columns 2-4. We report the worst-case $-\Delta r_{\pi_*}(a,s)$ (Eq. (22)), so negative values correspond to reward decreases. **(a)** Optimal policy ($Q_*(a,s) = r(a,s)$) using the environment reward, where the perturbed $r'(a,s) = c \; \forall a$ reflects the indifference condition. **(b)** Suboptimal policy where indifference does not hold. In all cases, actions with high $Q(a,s)$ are robust to reward decreases.

the reference policy $\pi_0(a|s)$. The constraint on their summation forces the adversary to trade off between perturbations of different actions in a particular state. Further, since the constraints in Prop. 1 integrate over the action space, the rewards for *all* actions in a particular state must be perturbed together. While it is clear that increasing the reward in both actions preserves the inequality in Eq. (2), Fig. 2 also includes regions where one reward decreases.

For high regularization strength ($\beta = 0.1$), we observe that the boundary of the feasible set is nearly linear, with the slope $-\frac{\pi_0(a_1|s)}{\pi_0(a_2|s)}$ based on the ratio of action probabilities in a policy that matches the prior. The boundary steepens for lower regularization strength. We can use the indifference condition to provide further geometric insight. First, drawing a line from the origin with slope 1 will intersect the feasible set at the worst-case modified reward (red star) in each panel, with $r'_*(a_1,s) = r'_*(a_2,s)$. At this point, the slope of the tangent line yields the ratio of action probabilities in the regularized policy, as we saw for the $\beta = 0.1$ case. With decreasing regularization as $\beta \to \infty$, the slope approaches 0 or $-\infty$ for a nearly deterministic policy and a rectangular feasible region.

Finally, we show the $\alpha$-divergence robust set with $\alpha \in \{-1, 3\}$ and $\beta = 10$ in Fig. 2 (d)-(e) and (i)-(j), with further visualizations in App. H. Compared to the KL divergence, we find a wider robust set boundary for $\alpha = -1$. For $\alpha = 3$ and $\beta = 10$, the boundary is more strict and we observe much smaller reward perturbations as the optimal policy becomes deterministic ($\pi(a_1|s) = 1$) for both reference distributions. However, in contrast to the unregularized deterministic policy, the reward perturbations $\Delta r_{\pi_*}(a,s) \neq 0$ are nonzero. We provide a worked example in App. G, and note that indifference does not hold in this case, $r'_{\pi_*}(a_1,s) \neq r'_{\pi_*}(a_2,s)$, due to the Lagrange multiplier $\lambda_*(a_2,s) > 0$.

### 4.2 Visualizing the Worst-Case Reward Perturbations

In this section, we consider KL divergence regularization to a uniform reference policy, which is equivalent to Shannon entropy regularization but more appropriate for analysis, as we discuss in Sec. 5.1.

**Single Step Case** In Fig. 3, we plot the *negative* worst-case reward perturbations $-\Delta r_\pi(a,s)$ and modified reward for a single step decision-making case. For the optimal policy in Fig. 3(a), the perturbations match the advantage function as in Eq. (26) and the perturbed reward for all actions matches the value function $V_*(s)$. While we have shown in Sec. 3.2 that any stochastic policy may be given an adversarial interpretation, we see in Fig. 3(b) that the indifference condition does not hold for suboptimal policies.

The nearly-deterministic policy in Fig. 3(a)(ii) also provides intuition for the unregularized case as $\beta \to \infty$. Although we saw in Sec. 3.3 that $\Delta r_{\pi_*}(a,s) = 0 \; \forall a$ in the unregularized case, Eq. (11) and (26) suggest that $\lambda(a,s) = V_*(s) - Q_*(a,s)$ plays a similar role to the (negative) reward perturbations in Fig. 3(a)(ii), with $\lambda(a_1,s) = 0$ and $\lambda(a,s) > 0$ for all other actions.

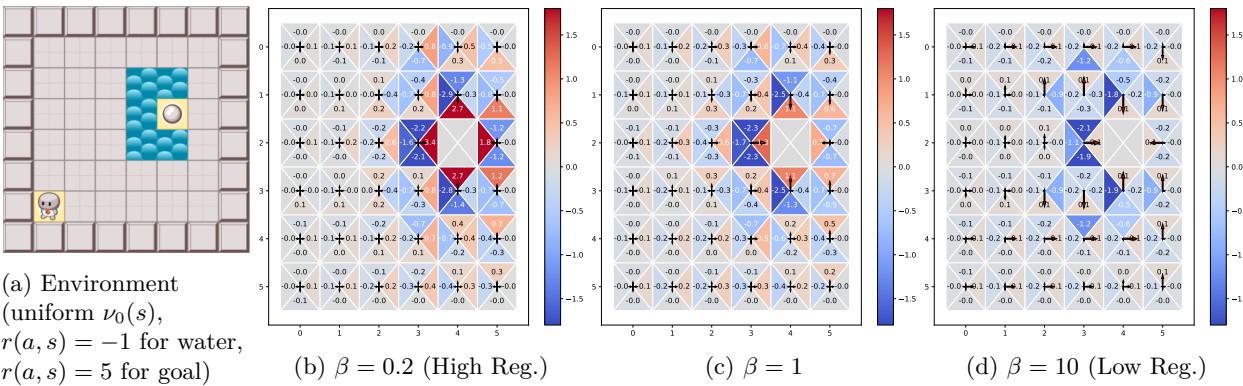

(a) Environment (uniform $\nu_0(s)$, $r(a, s) = -1$ for water, $r(a, s) = 5$ for goal)

(b) $\beta = 0.2$ (High Reg.)

(c) $\beta = 1$

(d) $\beta = 10$ (Low Reg.)

Figure 4: **Grid-World Reward Perturbations. (a)** Sequential task. **(b)-(d)** Policies trained with Shannon entropy regularization of different strength. Action probabilities are indicated via relative arrow lengths; the goal-state is gray and without annotations. Colors indicate worst-case adversarial reward perturbations $\Delta r_\pi(a, s) = \frac{1}{\beta} \log \frac{\pi(a|s)}{\pi_0(a|s)}$ for each state and action (up, down, left, right) against which the policy is robust. Red (or positive $\Delta r_\pi(a, s)$) implies that the policy is robust to reward decreases (up to the value shown) imposed by the adversary. These decreases are balanced by adversarial reward increases (blue) for other actions in the same state. We confirm the optimality of each policy using path consistency in App. H Fig. 8.

**Sequential Setting** In Fig. 4(a), we consider a grid world where the agent receives $+5$ for picking up the reward pill, $-1$ for stepping in water, and zero reward otherwise. We train an agent using tabular $Q$-learning and a discount factor $\gamma = 0.99$. We visualize the worst-case reward perturbations $\Delta r_\pi(a, s) = \frac{1}{\beta} \log \frac{\pi(a|s)}{\pi_0(a|s)}$ in each state-action pair for policies trained with various regularization strengths in Fig. 4(b)-(d). While it is well-known that there exists a unique optimal policy for a given regularized MDP, our results additionally display the adversarial strategies and resulting Nash equilibria which can be associated with a regularization scheme specified by $\Omega$, $\pi_0$, $\alpha$, and $\beta$ in a given MDP.

Each policy implicitly hedges against an adversary that perturbs the rewards according to the values and colormap shown. For example, inspecting the state to the left of the goal state in panel Fig. 4(b)-(c), we see that the adversary reduces the immediate reward for moving right (in red, $\Delta r_{\pi_*} > 0$). Simultaneously, the adversary raises the reward for moving up or down towards the water (in blue). This is in line with the constraints on the feasible set, which imply that the adversary must balance reward decreases with reward increases in each state. In App. E.4 Fig. 8, we certify the optimality of each policy using the path consistency conditions, which also confirms that the adversarial perturbations have rendered the agent indifferent across actions in each state.

Although we observe that the agent with high regularization in Fig. 4(b) is robust to a strong adversary, the value of the regularized objective is also lower in this case. As expected, lower regularization strength reduces robustness to negative reward perturbations. With low regularization in Fig. 4(d), the behavior of the agent barely deviates from the deterministic policy in the face of the weaker adversary.

## 5 Discussion

Our analysis in Sec. 3 unifies and extends several previous works analyzing the reward robustness of regularized policies (Ortega & Lee, 2014; Eysenbach & Levine, 2021; Husain et al., 2021), as summarized in Table 1. We highlight differences in the analysis of entropy-regularized policies in Sec. 5.1, and provide additional discussion of the closely-related work of Derman et al. (2021) in Sec. 5.2.

### 5.1 Comparison with Entropy Regularization

As argued in Sec. 3, the worst-case reward perturbations preserve the value of the regularized objective function. Thus, we should expect our robustness conclusions to depend on the exact form of the regularizer.

When regularizing with the Tsallis or Shannon ($\alpha = 1$) entropy, the worst-case reward perturbations become

$$\Delta r_\pi(a, s) = \frac{1}{\beta} \log_\alpha \pi(a|s) + \frac{1}{\beta} \frac{1}{\alpha} \left( 1 - \sum_{a \in \mathcal{A}} \pi(a|s)^\alpha \right). \tag{29}$$

See App. F.2, we also show that for $0 < \alpha \leq 1$, these perturbations cannot decrease the reward, with $-\Delta r_\pi(a, s) \geq 0$ and $r'_\pi(a, s) \geq r(a, s)$. In the rest of this section, we argue that this property leads to several unsatisfying conclusions in previous work (Lee et al., 2019; Eysenbach & Levine, 2021), which are resolved by using the KL and $\alpha$-divergence for analysis instead of the corresponding entropic quantities.[3]

First, this means that a Shannon entropy-regularized policy is only 'robust' to *increases* in the reward function. However, for useful generalization, we might hope that a policy still performs well when the reward function decreases in at least some states. Including the reference distribution via divergence regularization resolves this issue, and we observe in Fig. 2 and Fig. 4 that the adversary chooses reward decreases in some actions and increases in others. For example, for the KL divergence, $\Delta r_{\pi_*}(a, s) = \frac{1}{\beta} \log \frac{\pi_*(a|s)}{\pi_0(a|s)} = Q_*(a, s) - V_*(s)$ implies robustness to reward decreases when $\pi_*(a|s) > \pi_0(a|s)$ or $Q_*(a, s) > V_*(s)$.

Similarly, Lee et al. (2019) note that for any $\alpha$,

$$\frac{1}{\beta}\Omega_\beta^{*(H_\alpha)}(Q) = \max_{\pi \in \Delta^{|\mathcal{A}|}} \langle \pi, Q \rangle + \frac{1}{\beta}H_\alpha(\pi) \geq Q(a_{\max}, s)$$

where $a_{\max} = \arg\max_a Q(a, s)$ and the Tsallis entropy $H_\alpha(\pi)$ equals the Shannon entropy for $\alpha = 1$. This soft value aggregation yields a result that is *larger* than any particular $Q$-value. By contrast, for the $\alpha$-divergence, we show in App. F.3 that for fixed $\beta$ and $\alpha > 0$,

$$Q(a_{\max}, s) + \frac{1}{\beta}\frac{1}{\alpha}\log_{2-\alpha} \pi(a_{\max}|s) \leq \frac{1}{\beta}\Omega_{\pi_0,\beta}^{*(\alpha)}(Q) \leq Q(a_{\max}, s). \tag{30}$$

This provides a more natural interpretation of the Bellman optimality operator $V(s) \leftarrow \frac{1}{\beta}\Omega_{\pi_0,\beta}^{*(\alpha)}(Q)$ as a soft maximum operation. As a function of $\beta$, we see in App. C.4 and F.3 that the conjugate ranges between $\mathbb{E}_{\pi_0}[Q(a_{\max}, s)] \leq \frac{1}{\beta}\Omega_{\pi_0,\beta}^{*(\alpha)}(Q) \leq Q(a_{\max}, s)$.

Finally, using entropy instead of divergence regularization also affects interpretations of the feasible set. Eysenbach & Levine (2021) consider the same constraint as in Eq. (19), but without the reference $\pi_0(a|s)$

$$\sum_{a \in \mathcal{A}} \exp\left\{\beta \cdot \Delta r(a, s)\right\} \leq 1 \quad \forall s \in \mathcal{S}. \tag{31}$$

This constraint suggests that the original reward function ($\Delta r = 0$) is not feasible for the adversary. More surprisingly, Eysenbach & Levine (2021) App. A8 argues that *increasing* regularization strength (with lower $\beta$) may lead to *less* robust policies based on the constraint in Eq. (31). In App. F.4, we discuss how including $\pi_0(a|s)$ in the constraint via divergence regularization (Prop. 1) avoids this conclusion. As expected, Fig. 2 shows that increasing regularization strength leads to more robust policies.

## 5.2 Related Algorithms

Several recent works provide algorithmic insights which build upon convex duality and complement or extend our analysis. Derman et al. (2021) derive practical iterative algorithms based on a general equivalence between robustness and regularization, which can be used to enforce robustness to *both* reward perturbations (through policy regularization) and changes in environment dynamics (through value regularization). For policy regularization, Derman et al. (2021) translate the specification of a desired robust set into a regularizer using the convex conjugate of the set indicator function. In particular, Derman et al. (2021) associate KL divergence or (scaled) Tsallis entropy policy regularization with the robust set $\mathcal{R}_\pi^\Delta := \{\Delta r \,|\, \Delta r(a, s) \in [\frac{1}{\beta}\frac{1}{\alpha}\log_\alpha \frac{\pi(a|s)}{\pi_0(a|s)}, \infty) \,\forall\, (a, s) \in \mathcal{A} \times \mathcal{S}\}$. Our analysis proceeds in the opposite direction, from regularization to robustness, using the conjugate of the divergence. While the worst-case perturbations result in the same modified objective, our approach yields a larger robust set with qualitatively different shape (see Fig. 1).

Zahavy et al. (2021) analyze a general 'meta-algorithm' which alternates between updates of the occupancy measure $\mu(a, s)$ and modified reward $r'(a, s)$ in online fashion. This approach highlights the fact that the modified reward $r'_\pi$ or worst-case perturbations $\Delta r_\pi$ change as the policy or occupancy measure is optimized. The results of Zahavy et al. (2021) and Husain et al. (2021) hold for general convex MDPs, which encompass common exploration and imitation learning objectives beyond the policy regularization setting we consider.

---

[3]Entropy regularization corresponds to divergence regularization with the uniform reference distribution $\pi_0(a|s)$.

As discussed in Sec. 3.4, path consistency conditions have been used to derive practical learning objectives in (Nachum et al., 2017; Chow et al., 2018). These algorithms might be extended to general $\alpha$-divergence regularization via Eq. (28), which involves an arbitrary reference policy $\pi_0(a|s)$ that can be learned adaptively as in (Teh et al., 2017; Grau-Moya et al., 2018).

Finally, previous work has used dual optimizations similar to Eq. (14) to derive alternative Bellman error losses (Dai et al., 2018; Belousov & Peters, 2019; Nachum & Dai, 2020; Bas-Serrano et al., 2021), highlighting how convex duality can be used to bridge between policy regularization and Bellman error aggregation (Belousov & Peters, 2019; Husain et al., 2021).

## 6 Conclusion

In this work, we analyzed the robustness of convex-regularized RL policies to worst-case perturbations of the reward function, which implies generalization to adversarially chosen reward functions from within a particular robust set. We have characterized this robust set of reward functions for KL and $\alpha$-divergence regularization, provided a unified discussion of existing works on reward robustness, and clarified apparent differences in robustness arising from entropy versus divergence regularization. Our advantage function interpretation of the worst-case reward perturbations provides a complementary perspective on how $Q$-values appear as dual variables in convex programming forms of regularized MDPs. Compared to a deterministic, unregularized policy, a stochastic, regularized policy places probability mass on a wider set of actions and requires state-action value adjustments via the advantage function or adversarial reward perturbations. Conversely, a regularized agent, acting based on given $Q$-value estimates, implicitly hedges against the anticipated perturbations of an appropriate adversary.

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

# Appendix

## Table of Contents

# A  Implications of Conjugate Duality Optimality Conditions

In this section, we show several closely-related results which are derived from the conjugate optimality conditions. We provide additional commentary in later Appendix sections which more closely follow the sequence of the main text.

First, recall from Section 2.1 the definition of the conjugate optimizations for functions over $\mathcal{X} \coloneqq \mathcal{A} \times \mathcal{S}$. We restrict $\mu \in \mathbb{R}_+^{\mathcal{A} \times \mathcal{S}}$ to be a nonnegative function over $\mathcal{X}$, so that

$$\frac{1}{\beta}\Omega^*(\Delta r) = \sup_{\mu \in \mathbb{R}_+^{\mathcal{A} \times \mathcal{S}}} \langle \mu, \Delta r \rangle - \frac{1}{\beta}\Omega(\mu), \qquad \frac{1}{\beta}\Omega(\mu) = \sup_{\Delta r \in \mathbb{R}^{\mathcal{A} \times \mathcal{S}}} \langle \mu, \Delta r \rangle - \frac{1}{\beta}\Omega^*(\Delta r), \tag{32}$$

and the implied optimality conditions are

$$\Delta r = \frac{1}{\beta}\nabla_\mu \Omega(\mu) = \left(\nabla_{\Delta r} \frac{1}{\beta}\Omega^*\right)^{-1}(\mu) \qquad \mu = \frac{1}{\beta}\nabla_{\Delta r}\Omega^*(\Delta r) = \left(\nabla_\mu \frac{1}{\beta}\Omega\right)^{-1}(\Delta r). \tag{33}$$

## A.1  Proof of Prop. 2 : Policy Form Worst-Case Reward Perturbations

**Proposition 2.** *For a given policy $\pi(a|s)$ or state-action occupancy $\mu(a,s)$, the worst-case adversarial reward perturbations $\Delta r_\pi$ or $\Delta r_\mu$ associated with a convex function $\Omega(\mu)$ and regularization strength $1/\beta$ are*

$$\Delta r_\pi = \nabla_\mu \frac{1}{\beta}\Omega(\mu). \tag{20}$$

*Proof.* The reward perturbations are defined via conjugate optimization for $\Omega(\mu)$ in Eq. (32), where $\Delta r \in \mathbb{R}^{\mathcal{A} \times \mathcal{S}}$. The proposition follows directly from the optimality condition in Eq. (33), and we focus on the $\Delta r = \frac{1}{\beta}\nabla_\mu \Omega(\mu)$ condition for convenience. $\qquad\square$

In App. B, we derive the explicit forms for the worst-case reward perturbations for KL and $\alpha$-divergence regularization from Sec. 3.2 of the main text. See App. B Table 3 for references to particular derivations.

Note that we do not consider further constraints on $\mu$ in the conjugate optimization. Instead, we view the Bellman flow constraints $\mu(a,s) \in \mathcal{M}$ (and normalization constraint $\mu(a,s) \in \Delta^{|\mathcal{A}| \times |\mathcal{S}|}$) as arising from the overall (regularized) MDP optimization in Eq. (10) or (12), as we discuss in the next subsection.

## A.2  Optimal Policy in a Regularized MDP

In Lemma 1 below, we show that the Bellman flow constraints Eq. (10), which are enforced by the optimal Lagrange multipliers $V_*(s)$, ensure that the optimal $\mu_*(a,s)$ is normalized. This suggests that an explicit normalization constraint is not required. In Prop. 4, we then proceed to derive the optimal policy in a regularized MDP using the conjugate optimality conditions in Eq. (33).

**Lemma 1** (Flow Constraints Ensure Normalization). *Assume that the initial state distribution $\nu_0(s)$ and transition dynamics $P(s'|a,s)$ are normalized, with $\sum_s \nu_0(s) = 1$ and $\sum_{s'} P(s'|a,s) = 1$. If a state-occupancy measure satisfies the Bellman flow constraints $\mu(a,s) \in \mathcal{M}$, then it is a normalized distribution $\mu(a,s) \in \Delta^{|\mathcal{A}| \times |\mathcal{S}|}$.*

*Proof.* Starting from the Bellman flow constraints $\sum_a \mu(a,s) = (1-\gamma)\nu_0(s) + \gamma \sum_{a',s'} P(s|a',s')\mu(a',s')$, we consider taking the summation over states $s \in \mathcal{S}$,

$$\sum_{a,s} \mu(a,s) = (1-\gamma)\sum_s \nu_0(s) + \gamma \sum_s \sum_{a',s'} P(s|a',s')\mu(a',s') \overset{(1)}{=} (1-\gamma) + \gamma \sum_s P(s|a',s') \sum_{a',s'} \mu(a',s') \overset{(2)}{=} (1-\gamma) + \gamma \sum_{a',s'} \mu(a',s')$$

where (1) uses the normalization assumption on $\nu_0(s)$ and the distributive law, and (2) uses the normalization assumption on $P(s|a',s')$. Finally, we rearrange the first and last equality to obtain

$$(1-\gamma)\sum_{a,s} \mu(a,s) = (1-\gamma) \qquad \Longrightarrow \qquad \sum_{a,s} \mu(a,s) = 1 \tag{34}$$

which shows that $\mu(a,s)$ is normalized as a joint distribution over $a \in \mathcal{A}, s \in \mathcal{S}$, as desired. $\qquad\square$

**Proposition 4** (Optimal Policy in Regularized MDP). *Given the optimal value function $V_*(s)$ and Lagrange multipliers $\lambda_*(a,s)$, the optimal policy in the regularized MDP is given by*

$$\mu_* = \frac{1}{\beta}\nabla\Omega^*\left(r + \mathbb{E}_{a,s}^{s'}\big[V_*\big] - V_* + \lambda_*\right) = \left(\nabla_\mu \frac{1}{\beta}\Omega\right)^{-1}\left(r + \mathbb{E}_{a,s}^{s'}\big[V_*\big] - V_* + \lambda_*\right).$$

*This matches the conjugate conditions in Eq. (33) using the arguments $\Delta r(a,s) \leftarrow r(a,s) + \mathbb{E}_{a,s}^{s'}\big[V_*(s')\big] - V_*(s) + \lambda_*(a,s)$.*

*Proof.* In Sec. 2.4, we moved from the regularized primal optimization (Eq. (12)) to the dual optimization (Eq. (14)) via the regularized Lagrangian

$$\min_{V,\lambda} \max_\mu (1 - \gamma)\langle \nu_0, V\rangle + \langle \mu, r + \gamma\mathbb{E}_{a,s}^{s'}\big[V\big] - V + \lambda\rangle - \frac{1}{\beta}\Omega_{\pi_0}(\mu)$$

Note that the Lagrange multipliers $\lambda(a,s)$ enforce $\mu(a,s) \geq 0$ while $V(s)$ enforces the flow constraints and thus, by Lemma 1, normalization of $\mu(a,s)$. We recognized the final two terms as a conjugate optimization

$$\frac{1}{\beta}\Omega_{\pi_0,\beta}^*\left(r + \gamma\mathbb{E}_{a,s}^{s'}\big[V\big] - V + \lambda\right) = \max_\mu \langle\mu, r + \gamma\mathbb{E}_{a,s}^{s'}\big[V\big] - V + \lambda\rangle - \frac{1}{\beta}\Omega_{\pi_0}(\mu) \qquad (35)$$

to yield a dual optimization over $V(s)$ and $\lambda(a,s)$ only in Eq. (14). After solving the dual optimization for the optimal $V_*(s), \lambda_*(a,s)$, we can recover the optimal policy in the MDP using the optimizing argument of Eq. (35). Differentiating Eq. (35) and solving for $\mu$ yields $\nabla_\mu \frac{1}{\beta}\Omega(\mu) = r + \gamma\mathbb{E}_{a,s}^{s'}\big[V\big] - V + \lambda$ which we invert to obtain Prop. 4. The other equality follows from the conjugate optimality conditions in Eq. (33). $\qquad\square$

For $\alpha$-divergence regularization, the optimal policy or state-action occupancy is given by the 'optimizing argument' column of Table 2, up to reparameterization of $\Delta r(a,s) \leftarrow r(a,s) + \mathbb{E}_{a,s}^{s'}\big[V_*(s')\big] - V_*(s) + \lambda_*(a,s)$ as the dual variable. In this case, note that the argument to the conjugate function accounts for the flow and nonnegativity constraints via $V_*(s)$ and $\lambda_*(a,s)$. In particular, we have

Policy Reg., App. B.3 Eq. (63)

$$\mu_*(a,s) = \mu(s)\pi_0(a|s)\exp_\alpha\left\{\beta \cdot \left(r(a,s) + \gamma\mathbb{E}_{a,s}^{s'}\big[V_*(s')\big] - V_*(s) + \lambda(a,s) - \psi_{\Delta r_{\pi_*}}(s;\beta)\right)\right\} \quad (36)$$

Occupancy Reg., App. B.4 Eq. (68)

$$\mu_*(a,s) = \mu_0(a,s)\exp_\alpha\left\{\beta \cdot \left(r(a,s) + \gamma\mathbb{E}_{a,s}^{s'}\big[V_*(s')\big] - V_*(s) + \lambda(a,s)\right)\right\} \qquad (37)$$

where $\psi_{\Delta r_{\pi_*}}(s;\beta) = \frac{1}{\beta}\frac{1}{\alpha}(1 - \sum_a \pi_0(a|s)^{1-\alpha}\pi_*(a|s)^\alpha)$ appears from differentiating $\nabla\frac{1}{\beta}\Omega_{\pi_0}(\mu)$ as in Eq. (23) or App. B.3. This means that the optimal policy is only available in self-consistent fashion, with the normalization constant inside the $\exp_\alpha$, which can complicate practical applications (Lee et al., 2019; Chow et al., 2018).

### A.3 Proof of Prop. 3: Policy Form Worst-Case Perturbations match Value Form at Optimality

The substitution $\Delta r(a,s) \leftarrow r(a,s) + \mathbb{E}_{a,s}^{s'}\big[V_*(s')\big] - V_*(s) + \lambda_*(a,s)$ above already anticipates the result in Prop. 3, which links the reward perturbations for the optimal policy $\Delta r_{\pi_*}$ or state-action occupancy $\Delta r_{\mu_*}$ to the advantage function. See the proof of Thm. 1 in App. E.1 for additional context in relation to the value-form reward perturbations $\Delta r_V(a,s)$.

**Proposition 3.** *For the optimal policy $\pi_*(a|s)$ and value function $V_*(s)$ corresponding to $\alpha$-divergence policy regularization with strength $\beta$, the policy and value forms of the worst-case adversarial reward perturbations match, $\Delta r_{\pi_*} = \Delta r_{V_*}$, and are related to the advantage function via*

$$\Delta r_{\pi_*}(a,s) = Q_*(a,s) - V_*(s) + \lambda_*(a,s), \qquad (26)$$

*where we define $Q_*(a,s) \coloneqq r(a,s) + \gamma\mathbb{E}_{a,s}^{s'}\big[V_*(s')\big]$ and recall $\lambda_*(a,s)\pi_*(a|s) = 0$ by complementary slackness. Note that $V_*(s)$ depends on the regularization scheme via the conjugate function $\frac{1}{\beta}\Omega_{\pi_0,\beta}^{*(\alpha)}(\Delta r_V)$ in Eq. (25).*

*Proof.* The result follows by combining Prop. 2, which states that $\Delta r_\pi = \nabla_\mu \frac{1}{\beta}\Omega(\mu)$, and Prop. 4, which implies $\nabla_\mu \frac{1}{\beta}\Omega(\mu_*) = r(a,s) + \gamma\mathbb{E}_{a,s}^{s'}[V_*(s')] - V_*(s) + \lambda_*(a,s)$ as a condition for optimality of $\{\mu_*(a,s), V_*(s), \lambda_*(a,s)\}$. Thus, for the optimal policy $\pi_*(a|s)$ and Lagrange multipliers $\{V_*(s), \lambda_*(a,s)\}$, we have $\Delta r_{\pi_*}(a,s) = r(a,s) + \gamma\mathbb{E}_{a,s}^{s'}[V_*(s')] - V_*(s) + \lambda_*(a,s)$ and similarly for $\Delta r_{\mu_*}(a,s)$.

We can confirm this using the expression for the optimal policy in Eq. (36) and the worst-case reward perturbations in Sec. 3.2. For example, recalling $\mu_*(a,s) = \mu(s)\pi_*(a|s)$, we can write the $\alpha$-divergence policy regularization case as $\Delta r_{\pi_*}(a,s) = \frac{1}{\beta}\log_\alpha\frac{\pi_*(a|s)}{\pi_0(a|s)} + \psi_{\Delta r_{\pi_*}}(s;\beta) = r(a,s) + \gamma\mathbb{E}_{a,s}^{s'}[V_*(s')] - V_*(s) + \lambda(a,s) \pm \psi_{\Delta r_{\pi_*}}(s;\beta)$. $\qquad\square$

## A.4 Path Consistency and KKT Conditions

Finally, note that the KKT optimality conditions (Boyd & Vandenberghe, 2004) include the condition that we have used in the proof of Prop. 3. At optimality, we have

$$r(a,s) + \gamma\mathbb{E}_{a,s}^{s'}[V_*(s')] - V_*(s) + \lambda_*(a,s) - \nabla\frac{1}{\beta}\Omega(\mu^*) = 0. \tag{38}$$

This KKT condition is used to derive path consistency objectives in Nachum et al. (2017); Chow et al. (2018).

For general $\alpha$-divergence policy regularization, we substitute $\nabla\frac{1}{\beta}\Omega_{\pi_0}^{(\alpha)}(\mu_*) = \Delta r_{\pi_*}(a,s) = \frac{1}{\beta}\log_\alpha\frac{\pi_*(a|s)}{\pi_0(a|s)} + \psi_{\Delta r_{\pi_*}}(s;\beta)$ using Eq. (22) (see App. B.3 for detailed derivations). This leads to the condition

$$r(a,s) + \gamma\mathbb{E}_{a,s}^{s'}[V_*(s')] - V_*(s) + \lambda_*(a,s) - \frac{1}{\beta}\log_\alpha\frac{\pi_*(a|s)}{\pi_0(a|s)} - \psi_{\Delta r_{\pi_*}}(s;\beta) = 0, \tag{39}$$

which matches Eq. (28). We compare our $\alpha$-divergence path consistency conditions to previous work in App. E.2.

## A.5 Modified Rewards and Duality Gap for Suboptimal Policies

We can also use the conjugate duality of state-action occupancy measures and reward functions ($r(a,s)$ or $r'(a,s)$) to express the optimality gap for a suboptimal $\mu(a,s)$. Consider the regularized primal objective as a (constrained) conjugate optimization,

$$\mathcal{RL}_{\Omega,\beta}(r) := \frac{1}{\beta}\Omega^*(r) = \max_{\mu\in\mathcal{M}}\langle\mu,r\rangle - \frac{1}{\beta}\Omega(\mu) \tag{40}$$

$$\geq \langle\mu_{r'},r\rangle - \frac{1}{\beta}\Omega(\mu_{r'}) \tag{41}$$

where the inequality follows from the fact that any feasible $\mu_{r'} \in \mathcal{M}$ provides a lower bound on the objective. We use the notation $\mu_{r'}$ to anticipate the fact that, assuming appropriate domain considerations, we would like to associate this occupancy measure with a modified reward function $r'$ using the conjugate optimality conditions in Eq. (33) (with $r'$ as the dual variable). In particular, for a given $\Omega$, we use the fact that $\mu_{r'} = \frac{1}{\beta}\nabla\Omega^*(r')$ to recognize the conjugate duality gap as a Bregman divergence. Rearranging Eq. (41),

$$\frac{1}{\beta}\Omega^*(r) - \langle\mu_{r'},r\rangle + \frac{1}{\beta}\Omega(\mu_{r'}) \geq 0 \tag{42}$$

$$\frac{1}{\beta}\Omega^*(r) - \langle\mu_{r'},r\rangle + \langle\mu_{r'},r'\rangle - \frac{1}{\beta}\Omega^*(r') \geq 0 \tag{43}$$

$$\frac{1}{\beta}\Omega^*(r) - \frac{1}{\beta}\Omega^*(r') - \langle r - r', \underbrace{\frac{1}{\beta}\nabla\Omega^*(r')}_{\mu_{r'}}\rangle \geq 0 \tag{44}$$

$$D_{\Omega^*}[r:r'] \geq 0 \tag{45}$$

where the last line follows from the definition of the Bregman divergence (Amari, 2016). For example, using the KL divergence $\Omega(\mu) = D_{KL}[\mu:\mu_0]$, one can confirm that the Bregman divergence generated by $\Omega^*$ is also a KL divergence, $D_{KL}[\mu_{r'}:\mu_{r^*}]$ (Belousov, 2017; Banerjee et al., 2005).

# B  Convex Conjugate Derivations

In this section, we derive the convex conjugate associated with KL and $\alpha$-divergence regularization of the policy $\pi(a|s)$ or state-action occupancy $\mu(a,s)$. We summarize these results in Table 2, with equation references in Table 3. In both cases, we treat the regularizer $\frac{1}{\beta}\Omega(\mu)$ as a function of $\mu(a,s)$ and optimize over all states jointly,

$$\frac{1}{\beta}\Omega^*(\cdot) = \sup_{\mu \in \mathbb{R}_+^{\mathcal{A} \times \mathcal{S}}} \langle \mu, \cdot \rangle - \frac{1}{\beta}\Omega(\mu). \tag{46}$$

These conjugate derivations can be used to reason about the optimal policy via $\frac{1}{\beta}\Omega^*\big(r + \gamma \mathbb{E}_{a,s}^{s'}[V] - V - \lambda\big)$, as argued in App. A.2, or the worst case reward perturbations using $\frac{1}{\beta}\Omega^*(\Delta r)$. We use $\Delta r$ as the argument or dual variable throughout this section.

In App. C, we derive alternative conjugate functions which optimize over the policy in each state, where $\pi(a|s) \in \Delta^{|\mathcal{A}|}$ is constrained to be a normalized probability distribution. This conjugate arises in considering soft value aggregation or regularized iterative algorithms as in Sec. 2.4. See Table 4 for equation references.

| Divergence $\Omega$ | $\frac{1}{\beta}\Omega^*(\Delta r)$ | $\Delta r_\mu(a,s)$ | $\mu_{\Delta r}(a,s)$ |
|---|---|---|---|
| $\frac{1}{\beta}D_{KL}[\pi:\pi_0]$ | Eq. (47) | Eq. (50) | Eq. (51) |
| $\frac{1}{\beta}D_{KL}[\mu:\mu_0]$ | Eq. (55) | Eq. (53) | Eq. (54) |
| $\frac{1}{\beta}D_\alpha[\pi_0:\pi]$ | Eq. (56) | Eq. (62) | Eq. (63) |
| $\frac{1}{\beta}D_\alpha[\mu_0:\mu]$ | Eq. (66) | Eq. (67) | Eq. (68) |

| Divergence $\Omega$ | $\frac{1}{\beta}\Omega^*(Q)$ | $Q_\pi(a,s)$ | $\pi_Q(a,s)$ |
|---|---|---|---|
| $\frac{1}{\beta}D_{KL}[\pi:\pi_0]$ | Eq. (78) | Eq. (75) | Eq. (76) |
| $\frac{1}{\beta}D_\alpha[\pi_0:\pi]$ | Eq. (82) | Eq. (81) | Eq. (80) |

Table 3: Equations for $\Delta r$ or 'MDP Optimality' Conjugate ($\mu$ Optimization, No Normalization Constraint)

Table 4: Equations for 'Soft Value' $V_*(s)$ Conjugates ($\pi$ Optimization, Normalization Constraint)

## B.1  KL Divergence Policy Regularization: $\frac{1}{\beta}\Omega^*_{\pi_0,\beta}(\Delta r)$

The conjugate function for KL divergence from the policy $\pi(a|s)$ to a reference $\pi_0(a|s)$ has the following closed form

$$\frac{1}{\beta}\Omega^*_{\pi_0,\beta}(\Delta r) = \frac{1}{\beta}\sum_s \mu(s)\left(\sum_s \pi_0(a|s)\exp\big\{\beta \cdot \Delta r(a,s)\big\} - 1\right). \tag{47}$$

*Proof.* We start from the optimization in Eq. (3) or (32), using conditional KL divergence regularization $\Omega_{\pi_0}(\mu) = \mathbb{E}_{\mu(s)}[D_{KL}[\pi:\pi_0]]$ as in Eq. (6).

$$\frac{1}{\beta}\Omega^*_{\pi_0,\beta}(\Delta r) = \max_\mu \langle \mu, \Delta r \rangle - \frac{1}{\beta}\sum_{a,s} \mu(a,s)\log\frac{\mu(a,s)}{\mu(s)\pi_0(a|s)} + \frac{1}{\beta}\sum_{a,s}\mu(a,s) - \frac{1}{\beta}\sum_{a,s}\mu(s)\pi_0(a|s) \tag{48}$$

$$\implies \Delta r = \nabla_\mu \left(\frac{1}{\beta}\sum_{a,s}\mu(a,s)\log\frac{\mu(a,s)}{\mu(s)\pi_0(a|s)} + \frac{1}{\beta}\sum_{a,s}\mu(a,s) - \frac{1}{\beta}\sum_{a,s}\mu(s)\pi_0(a|s)\right) \tag{49}$$

**Worst-Case Reward Perturbations** $\Delta r_\pi(a|s)$  We can recognize Eq. (49) as an instance of Prop. 2. Noting that the marginal $\mu(s)$ depends on $\mu(a,s)$, we make use of the identity $\sum_{s'}\frac{\partial}{\partial\mu(a,s)}\mu(s') = \sum_{s',a'}\frac{\partial}{\partial\mu(a,s)}\mu(a',s') = \sum_{s',a'}\delta(a',s' = a,s) = 1$ as in (Neu et al., 2017; Lee et al., 2019). Differentiat-

ing, we obtain

$$\Delta r(a,s) = \frac{1}{\beta} \log \frac{\mu(a,s)}{\mu(s)\pi_0(a|s)} - \frac{1}{\beta} \underbrace{\sum_{a,s} \frac{\partial \mu(a,s)}{\partial \mu(a',s')}}_{1} + \frac{1}{\beta} \sum_{a,s} \frac{\mu(a,s)}{\mu(s)} \underbrace{\frac{\partial \sum_{a''} \mu(a'',s)}{\partial \mu(a',s')}}_{\delta(s=s')} + \frac{1}{\beta} - \frac{1}{\beta} \sum_{a,s} \underbrace{\frac{\partial \sum_{a''} \mu(a'',s)}{\partial \mu(a',s')}}_{\delta(s=s')} \pi_0(a|s)$$

$$= \frac{1}{\beta} \log \frac{\mu(a,s)}{\mu(s)\pi_0(a|s)} + \frac{1}{\beta} \sum_a \frac{\mu(a,s)}{\mu(s)} - \frac{1}{\beta} \sum_a \pi_0(a|s)$$

$$= \frac{1}{\beta} \log \frac{\mu(a,s)}{\mu(s)\pi_0(a|s)} . \tag{50}$$

In the last line, we assume $\sum_a \pi_0(a|s) = 1$ and note that $\sum_a \frac{\mu(a,s)}{\mu(s)} = \frac{\sum_a \mu(a,s)}{\mu(s)} = \frac{\mu(s)}{\mu(s)} = 1$.

**Optimizing Argument $\pi_{\Delta r}(a|s)$**  We derive the conjugate function by solving for the optimizing argument $\mu(a,s)$ in terms of $\Delta r(a,s)$ and substituting back into Eq. (48). Defining $\pi_{\Delta r}(a|s) = \frac{\mu_{\Delta r}(a,s)}{\mu_{\Delta r}(s)}$ as the policy induced by the optimizing argument $\mu_{\Delta r}(a,s)$ in Eq. (50), we can solve for $\pi_{\Delta r}$ to obtain

$$\pi_{\Delta r}(a|s) = \pi_0(a|s) \exp \left\{ \beta \cdot \Delta r(a,s) \right\} \tag{51}$$

**Conjugate Function $\frac{1}{\beta}\Omega^*_{\pi_0,\beta}(\Delta r)$**  We plug this back into the conjugate optimization Eq. (48), with $\mu_{\Delta r}(a,s) = \mu(s)\pi_{\Delta r}(a,s)$. Assuming $\pi_0(a|s)$ is normalized, we also have $\sum_{a,s} \mu(s)\pi_0(a|s) = 1$ and

$$\frac{1}{\beta}\Omega^*_{\pi_0,\beta}(\Delta r) = \sum_{a,s} \mu(s)\pi_{\Delta r}(a|s) \cdot \Delta r(a,s) - \frac{1}{\beta}\Big( \sum_{a,s} \mu(s)\pi_{\Delta r}(a|s) \cdot \log \frac{\pi_0}{\pi_0}\exp\{\beta\Delta r(a,s)\} - \mu(s)\pi_0(a|s)\exp\left\{\beta\Delta r(a,s)\right\} + \mu(s)\pi_0(a|s) \Big)$$

$$= \frac{1}{\beta} \sum_s \mu(s) \left( \sum_s \pi_0(a|s)\exp\left\{ \beta \cdot \Delta r(a,s) \right\} - 1 \right) \tag{52}$$

as desired. Note that the form of the conjugate function also depends on the regularization strength $\beta$.

Finally, we verify that our other conjugate optimality condition $\Delta r_\pi = \left( \nabla_{\Delta r} \frac{1}{\beta}\Omega^*_{\pi_0,\beta} \right)^{-1}(\mu_{\Delta r})$, or $\mu_{\Delta r} = \nabla_{\Delta r} \frac{1}{\beta}\Omega^*_{\pi_0,\beta}(\Delta r_\pi)$ holds for this conjugate function. Indeed, differentiating with respect to $\Delta r(a,s)$ above, we see that $\frac{\partial}{\partial \Delta r(a,s)} \frac{1}{\beta}\Omega^*_{\pi_0,\beta}(\Delta r) = \mu(s)\pi_0(a|s)\exp\{\beta \cdot \Delta r(a,s)\}$ matches $\mu_{\Delta r}(a,s) = \mu(s)\pi_{\Delta r}(a|s)$ via Eq. (51). □

Although our regularization $\Omega_{\pi_0}(\mu) = \mathbb{E}_{\mu(s)}[D_{KL}[\pi : \pi_0]]$ applies at each $\pi_0(a|s)$, we saw that performing the conjugate optimization over $\mu(a,s)$ led to an expression for a policy $\pi_{\Delta r}(a|s) = \mu_{\Delta r}(a,s)/\mu(s)$ that is normalized *by construction* $\sum_a \pi_{\Delta r}(a|s) = \frac{\sum_a \mu_{\Delta r}(a,s)}{\mu(s)} = 1$. Conversely, for a given normalized $\pi(a|s)$, the above conjugate conditions yield $\Delta r_\pi(a,s)$ such that Eq. (51) is also normalized.

## B.2  KL Divergence Occupancy Regularization: $\frac{1}{\beta}\Omega^*_{\mu_0,\beta}(\Delta r)$

Nearly identical derivations as App. B.1 apply when regularizing the divergence $\Omega_{\mu_0}(\mu) = D_{KL}[\mu : \mu_0]$ between the joint state-action occupancy $\mu(a,s)$ and a reference $\mu_0(a,s)$. This leads to the following results

**Worst-Case Perturbations:** $\qquad \Delta r_\mu(a,s) = \frac{1}{\beta} \log \frac{\mu(a,s)}{\mu_0(a,s)} \tag{53}$

**Optimizing Argument:** $\qquad \mu_{\Delta r}(a,s) = \mu_0(a,s) \exp \left\{ \beta \cdot \Delta r(a,s) \right\} . \tag{54}$

**Conjugate Function:** $\qquad \frac{1}{\beta}\Omega^*_{\mu_0,\beta}(\Delta r) = \frac{1}{\beta} \sum_{a,s} \mu_0(a,s) \exp \left\{ \beta \cdot \Delta r(a,s) \right\} - \frac{1}{\beta} \sum_{a,s} \mu_0(a,s) \tag{55}$

Such regularization schemes appear in REPS (Peters et al., 2010), while Bas-Serrano et al. (2021) consider both policy and occupancy regularization.

## B.3 $\alpha$-**Divergence Policy Regularization:** $\frac{1}{\beta}\Omega_{\pi_0,\beta}^{*(\alpha)}(\Delta r)$

The conjugate for $\alpha$-divergence regularization of the policy $\pi(a|s)$ to a reference $\pi_0(a|s)$ takes the form

$$\frac{1}{\beta}\Omega_{\pi_0,\beta}^{*(\alpha)}(\Delta r) = \frac{1}{\beta}\frac{1}{\alpha}\sum_{a,s}\mu(s)\left(\pi_0(a|s)\left[1+\beta(\alpha-1)\big(\Delta r(a,s)-\psi_{\Delta r}(s;\beta)\big)\right]^{\frac{\alpha}{\alpha-1}}-1\right)+\sum_s\mu(s)\,\psi_{\Delta r}(s;\beta). \quad (56)$$

where $\psi_{\Delta r}(s;\beta)$ is a normalization constant for the optimizing argument $\pi_{\Delta r}(a|s)$ corresponding to $\Delta r(a,s)$.

We provide explicit derivations of the conjugate function instead of leveraging $f$-divergence duality (Belousov & Peters, 2019; Nachum & Dai, 2020) in order to account for the effect of optimization over the joint distribution $\mu(a,s)$. We will see in App. C.2 see that the conjugate in Eq. (56) takes a similar to form as the conjugate with restriction to normalized $\pi(a|s)\in\Delta^{|\mathcal{A}|}$, where this constraint is not captured using $f$-divergence function space duality.

*Proof.* We begin by writing the $\alpha$-divergence $\Omega_{\pi_0}^{(\alpha)}(\mu) = \mathbb{E}_{\mu(s)}[D_\alpha[\pi_0:\pi]]$ as a function of the occupancy measure $\mu$, with $\pi(a|s) = \frac{\mu(a,s)}{\mu(s)}$. As in Prop. 2, the conjugate optimization implies an optimality condition for $\Delta r(a,s)$.

$$\frac{1}{\beta}\Omega_{\pi_0,\beta}^{*(\alpha)}(\Delta r) = \max_\mu\left\langle\mu,\Delta r\right\rangle-\frac{1}{\beta}\frac{1}{\alpha(1-\alpha)}\left((1-\alpha)\sum_{a,s}\mu(s)\pi_0(a|s)+\alpha\sum_{a,s}\mu(a,s)-\sum_{a,s}\mu(s)\pi_0(a|s)^{1-\alpha}\left(\frac{\mu(a,s)}{\mu(s)}\right)^\alpha\right) \quad (57)$$

$$\implies\Delta r = \nabla_\mu\frac{1}{\beta}\frac{1}{\alpha(1-\alpha)}\left((1-\alpha)\sum_{a,s}\mu(s)\pi_0(a|s)+\alpha\sum_{a,s}\mu(a,s)-\sum_{a,s}\mu(s)\pi_0(a|s)^{1-\alpha}\left(\frac{\mu(a,s)}{\mu(s)}\right)^\alpha\right) \quad (58)$$

**Worst-Case Reward Perturbations** $\Delta r_\pi(a|s)$  We now differentiate with respect to $\mu(a,s)$, using similar derivations as in Lee et al. (2019). While we have already written Eq. (58) using $\mu(a,s) = \mu(s)\pi(a|s)$, we again emphasize that $\mu(s)$ depends on $\mu(a,s)$. Differentiating, we obtain

$$\Delta r(a,s) = \nabla_\mu\frac{1}{\beta}\Omega_{\pi_0}^{(\alpha)}(\mu) \qquad (59)$$

$$= \frac{1}{\beta}\frac{1}{\alpha(1-\alpha)}\sum_{a',s'}\frac{\partial}{\partial\mu(a,s)}\left((1-\alpha)\mu(s')\pi_0(a'|s')+\alpha\mu(a',s')-\mu(s')^{1-\alpha}\pi_0(a'|s')^{1-\alpha}\mu(a',s')^\alpha\right)$$

$$= \frac{1}{\beta}\frac{1}{\alpha(1-\alpha)}\Bigg((1-\alpha)\sum_{s'}\underbrace{\frac{\partial\sum_{a'}\mu(a',s')}{\partial\mu(a,s)}}_{\delta(s=s')}\sum_{a'}\pi_0(a'|s)+\alpha\sum_{a',s'}\underbrace{\frac{\partial\mu(a',s')}{\partial\mu(a,s)}}_{\delta(a',s'=a,s)}-$$

$$-\sum_{a',s'}\alpha\mu(a',s')^{\alpha-1}\underbrace{\frac{\partial\mu(a',s')}{\partial\mu(a,s)}}_{\delta(a',s'=a,s)}\mu(s')^{1-\alpha}\pi_0(a'|s')^{1-\alpha}-(1-\alpha)\sum_{a',s'}\mu(s')^{-\alpha}\underbrace{\frac{\partial\sum_{a'}\mu(a',s')}{\partial\mu(a,s)}}_{\delta(s=s')}\pi_0(a'|s')^{1-\alpha}\mu(a',s')^\alpha\Bigg)$$

$$= \frac{1}{\beta}\frac{1}{\alpha(1-\alpha)}\left((1-\alpha)\sum_a\pi_0(a|s)+\alpha-\alpha\left(\frac{\mu(a,s)}{\mu(s)\pi_0(a|s)}\right)^{\alpha-1}-(1-\alpha)\sum_{a'}\pi_0(a|s)^{1-\alpha}\left(\frac{\mu(a,s)}{\mu(s)}\right)^\alpha\right)$$

$$\overset{(1)}{=}\frac{1}{\beta}\frac{1}{\alpha}+\frac{1}{\beta}\frac{1}{1-\alpha}-\frac{1}{\beta}\frac{1}{1-\alpha}\left(\frac{\pi(a|s)}{\pi_0(a|s)}\right)^{\alpha-1}-\frac{1}{\beta}\frac{1}{\alpha}\sum_a\pi_0(a|s)^{1-\alpha}\pi(a|s)^\alpha$$

$$= \frac{1}{\beta}\frac{1}{\alpha-1}\left(\left(\frac{\pi(a|s)}{\pi_0(a|s)}\right)^{\alpha-1}-1\right)+\frac{1}{\beta}\frac{1}{\alpha}\left(1-\sum_a\pi_0(a|s)^{1-\alpha}\pi(a|s)^\alpha\right) \qquad (60)$$

where we have rewritten (1) in terms of the policy $\pi(a|s) = \frac{\mu(a,s)}{\mu(s)}$ and assumed $\pi_0(a|s)$ is normalized.

Letting $\pi_{\Delta r}(a|s)$ indicate the policy which is in dual correspondence with $\Delta r(a,s)$, we would eventually like to invert the equality in Eq. (60) to solve for $\pi(a|s)$ in each $(a,s)$. However, the final term depends on a sum over all actions. To handle this, we define

$$\psi_{\Delta r}(s;\beta) = \frac{1}{\beta}\frac{1}{\alpha}\Big(\sum_a\pi_0(a|s)-\sum_a\pi_0(a|s)^{1-\alpha}\pi_{\Delta r}(a|s)^\alpha\Big). \qquad (61)$$

Since $\pi_{\Delta r}(a|s) = \frac{\mu_{\Delta r}(a,s)}{\mu(s)}$ is normalized by construction, the constant $\psi_{\Delta r}(s;\beta)$ with respect to actions has appeared naturally when optimizing with respect to $\mu(a,s)$. In App. C.2-C.3, we will relate this quantity to the Lagrange multiplier used to enforce normalization when optimizing over $\pi(a|s) \in \Delta^{|\mathcal{A}|}$.

Finally, we use Eq. (60) to write $\Delta r_\pi(a,s)$ as

$$\Delta r_\pi(a,s) = \frac{1}{\beta} \log_\alpha \frac{\pi(a|s)}{\pi_0(a|s)} + \psi_{\Delta r}(s;\beta) . \tag{62}$$

**Optimizing Argument $\pi_{\Delta r}(a|s)$** Solving for the policy in Eq. (62) and denoting this as $\pi_{\Delta r}(a|s)$,

$$\pi_{\Delta r}(a|s) = \pi_0(a|s) \exp_\alpha \left\{ \beta \cdot \left( \Delta r(a,s) - \psi_{\Delta r}(s;\beta) \right) \right\} = \pi_0(a|s) \left[ 1 + \beta(\alpha-1) \left( \Delta r(a,s) - \psi_{\Delta r}(s;\beta) \right) \right]_+^{\frac{1}{\alpha-1}}. \tag{63}$$

Note that $\Delta r_\pi(a|s)$ is defined in self-consistent fashion due to the dependence of $\psi_{\Delta r}(s;\beta)$ on $\pi_{\Delta r}(a|s)$ in Eq. (61). Further, $\psi_{\Delta r}(s;\beta)$ does not appear as a divisive normalization constant for general $\alpha$, which is inconvenient for practical applications (Lee et al., 2019; Chow et al., 2018).

**Conjugate Function $\frac{1}{\beta} \Omega_{\pi_0,\beta}^{*(\alpha)}(\Delta r)$** Finally, we plug this into the conjugate optimization Eq. (57). Although we eventually need to obtain a function of $\Delta r(a,s)$ only, we write $\pi_{\Delta r}(a|s)$ in initial steps to simplify notation.

$$\frac{1}{\beta} \Omega_{\pi_0,\beta}^{*(\alpha)}(\Delta r) = \sum_{a,s} \mu(s)\pi_{\Delta r}(a|s) \cdot \Delta r(a,s) - \frac{1}{\beta} \frac{1}{\alpha(1-\alpha)} \left( (1-\alpha) \sum_{a,s} \mu(s)\pi_0(a|s) + \alpha \sum_{a,s} \mu(s)\pi_{\Delta r}(a|s) - \sum_{a,s} \mu(s)\pi_{\Delta r}(a|s) \left( \frac{\pi_{\Delta r}(a|s)}{\pi_0(a|s)} \right)^{\alpha-1} \right)$$

$$= \sum_{a,s} \mu(s)\pi_{\Delta r}(a|s) \cdot \Delta r(a,s) - \frac{1}{\beta} \frac{1}{\alpha} \sum_{a,s} \mu(s)\pi_0(a|s) - \frac{1}{\beta} \frac{1}{1-\alpha} \sum_{a,s} \mu(s)\pi_{\Delta r}(a|s) \tag{64}$$

$$+ \frac{1}{\beta} \frac{1}{\alpha(1-\alpha)} \sum_{a,s} \mu(s)\pi_{\Delta r}(a|s) \left( 1 + \underbrace{\beta(\alpha-1)}_{-1} \left( \Delta r(a,s) - \psi_{\Delta r}(s;\beta) \right) \right)$$

$$\overset{(1)}{=} \left( \underbrace{1 - \frac{1}{\alpha}}_{\frac{\alpha-1}{\alpha}} \right) \sum_{a,s} \mu(s)\pi_{\Delta r}(a|s) \cdot \Delta r(a,s) - \frac{1}{\beta} \frac{1}{\alpha} \sum_{a,s} \mu(s)\pi_0(a|s) - \underbrace{\left( \frac{1}{\beta} \frac{1}{1-\alpha} - \frac{1}{\beta} \frac{1}{\alpha(1-\alpha)} \right)}_{-\frac{1}{\beta} \frac{1}{\alpha}} \sum_{a,s} \mu(s)\pi_{\Delta r}(a|s) + \frac{1}{\alpha} \psi_{\Delta r}(s;\beta)$$

$$\overset{(2)}{=} \frac{1}{\beta} \frac{1}{\alpha} \sum_{a,s} \mu(s)\pi_{\Delta r}(a|s) + \frac{\beta}{\beta} \frac{\alpha-1}{\alpha} \sum_{a,s} \mu(s)\pi_{\Delta r}(a|s) \cdot \Delta r(a,s) \pm \frac{\beta}{\beta} \frac{\alpha-1}{\alpha} \psi_{\Delta r}(s;\beta) + \frac{1}{\alpha} \psi_{\Delta r}(s;\beta) - \frac{1}{\beta} \frac{1}{\alpha} \sum_{a,s} \mu(s)\pi_0(a|s)$$

where in (1) we note that $\frac{1}{\beta} \frac{1}{\alpha(1-\alpha)} \cdot \beta(\alpha-1) = -\frac{1}{\alpha}$. In (2), we add and subtract the term in blue, which will allow to factorize an additional term of $[1 + \beta(\alpha-1)(\Delta r - \psi_{\Delta r}(s;\beta))]$ and obtain a function of $\Delta r(a,s)$ only

$$\frac{1}{\beta} \Omega_{\pi_0,\beta}^{*(\alpha)}(\Delta r) = \frac{1}{\beta} \frac{1}{\alpha} \sum_{a,s} \mu(s)\pi_{\Delta r}(a|s) \left( 1 + \beta(\alpha-1) \left( \Delta r(a,s) - \psi_{\Delta r}(s;\beta) \right) \right) + \left( \underbrace{\frac{\alpha-1}{\alpha} + \frac{1}{\alpha}}_{1} \right) \psi_{\Delta r}(s;\beta) - \frac{1}{\beta} \frac{1}{\alpha} \sum_{a,s} \mu(s)\pi_0(a|s)$$

$$\overset{(1)}{=} \frac{1}{\beta} \frac{1}{\alpha} \sum_{a,s} \mu(s) \left( \pi_0(a|s) \left[ 1 + \beta(\alpha-1) \left( \Delta r(a,s) - \psi_{\Delta r}(s;\beta) \right) \right]^{\frac{\alpha}{\alpha-1}} - 1 \right) + \sum_s \mu(s) \psi_{\Delta r}(s;\beta) \rangle \tag{65}$$

where in (1) we have used Eq. (63) and $\frac{1}{\alpha-1} + 1 = \frac{\alpha}{\alpha-1}$, along with $\sum_a \pi_0(a|s) = 1$. $\qquad \square$

**Confirming Conjugate Optimality Conditions** Finally, we confirm that differentiating Eq. (65) with respect to $\Delta r(a,s)$ yields the conjugate condition $\pi_{\Delta r} = \nabla \frac{1}{\beta} \Omega_{\pi_0,\beta}^{*(\alpha)}(\Delta r)$. Noting that $\frac{\alpha}{\alpha-1} - 1 = \frac{1}{\alpha-1}$,

$$\nabla \frac{1}{\beta} \Omega_{\pi_0,\beta}^{*(\alpha)}(\Delta r) = \frac{\beta(\alpha-1)}{\beta\alpha} \frac{\alpha}{\alpha-1} \sum_s \mu(s) \sum_a \pi_0(a|s) \left[ 1 + \beta(\alpha-1) \left( \Delta r(a,s) - \psi_{\Delta r}(s;\beta) \right) \right]^{\frac{1}{\alpha-1}} \left( \frac{\partial \Delta r(a,s)}{\partial \Delta r(a',s')} - \frac{\partial \psi_{\Delta r}(s;\beta)}{\partial \Delta r(a',s')} \right) + \sum_s \mu(s) \frac{\partial \psi_{\Delta r}(s;\beta)}{\partial \Delta r(a',s')}$$

which simplifies to $\pi_{\Delta r}(a|s) = \frac{\partial}{\partial \Delta r(a,s)} \frac{1}{\beta} \Omega_{\pi_0,\beta}^{*(\alpha)}(\Delta r) = \pi_0(a|s) \left[ 1 + \beta(\alpha-1) \left( \Delta r(a,s) - \psi_{\Delta r}(s;\beta) \right) \right]^{\frac{1}{\alpha-1}}$ and matches Eq. (63).

## B.4 $\alpha$-Divergence Occupancy Regularization: $\frac{1}{\beta}\Omega_{\mu_0,\beta}^{*(\alpha)}(\Delta r)$

The conjugate function $\frac{1}{\beta}\Omega_{\mu_0,\beta}^{*(\alpha)}(\Delta r)$ for $\alpha$-divergence regularization of the state-action occupancy $\mu(a,s)$ to a reference $\mu_0(a,s)$ can be written in the following form

$$\frac{1}{\beta}\Omega_{\mu_0,\beta}^{*(\alpha)}(\Delta r) = \frac{1}{\beta}\frac{1}{\alpha}\sum_{a,s}\mu_0(a,s)\big[1 + \beta(\alpha-1)\Delta r(a,s)\big]^{\frac{\alpha}{\alpha-1}} - \frac{1}{\beta}\frac{1}{\alpha} \tag{66}$$

Note that this conjugate function can also be derived directly from the duality of general $f$-divergences, and matches the form of conjugate considered in (Belousov & Peters, 2019; Nachum & Dai, 2020).

*Proof.* **Worst-Case Reward Perturbations** $\Delta r_\mu(a|s)$

$$\Delta r(a,s) = \frac{1}{\beta}\frac{1}{\alpha(1-\alpha)}\nabla_\mu\left((1-\alpha)\sum_{a,s}\mu_0(a,s) + \alpha\sum_{a,s}\mu(a,s) - \sum_{a,s}\mu_0(a,s)^{1-\alpha}\mu(a,s)^\alpha\right) \tag{67}$$

$$= \frac{1}{\beta}\frac{1}{1-\alpha} - \frac{1}{\beta}\frac{1}{1-\alpha}\mu_0(a,s)^{1-\alpha}\mu(a,s)^{\alpha-1}$$

**Optimizing Argument** $\mu_{\Delta r}(a,s)$. Solving for $\mu_{\Delta r}(a,s)$,

$$\mu_{\Delta r}(a,s) = \mu_0(a,s)\exp_\alpha\{\beta\cdot\Delta r(a,s)\} = \mu_0(a,s)[1 + \beta(1-\alpha)\Delta r(a,s)]_+^{\frac{1}{\alpha-1}} \tag{68}$$

**Conjugate Function** $\frac{1}{\beta}\Omega_{\mu_0,\beta}^{*(\alpha)}(\Delta r)$. Plugging this back into the conjugate optimization, we finally obtain

$$\frac{1}{\beta}\Omega_{\mu_0,\beta}^{*(\alpha)} = \sum_{a,s}\mu_{\Delta r}(a,s)\cdot\Delta r(a,s) - \frac{1}{\beta}\frac{1}{\alpha(1-\alpha)}\left((1-\alpha)\sum_{a,s}\mu_0(a,s) + \alpha\sum_{a,s}\mu_{\Delta r}(a,s) - \sum_{a,s}\mu_{\Delta r}(a,s)\underbrace{\left(\frac{\mu_{\Delta r}(a,s)}{\mu_0(a,s)}\right)^{\alpha-1}}_{=1+\beta(\alpha-1)\Delta r(a,s)\,(Eq.\,(68))}\right)$$

$$= \left(1-\frac{1}{\alpha}\right)\sum_{a,s}\mu_{\Delta r}(a,s)\cdot\Delta r(a,s) - \frac{1}{\beta}\frac{1}{\alpha}\sum_{a,s}\mu_0(a,s) + \left(\frac{1}{\beta}\frac{1}{\alpha(1-\alpha)} - \frac{1}{\beta}\frac{1}{1-\alpha}\right)\sum_{a,s}\mu_{\Delta r}(a,s) \tag{69}$$

$$= \frac{\alpha-1}{\alpha}\sum_{a,s}\mu_0(a,s)\big[1 + \beta(\alpha-1)\Delta r(a,s)\big]^{\frac{1}{\alpha-1}}\cdot\Delta r(a,s) + \frac{1}{\beta}\frac{1}{\alpha}\sum_{a,s}\mu_0(a,s)\big[1 + \beta(\alpha-1)\Delta r(a,s)\big]^{\frac{1}{\alpha-1}} - \frac{1}{\beta}\frac{1}{\alpha}\sum_{a,s}\mu_0(a,s)$$

$$= \sum_{a,s}\mu_0(a,s)\big[1 + \beta(\alpha-1)\Delta r(a,s)\big]^{\frac{1}{\alpha-1}}\cdot\frac{1}{\beta}\frac{1}{\alpha}\big(1 + \beta(\alpha-1)\Delta r(a,s)\big) - \frac{1}{\beta}\frac{1}{\alpha}\sum_{a,s}\mu_0(a,s) \tag{70}$$

$$= \frac{1}{\beta}\frac{1}{\alpha}\sum_{a,s}\mu_0(a,s)\big[1 + \beta(\alpha-1)\Delta r(a,s)\big]^{\frac{\alpha}{\alpha-1}} - \frac{1}{\beta}\frac{1}{\alpha}\sum_{a,s}\mu_0(a,s) \tag{71}$$

where, to obtain the exponent in the last line, note that $\frac{1}{\alpha-1} + 1 = \frac{\alpha}{\alpha-1}$. $\qquad\square$

## C Soft Value Aggregation

Soft value aggregation (Fox et al., 2016; Haarnoja et al., 2017) and the regularized Bellman optimality operator (Neu et al., 2017; Geist et al., 2019) also rely on the convex conjugate function, but with a slightly different setting than our derivations for the optimal regularized policy or reward perturbations in App. B. In particular, in each state we optimize over the policy $\pi(a|s) \in \Delta^{|\mathcal{A}|}$ using an explicit normalization constraint (Eq. (74)).

We derive the regularized Bellman optimality operator from the primal objective in Eq. (12). Factorizing $\mu(a,s) = \mu(s)\pi(a|s)$, we can imagine optimizing over $\mu(s)$ and $\pi(a|s) \in \Delta^{|\mathcal{A}|}$ separately,

$$\max_{\mu(s)\to\mathcal{M}}\max_{\pi(a|s)\in\Delta^{|\mathcal{A}|}}\min_{V(s)}(1-\gamma)\langle\nu_0(s),V(s)\rangle + \langle\mu(a,s),r(a,s) + \gamma\mathbb{E}_{a,s}^{s'}\big[V(s')\big] - V(s)\rangle - \frac{1}{\beta}\Omega_{\pi_0}(\mu). \tag{72}$$

Eliminating $\mu(s)$ (by setting $d/d\mu(s) = 0$) leads to a constraint on the form of $V(s)$, since both may be viewed as enforcing the Bellman flow constraints.

$$V(s) = \left\langle \pi(a|s), \, r(s,a) + \gamma \mathbb{E}_{a,s}^{s'}\big[V(s')\big] \right\rangle - \frac{1}{\beta}\Omega_{\pi_0}(\pi). \tag{73}$$

We define $Q(s,a) \coloneqq r(s,a) + \gamma\mathbb{E}_{a,s}^{s'}\big[V(s')\big]$ and write $V(s) = \langle \pi, \, Q\rangle - \frac{1}{\beta}\Omega_{\pi_0}(\pi)$ moving forward.

As an operator for iteratively updating $V(s)$, Eq. (73) corresponds to the regularized Bellman operator $\mathcal{T}_{\Omega_{\pi_0},\beta}$ and may be used to perform policy evaluation for a given $\pi(a|s)$ (Geist et al., 2019). The regularized Bellman *optimality* operator $\mathcal{T}_{\Omega_{\pi_0},\beta}^*$, which can be used for value iteration or modified policy iteration (Geist et al., 2019), arises from including the maximization over $\pi(a|s) \in \Delta^{|\mathcal{A}|}$ from Eq. (72)

$$V(s) \leftarrow \frac{1}{\beta}\Omega_{\pi_0,\beta}^*(Q) = \max_{\pi \in \Delta^{|\mathcal{A}|}} \langle \pi, Q\rangle - \frac{1}{\beta}\Omega_{\pi_0}(\pi)\,. \tag{74}$$

**Comparison of Conjugate Optimizations**  Eq. (74) has the form of a conjugate optimization $\frac{1}{\beta}\Omega_{\pi_0,\beta}^*(Q)$ (Geist et al., 2019). However, in contrast to the setting of App. A.2 and App. B, we optimize over the policy in each state, rather than the state-action occupancy $\mu(a,s)$. Further, we must include normalization and nonnegativity constraints $\pi(a|s) \in \Delta^{|\mathcal{A}|}$, which can be enforced using Lagrange multipliers $\psi_Q(s;\beta)$ and $\lambda(a,s)$. We derive expressions for this conjugate function for the KL divergence in App. C.1 and $\alpha$-divergence in App. C.2, and plot the value $V_*(s)$ as a function of $\alpha$ and $\beta$ in App. C.4.

Compared with the optimization for the optimal policy in Eq. (35), note that the argument of the conjugate function does not include the value function $V(s)$ in this case. We will highlight relationship between the normalization constants $\psi_{Q_*}(s;\beta)$, $\psi_{\Delta r_{\pi_*}}(s;\beta)$, and $V_*(s)$ in App. C.3, where $\psi_{Q_*}(s;\beta) = V_*(s) + \psi_{\Delta r_{\pi_*}}(s;\beta)$ as in Lee et al. (2019) App. D.

## C.1  KL Divergence Soft Value Aggregation: $\frac{1}{\beta}\Omega_{\pi_0,\beta}^*(Q)$

We proceed to derive a closed form for the conjugate function of the KL divergence $\Omega_{\pi_0}(\pi)$ as a function of $\pi(a|s) \in \Delta^{|\mathcal{A}|}$, which we write using the $Q$-values as input

$$\frac{1}{\beta}\Omega_{\pi_0,\beta}^*(Q) = \max_{\pi(a|s)} \langle \pi, Q\rangle - \frac{1}{\beta}\left(\sum_a \pi(a|s)\log\frac{\pi(a|s)}{\pi_0(a|s)} + \sum_a \pi(a|s) - \sum_a \pi_0(a|s)\right) - \psi_Q(s;\beta)\left(\sum_{a\in\mathcal{A}}\pi(a|s) - 1\right) + \sum_{a\in\mathcal{A}}\lambda(a,s)$$

$$\implies Q(a,s) = \frac{1}{\beta}\log\frac{\pi(a|s)}{\pi_0(a|s)} + \psi_Q(s;\beta) - \lambda(a,s) \tag{75}$$

**Optimizing Argument**  Solving for $\pi$ yields the optimizing argument

$$\pi_Q(a|s) = \pi_0(a|s)\exp\left\{\beta\cdot\big(Q(a,s) - \psi_Q(s;\beta))\big)\right\} \tag{76}$$

where we can ignore the Lagrange multiplier for the nonnegativity constraint $\lambda(a,s)$ since $\exp\{\cdot\} \geq 0$ ensures $\pi_Q(a|s) \geq 0$. We can pull the normalization constant out of the exponent to solve for

$$\psi_Q(s;\beta) = \frac{1}{\beta}\log\sum_a \pi_0(a|s)\exp\left\{\beta\cdot Q(a,s)\right\}. \tag{77}$$

Plugging Eq. (76) into the conjugate optimization,

$$\frac{1}{\beta}\Omega_{\pi_0,\beta}^*(Q) = \left\langle\pi_Q,Q\right\rangle - \frac{1}{\beta}\left(\sum_a \pi_Q(a|s)\cdot\log\frac{\pi_0(a|s)}{\pi_0(a|s)}\exp\left\{\beta\cdot\big(Q(a,s) - \psi_Q(s;\beta)\big)\right\} + \underbrace{\sum_a \pi_Q(a|s)}_{1} - 1\right) - \psi_Q(s;\beta)\left(\underbrace{\sum_{a\in\mathcal{A}}\pi_Q(a|s)}_{1} - 1\right).$$

**Conjugate Function** We finally recover the familiar log-mean-exp form for the KL-regularized value function

$$V(s) \leftarrow \frac{1}{\beta}\Omega^*_{\pi_0,\beta}(Q) = \psi_Q(s;\beta) = \frac{1}{\beta}\log\sum_a \pi_0(a|s)\exp\{\beta \cdot Q(a,s)\}. \tag{78}$$

Notice that the conjugate or value function $V(s) \leftarrow \frac{1}{\beta}\Omega^*_{\pi_0,\beta}(Q)$ is exactly equal to the normalization constant of the policy $\psi_Q(s;\beta)$. We will show in App. C.2 that this property *does not hold* for general $\alpha$-divergences, with example visualizations in App. C.3 Fig. 5.

## C.2 $\alpha$-Divergence Soft Value Aggregation: $\frac{1}{\beta}\Omega^*_{\pi_0,\beta}(Q)$

We now consider soft value aggregation using the $\alpha$-divergence, where in contrast to App. B.3, we perform the conjugate optimization over $\pi(a|s) \in \Delta^{|\mathcal{A}|}$ in each state, with Lagrange multipliers $\psi_Q(s;\beta)$ and $\lambda(a,s)$ to enforce normalization and nonnegativity.

$$\frac{1}{\beta}\Omega^*_{\pi_0,\beta}(Q) = \max_{\pi(a|s)}\langle\pi,Q\rangle - \frac{1}{\beta}\frac{1}{\alpha}\sum_a \pi_0(a|s) - \frac{1}{\beta}\frac{1}{1-\alpha}\sum_a \pi(a|s) + \frac{1}{\beta}\frac{1}{\alpha(1-\alpha)}\sum_a \pi_0(a|s)^{1-\alpha}\pi(a|s)^\alpha \tag{79}$$

$$- \psi_Q(s;\beta)\left(\sum_a \pi(a|s) - 1\right) + \sum_a \lambda(a,s)$$

$$\implies Q(a,s) = \frac{1}{\beta}\log_\alpha\frac{\pi(a|s)}{\pi_0(a|s)} + \psi_Q(s;\beta) - \lambda(a,s) \tag{80}$$

**Optimizing Argument** Solving for $\pi$ yields the optimizing argument for the soft value aggregation conjugate,

$$\pi_Q(a|s) = \pi_0(a|s)\exp_\alpha\left\{\beta \cdot \big(Q(a,s) + \lambda(a,s) - \psi_Q(s;\beta)\big)\right\}. \tag{81}$$

Unlike the case of the standard exponential, we cannot easily derive a closed-form solution for $\psi_Q(s;\beta)$.

Note that the expressions in Eq. (80) and Eq. (81) are similar to the form of the worst-case reward perturbations $\Delta r_\pi(a|s)$ in Eq. (62) and optimizing policy $\pi_{\Delta r}(a|s)$ in Eq. (63), except for the fact that $\psi_Q(s;\beta)$ arises as a Lagrange multiplier and does not have the same form as $\psi_{\Delta r}(s;\beta) = \frac{1}{\beta}(1-\alpha)D_\alpha[\pi_0:\pi]$ as in Eq. (23) and Eq. (61). We will find that $\psi_Q(s;\beta)$ and $\psi_{\Delta r}(s;\beta)$ differ by a term of $V_*(s)$ in App. C.3 (Eq. (86)).

**Conjugate Function** Plugging Eq. (81) into the conjugate optimization, we use similar derivations as in Eq. (64)-Eq. (65) to write the conjugate function, or regularized Bellman optimality operator as

$$V(s) \leftarrow \frac{1}{\beta}\Omega^{*(\alpha)}_{\pi_0,\beta}(Q) = \frac{1}{\beta}\frac{1}{\alpha}\sum_a \pi_0(a|s)\Big[1 + \beta(\alpha-1)\big(Q(a,s) + \lambda(a,s) - \psi_Q(s;\beta)\big)\Big]_+^{\frac{\alpha}{\alpha-1}} - \frac{1}{\beta}\frac{1}{\alpha} + \psi_Q(s;\beta)$$

$$\tag{82}$$

$$= \frac{1}{\beta}\frac{1}{\alpha}\sum_a \pi_0(a|s)\exp_\alpha\left\{\beta \cdot \big(Q(a,s) + \lambda(a,s) - \psi_Q(s;\beta)\big)\right\}^\alpha - \frac{1}{\beta}\frac{1}{\alpha} + \psi_Q(s;\beta)$$

**Comparison with KL Divergence Regularization** Note that for general $\alpha$, the conjugate or value function $V(s) = \frac{1}{\beta}\Omega^*_{\pi_0,\beta}(Q)$ in Eq. (82) is *not* equal to the normalization constant of the policy $\psi_Q(s;\beta)$. We discuss this further in the next section.

We also note that the form of the conjugate function is similar using two different approaches: optimizing over $\pi$ with an explicit normalization constraint, as in Eq. (82), or optimizing over $\mu$ with regularization of $\pi$ but no explicit normalization constraint, as in App. B.3 or Table 2. This is in contrast to the KL divergence, where the normalization constraint led to a log-mean-exp conjugate in Eq. (75) which is different from App. B Eq. (47).

### C.3 Relationship between Normalization Constants $\psi_{\Delta r_{\pi_*}}$, $\psi_{Q_*}$, and Value Function $V_*(s)$

In this section, we analyze the relationship between the conjugate optimizations that we have considered above, either optimizing over $\mu(a,s)$ as in deriving the optimal policy, or optimizing over $\pi(a|s) \in \Delta^{|\mathcal{A}|}$ as in the regularized Bellman optimality operator or soft-value aggregation. Using $Q(a,s) = r(a,s) + \gamma \mathbb{E}_{a,s}^{s'}[V(s')]$,

*Optimal Policy (or Worst-Case Reward Perturbations)* (App. B.3) $\hspace{2cm}$ (83)

$$\frac{1}{\beta}\Omega_{\pi_0,\beta}^{*(\alpha)}\Big(r + \gamma\mathbb{E}_{a,s}^{s'}[V] - V + \lambda\Big) = \max_{\mu(a,s)\in\mathcal{F}}\Big\langle\mu, r + \gamma\mathbb{E}_{a,s}^{s'}[V] - V + \lambda\Big\rangle - \frac{1}{\beta}\Omega_{\pi_0}^{(\alpha)}(\mu)$$

*Soft Value Aggregation* (App. C.2), $\hspace{5cm}$ (84)

$$V(s) \leftarrow \frac{1}{\beta}\Omega_{\pi_0,\beta}^{*(\alpha)}\Big(r + \gamma\mathbb{E}_{a,s}^{s'}[V]\Big) = \max_{\pi\in\Delta^{|\mathcal{A}|}}\Big\langle\mu, r + \gamma\mathbb{E}_{a,s}^{s'}[V]\Big\rangle - \frac{1}{\beta}\Omega_{\pi_0}^{(\alpha)}(\pi)$$

Note that the arguments differ by a term of $V(s)$. We ignore the apparent difference in the $\lambda(a,s)$ term, which can be considered as an argument of the conjugate in Eq. (84) since a linear term of $\langle\mu,\lambda\rangle$ appears when enforcing $\pi \in \Delta^{|\mathcal{A}|}$. Evaluating the optimizing arguments,

*Optimal Policy (or Worst-Case Reward Perturbations)* (App. B.3, Eq. (63), Table 2)

$$\pi(a|s) = \pi_0(a|s)\exp_\alpha\Big\{\beta\cdot\big(Q(a,s) + \lambda(a,s) - V(s) - \psi_{\Delta r}(s;\beta)\big)\Big\} \hspace{1cm} (85)$$

*Soft Value Aggregation (App. C.2, Eq. (82))*,

$$\pi(a|s) = \pi_0(a|s)\exp_\alpha\Big\{\beta\cdot\big(Q(a,s) + \lambda(a,s) - \psi_Q(s;\beta)\big)$$

For the optimal $V_*(s)$ and $Q_*(a,s) = r(a,s) + \gamma\mathbb{E}_{a,s}^{s'}[V_*(s')] - V_*(s)$, the two policies match. This can be confirmed using similar reasoning as in Lee et al. (2019) App. D-E or (Geist et al., 2019) to show that iterating the regularized Bellman optimality operator leads to the optimal policy and value.

**Relationship between $V_*(s)$ and $\psi_{Q_*}(s;\beta)$** This implies the condition which is the main result of this section.

$$\psi_{Q_*}(s;\beta) = V_*(s) + \psi_{\Delta r_{\pi_*}}(s;\beta). \hspace{2cm} (86)$$

In Fig. 5 we empirically confirm this identity and inspect how each quantity varies with $\beta$ and $\alpha$ (App. C.4)[4]

Eq. (86) highlights distinct roles for the value function $V_*(s)$ and the Lagrange multiplier $\psi_{Q_*}(s;\beta)$ enforcing normalization of $\pi(a|s)$ in soft-value aggregation ( Eq. (79) or Eq. (84)). It is well known that these coincide in the case of KL divergence regularization, with $V_*(s) = \psi_{Q_*}(s;\beta)$ as in App. C.1. We can also confirm that $\psi_{\Delta r_{\pi_*}}(s;\beta) = \frac{1}{\beta}\frac{1}{\alpha}\big(\sum_a\pi_0(a|s) - \sum_a\pi_0(a|s)^{1-\alpha}\pi_*(a|s)^\alpha\big) = 0$ vanishes for KL regularization ($\alpha = 1$) and normalized $\pi_0$ and the normalized optimal policy $\pi_*$.

However, in the case of $\alpha$-divergence regularization, optimization over the joint $\mu(a,s)$ in Eq. (83) introduces the additional term $\psi_{\Delta r_{\pi_*}}(s;\beta)$, which is not equal to 0 in general.

**Relationship between Conjugate Functions** We might also like to compare the value of the conjugate functions in Eq. (83) and Eq. (84), in particular to understand how including $V_*(s)$ as an argument and optimizing over $\pi$ versus $\mu$ affect the optima. We write the expressions for the conjugate function in each

---

[4]Note that $\psi_{\Delta r_{\pi_*}}(s;\beta) = \frac{1}{\beta}\frac{1}{\alpha}\big(\sum_a\pi_0(a|s) - \sum_a\pi_0(a|s)^{1-\alpha}\pi_*(a|s)^\alpha\big)$ appears from differentiating $\frac{1}{\beta}\Omega_{\pi_0}^{(\alpha)}(\mu)$ with respect to $\mu$ (App. B.3 Eq. (61)). We also write this as $\psi_{\Delta r_{\pi_*}}(s;\beta) = \frac{1}{\beta}(1-\alpha)D_\alpha[\pi_0:\pi_*]$ for normalized $\pi_0$ and $\pi_*$.

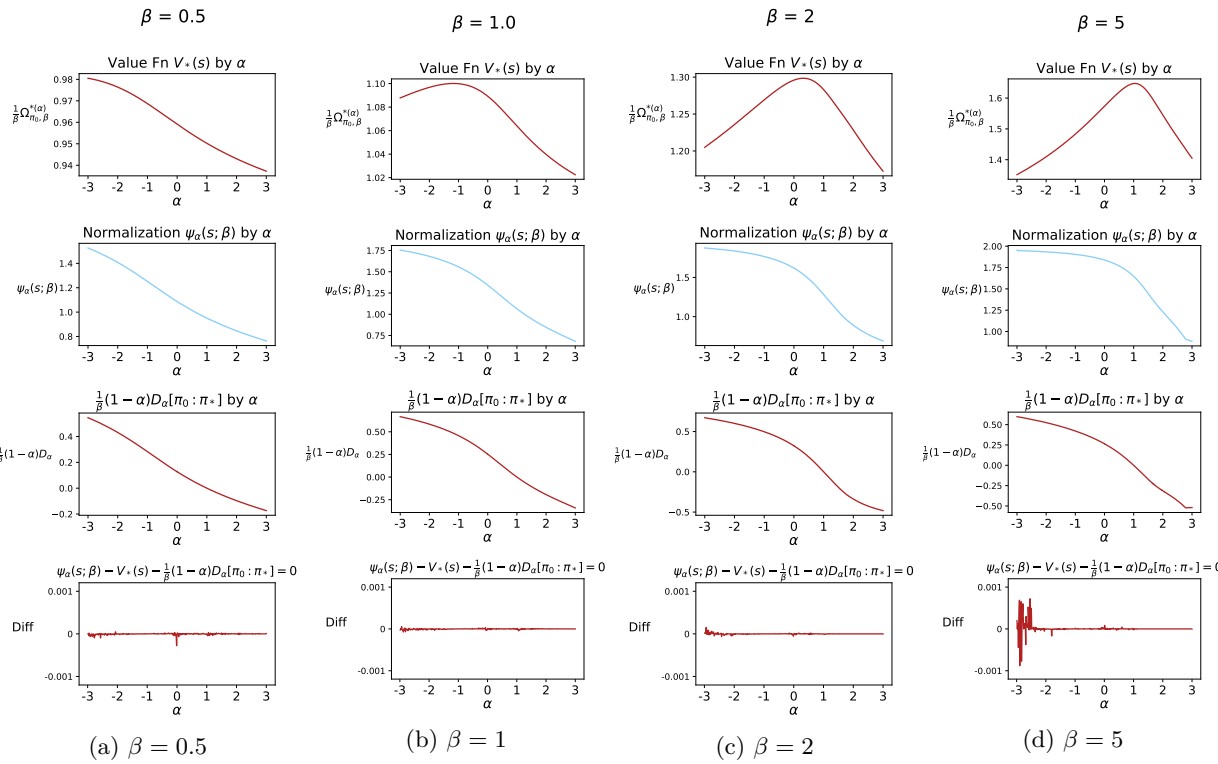

Figure 5: Value Function $V_*(s) = \frac{1}{\beta}\Omega^{*(\alpha)}_{\pi_0,\beta}(Q_*)$ (first row) and Normalization Constant $\psi_{Q_*}(s;\beta)$ (second row) as a function of $\alpha$ for various regularization strengths $\beta$. We use the same rewards as in Fig. 3 and Fig. 7 and a uniform reference. We plot $\psi_{\Delta r}(s;\beta) = \frac{1}{\beta}(1-\alpha)D_\alpha[\pi_0 : \pi_*]$ in the third row, and confirm the identity $V_*(s) = \psi_{Q_*}(s;\beta) - \psi_{\Delta r}(s;\beta)$ from Eq. (86) and (90) in the last row. We find that this equality holds for all $\alpha$ up to small optimization errors on the order of $10^{-3}$.

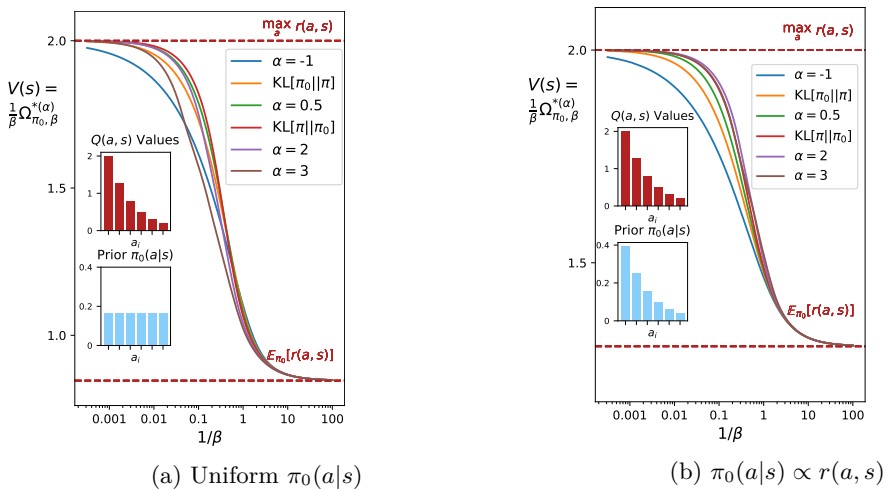

Figure 7: Value function $V(s) = \frac{1}{\beta}\Omega^{*(\alpha)}_{\mu_0,\beta}(Q)$ as a function of $\beta$ (x-axis) and $\alpha$ (colored lines), using $Q(a,s)$ and $\pi_0(a|s)$ from the left inset. See Eq. (78) and Eq. (82) for closed forms.

case, highlighting the terms from Eq. (86) in blue.

*Optimal Policy  (or Worst-Case Reward Perturbations)* (App. B.3, Eq. (56), Table 2)

$$\frac{1}{\beta}\Omega^{*(\alpha)}_{\pi_0,\beta}\Big(r + \gamma\mathbb{E}^{s'}_{a,s}\big[V_*\big] - V_* + \lambda_*\Big) \tag{87}$$
$$= \frac{1}{\beta}\frac{1}{\alpha}\sum_a \pi_0(a|s)\exp_\alpha\Big\{\beta\cdot\big(Q_*(a,s) + \lambda_*(a,s) - V_*(s) - \psi_{\Delta r}(s;\beta)\big)\Big\}^\alpha - \frac{1}{\beta}\frac{1}{\alpha} + \psi_{\Delta r_{\pi_*}}(s;\beta)$$

*Soft Value Aggregation* (App. C.2, Eq. (82)),

$$V_*(s) \leftarrow \frac{1}{\beta}\Omega^*_{\pi_0,\beta}\Big(r + \gamma\mathbb{E}^{s'}_{a,s}\big[V_*\big]\Big) \tag{88}$$
$$= \frac{1}{\beta}\frac{1}{\alpha}\sum_a \pi_0(a|s)\exp_\alpha\Big\{\beta\cdot\big(Q_*(a,s) + \lambda_*(a,s) - \psi_Q(s;\beta)\big)\Big\}^\alpha - \frac{1}{\beta}\frac{1}{\alpha} + \psi_Q(s;\beta)$$

Note that we have rewritten $V_*(s) \leftarrow \frac{1}{\beta}\Omega^*_{\pi_0,\beta}(r + \gamma\mathbb{E}^{s'}_{a,s}\big[V_*\big])$ directly as $V_*(s)$. To further simplify, note that the optimal policy matches as in Eq. (85), with

$$\pi_*(a|s) = \pi_0(a|s)\exp_\alpha\{\beta\cdot(Q_*(a,s) + \lambda_*(a,s) - \psi_{Q_*}(s;\beta))\}$$
$$= \pi_0(a|s)\exp_\alpha\{\beta\cdot(Q_*(a,s) + \lambda_*(a,s) - V_*(s) - \psi_{\Delta r_{\pi_*}}(s;\beta))\}.$$

Since $\pi_*(a|s) = \pi_0(a|s)\exp_\alpha\{\cdot\}$, we can write terms of the form $\pi_0(a|s)\exp_\alpha\{\cdot\}^\alpha$ in Eq. (87)-(88) as $\pi_0(a|s)^{1-\alpha}\pi_*(a|s)^\alpha$, where the exponents of $\pi_0(a|s)$ add to 1. Finally, we use this expression to simplify the value function expression in Eq. (88), eventually recovering the equality in Eq. (86)

$$\frac{1}{\beta}\Omega^{*(\alpha)}_{\pi_0,\beta}\Big(r + \gamma\mathbb{E}^{s'}_{a,s}\big[V_*\big]\Big) = V_*(s) = \frac{1}{\beta}\frac{1}{\alpha}\sum_a \pi_0(a|s)^{1-\alpha}\pi_*(a|s)^\alpha - \frac{1}{\beta}\frac{1}{\alpha} + \psi_Q(s;\beta) \tag{89}$$

$$= \psi_Q(s;\beta) - \psi_{\Delta r_{\pi_*}}(s;\beta) \tag{90}$$

In the second line, we use the fact that $\psi_{\Delta r_{\pi_*}}(s;\beta) = \frac{1}{\beta}\frac{1}{\alpha}\sum_a \pi_0(a|s) - \frac{1}{\beta}\frac{1}{\alpha}\sum_a \pi_0(a|s)^{1-\alpha}\pi_*(a|s)^\alpha$ from Eq. (61). We can use the same identity to show that the conjugate in Eq. (87) evaluates to zero,

$$\frac{1}{\beta}\Omega^{*(\alpha)}_{\pi_0,\beta}\Big(r + \gamma\mathbb{E}^{s'}_{a,s}\big[V_*\big] - V_* + \lambda_*\Big) = \underbrace{\frac{1}{\beta}\frac{1}{\alpha}\sum_a \pi_0(a|s)^{1-\alpha}\pi_*(a|s)^\alpha - \frac{1}{\beta}\frac{1}{\alpha}}_{\psi_{\Delta r_{\pi_*}}(s;\beta)} + \psi_{\Delta r_{\pi_*}}(s;\beta) = 0 \tag{91}$$

In Lemma 2, we provide a more detailed proof and show that this identity also holds for suboptimal policies and their worst-case reward perturbations, $\frac{1}{\beta}\Omega^{*(\alpha)}_{\pi_0,\beta}(\Delta r_\pi) = 0$, where Eq. (91) is a special case for $\Delta r_{\pi_*}(a,s) = r(a,s) + \gamma\mathbb{E}^{s'}_{a,s}\big[V_*(s')\big] - V_*(s) + \lambda_*(a,s)$.

Finally, note that the condition in Eq. (91) implies that for the optimal $V_*(s)$, the regularized dual objective $\mathcal{RL}^*_{\Omega,\beta}(r) = \min_{V,\lambda}(1-\gamma)\langle\nu_0, V\rangle + \frac{1}{\beta}\Omega^*_{\pi_0,\beta}\Big(r + \gamma\mathbb{E}^{s'}_{a,s}V - V + \lambda\Big)$ in Eq. (14) reduces to the value function averaged over initial states, $\mathcal{RL}^*_{\Omega,\beta}(r) = (1-\gamma)\langle\nu_0, V_*\rangle$. This is intuitive since $V_*(s)$ measures the regularized objective attained from running the optimal policy for infinite time in the discounted MDP.

## C.4  Plotting Value Function as a Function of Regularization Parameters $\alpha, \beta$

**Confirming Relationship between Normalization $\psi_{\Delta r_{\pi_*}}(s;\beta)$, $\psi_{Q_*}(s;\beta)$ and Value Function $V_*(s)$**
In Fig. 5, we plot both $V_*(s)$ and $\psi_{Q_*}(s;\beta)$ for various values of $\alpha$ (x-axis) and $\beta$ (in each panel). We also plot $\psi_{\Delta r_{\pi_*}}(s;\beta) = \frac{1}{\beta}(1-\alpha)D_\alpha[\pi_0:\pi_*]$ in the third row, and confirm the identity in Eq. (86) in the fourth row.

As we also observe in Fig. 7, the soft value function or certainty equivalent $V_*(s) = \frac{1}{\beta}\Omega^{*(\alpha)}_{\pi_0,\beta}(Q)$ is not monotonic in $\alpha$ for this particular set of single-step rewards (the same $r(a,s)$ as in Fig. 3 or Fig. 7). Note the small scale of the $y$-axis in the top row of Fig. 5.

While it can be shown that $\psi_{\Delta r}(s;\beta)$ is convex as a function of $\beta$, we see that $\psi_{\Delta r}(s;\beta)$ is not necessarily convex in $\alpha$ and appears to be monotonically decreasing in $\alpha$. Finally, we find that the identity in Eq. (86)-(90) holds empirically, with only small numerical optimization issues.

**Value $V_*(s)$ as a Function of $\beta$, $\alpha$** In Fig. 7, we visualize the optimal value function $V_*(s) = \frac{1}{\beta}\Omega^*_{\pi_0,\beta}(Q)$, for KL or $\alpha$-divergence regularization and different choices of regularization strength $1/\beta$. The choice of divergence particularly affects the aggregated value at low regularization strength, although we do not observe a clear pattern with respect to $\alpha$.[5] In all cases, the value function ranges between $\max_a Q(a, s)$ for an unregularized deterministic policy as $\beta \to \infty$, and the expectation under the reference policy $\mathbb{E}_{\pi_0}[Q(a, s)]$ for strong regularization as $\beta \to 0$. We also discuss this property in Sec. 5.1.

## D  Robust Set of Perturbed Rewards

In this section, we characterize the robust set of perturbed rewards to which a given policy $\pi(a|s)$ or $\mu(a, s)$ is robust, which also provides performance guarantees as in Eq. (2) and also describes the set of strategies available to the adversary. For proving Prop. 1, we focus our discussion on policy regularization with KL or $\alpha$-divergence regularization and compare with state-occupancy regularization in App. D.2.

### D.1  Proof of Prop. 1: Robust Set of Perturbed Rewards for Policy Regularization

We begin by stating two lemmas, which we will use to characterize the robust set of perturbed rewards. All proofs are organized under paragraph headers below the statement of Prop. 1.

**Lemma 2.** *For the worst-case reward perturbation $\Delta r_\pi(a, s)$ associated with a given, normalized policy $\pi(a|s)$ and $\alpha$- or KL-divergence regularization, the conjugate function evaluates to zero,*

$$\frac{1}{\beta}\Omega^{*(\alpha)}_{\pi_0,\beta}(\Delta r_\pi) = 0\,. \tag{92}$$

**Lemma 3.** *The conjugate function $\frac{1}{\beta}\Omega^{*(\alpha)}_{\pi_0,\beta}(\Delta r)$ is increasing in $\Delta r$. In other words, if $\Delta\tilde{r}(a, s) \geq \Delta r(a, s)$ for all $(a, s) \in \mathcal{A} \times \mathcal{S}$, then $\frac{1}{\beta}\Omega^{*(\alpha)}_{\pi_0,\beta}(\Delta\tilde{r}) \geq \frac{1}{\beta}\Omega^{*(\alpha)}_{\pi_0,\beta}(\Delta r)$.*

**Proposition 1.** *Assume a normalized policy $\pi(a|s)$ for the agent is given, with $\sum_a \pi(a|s) = 1 \forall s \in \mathcal{S}$. Under $\alpha$-divergence policy regularization to a normalized reference $\pi_0(a|s)$, the optimization over $\Delta r(a, s)$ in Eq. (17) can be written in the following constrained form*

$$\min_{\Delta r \in \mathcal{R}^\Delta_\pi} \langle \mu, r - \Delta r \rangle \qquad where \quad \mathcal{R}^\Delta_\pi := \left\{ \Delta r \in \mathbb{R}^{\mathcal{A} \times \mathcal{S}} \,\middle|\, \Omega^{*(\alpha)}_{\pi_0,\beta}(\Delta r) \leq 0 \right\}, \tag{18}$$

*We refer to $\mathcal{R}^\Delta_\pi \subset \mathbb{R}^{\mathcal{A} \times \mathcal{S}}$ as the feasible set of reward perturbations available to the adversary. This translates to a robust set $\mathcal{R}_\pi$ of modified rewards $r'(a, s) = r(a, s) - \Delta r(a, s)$ for the given policy. These sets depend on the $\alpha$-divergence and regularization strength $\beta$ via the conjugate function.*

*For KL divergence regularization, the constraint is*

$$\sum_{a \in \mathcal{A}} \pi_0(a|s) \exp\left\{ \beta \cdot \Delta r(a, s) \right\} \leq 1\,. \tag{19}$$

*Proof.* Recall the adversarial optimization in Eq. (17) for a fixed $\mu(a, s) = \mu(s)\pi(a|s)$

$$\min_{\Delta r} \langle \mu, r - \Delta r \rangle + \frac{1}{\beta}\Omega^{*(\alpha)}_{\pi_0,\beta}(\Delta r), \tag{93}$$

which we would like to transform into a constrained optimization. From Lemma 2, we know that $\frac{1}{\beta}\Omega^{*(\alpha)}_{\pi_0,\beta}(\Delta r_\pi) = 0$ for the optimizing argument $\Delta r_\pi$ in Eq. (93), but it is not clear whether this should appear as an equality or inequality constraint. We now show that the constraint $\frac{1}{\beta}\Omega^{*(\alpha)}_{\pi_0,\beta}(\Delta r) \geq 0$ changes

---

[5]See Belousov & Peters (2019); Lee et al. (2018; 2019), or Appendix C.4 for additional discussion of the effect of $\alpha$-divergence regularization on learned policies.

the value of the objective, whereas the constraint $\frac{1}{\beta}\Omega^{*(\alpha)}_{\pi_0,\beta}(\Delta r) \leq 0$ does not change the value of the optimization.

$\geq$ **Inequality** First, consider the optimization $\min_{\Delta r}\langle\mu, r - \Delta r\rangle$ subject to $\frac{1}{\beta}\Omega^{*(\alpha)}_{\pi_0,\beta}(\Delta r) \geq 0$. From the optimizing argument $\Delta r_\pi$, consider an increase in the reward perturbations $\Delta\tilde{r}(a,s) \geq \Delta r_\pi(a,s) \; \forall(a,s)$ where $\exists(a,s)$ s.t. $\mu(a,s) > 0$ and $\Delta\tilde{r}(a,s) > \Delta r_\pi(a,s)$. By Lemma 3, we have $\frac{1}{\beta}\Omega^{*(\alpha)}_{\pi_0,\beta}(\Delta r) \geq \frac{1}{\beta}\Omega^{*(\alpha)}_{\pi_0,\beta}(\Delta r_\pi) = 0$. However, the objective now satisfies $\langle\mu, r - \Delta\tilde{r}\rangle < \langle\mu, r - \Delta r_\pi\rangle$ for fixed $\mu(a,s)$, which is a contradiction since $\Delta r_\pi$ provides a global minimum of the convex objective in Eq. (93).

$\leq$ **Inequality** We would like to show that this constraint does not introduce a different global minimum of Eq. (93). Assume there exists $\Delta\tilde{r}(a,s)$ with $\frac{1}{\beta}\Omega^{*(\alpha)}_{\pi_0,\beta}(\Delta\tilde{r}) < 0$ and $\langle\mu_\pi, r - \Delta\tilde{r}\rangle < \langle\mu_\pi, r - \Delta r_\pi\rangle$ for the occupancy measure $\mu_\pi$ associated with the given policy $\pi$. By convexity of $\frac{1}{\beta}\Omega^{*(\alpha)}_{\pi_0,\beta}(\Delta r)$, we know that a first-order Taylor approximation around $\Delta r_\pi$ everywhere underestimates the function, $\frac{1}{\beta}\Omega^{*(\alpha)}_{\pi_0,\beta}(\Delta\tilde{r}) \geq \frac{1}{\beta}\Omega^{*(\alpha)}_{\pi_0,\beta}(\Delta r_\pi) + \langle\Delta\tilde{r} - \Delta r_\pi, \nabla\frac{1}{\beta}\Omega^{*(\alpha)}_{\pi_0,\beta}(\Delta r_\pi)\rangle$. Noting that $\mu_\pi = \nabla\frac{1}{\beta}\Omega^{*(\alpha)}_{\pi_0,\beta}(\Delta r_\pi)$ by the conjugate optimality conditions (Eq. (5), App. A), we have $\frac{1}{\beta}\Omega^{*(\alpha)}_{\pi_0,\beta}(\Delta\tilde{r}) - \frac{1}{\beta}\Omega^{*(\alpha)}_{\pi_0,\beta}(\Delta r_\pi) \geq \langle\mu_\pi, \Delta\tilde{r}\rangle - \langle\mu_\pi, \Delta r_\pi\rangle$. This now introduces a contradiction, since we have assumed both that $\frac{1}{\beta}\Omega^{*(\alpha)}_{\pi_0,\beta}(\Delta\tilde{r}) - \frac{1}{\beta}\Omega^{*(\alpha)}_{\pi_0,\beta}(\Delta r_\pi) < 0$, and that $\Delta\tilde{r}(a,s)$ provides a global minimum, where $\langle\mu_\pi, r - \Delta\tilde{r}\rangle < \langle\mu_\pi, r - \Delta r_\pi\rangle$ implies $\langle\mu_\pi, \Delta\tilde{r}\rangle - \langle\mu_\pi, \Delta r_\pi\rangle > 0$. Thus, including the inequality constraint $\frac{1}{\beta}\Omega^{*(\alpha)}_{\pi_0,\beta}(\Delta\tilde{r}) \leq 0$ cannot introduce different minima.

This constraint is consistent with the constrained optimization and generalization guarantee in Eq. (1)-(2), where it is clear that increasing the modified reward away from the boundary of the robust set (i.e. decreasing $\Delta r(a,s)$ and $\frac{1}{\beta}\Omega^{*(\alpha)}_{\pi_0,\beta}(\Delta r)$) is feasible for the adversary and preserves our performance guarantee. See Eysenbach & Levine (2021) A2 and A6 for alternative reasoning. $\qquad\square$

**Proof of Lemma 2** For $\alpha$-divergence policy regularization and a given $\pi(a|s)$, we substitute the worst-case reward perturbations $\Delta r_\pi(a,s) = \frac{1}{\beta}\log_\alpha\frac{\pi(a|s)}{\pi_0(a|s)} + \psi_{\Delta r}(s;\beta)$ (Eq. (22) or Eq. (62)) in the conjugate function $\frac{1}{\beta}\Omega^{*(\alpha)}_{\pi_0,\beta}(\Delta r_\pi)$ (Eq. (56) or Table 2). Assuming $\sum_a \pi(a|s) = \sum_a \pi_0(a|s) = 1$, we have

$$\frac{1}{\beta}\Omega^{*(\alpha)}_{\pi_0,\beta}(\Delta r_\pi) = \frac{1}{\beta}\frac{1}{\alpha}\sum_a \pi_0(a|s)\exp_\alpha\left\{\beta\cdot\left(\Delta r_\pi(a,s) - \psi_{\Delta r}(s;\beta)\right)\right\}^\alpha - \frac{1}{\beta}\frac{1}{\alpha} + \psi_{\Delta r}(s;\beta)$$

$$= \frac{1}{\beta}\frac{1}{\alpha}\sum_a \pi_0(a|s)\left[1 + \beta(\alpha-1)\left(\frac{1}{\beta}\frac{1}{\alpha-1}\left(\frac{\pi(a|s)}{\pi_0(a|s)}^{\alpha-1} - 1\right) + \psi_{\Delta r}(s;\beta) - \psi_{\Delta r}(s;\beta)\right)\right]^{\frac{\alpha}{\alpha-1}}_+ - \frac{1}{\beta}\frac{1}{\alpha} + \psi_{\Delta r}(s;\beta)$$

$$= \frac{1}{\beta}\frac{1}{\alpha}\sum_a \pi_0(a|s)^{1-\alpha}\pi(a|s)^\alpha - \frac{1}{\beta}\frac{1}{\alpha} + \psi_{\Delta r}(s;\beta)$$

$$= 0.$$

In the last line, we recall that $\psi_{\Delta r}(s;\beta) = \frac{1}{\beta}\frac{1}{\alpha}\sum_a \pi_0(a|s) - \frac{1}{\beta}\frac{1}{\alpha}\sum_a \pi_0(a|s)^{1-\alpha}\pi(a|s)^\alpha$ from Eq. (23) or (61).

For KL regularization, we plug $\Delta r_\pi(a,s) = \frac{1}{\beta}\log\frac{\pi(a|s)}{\pi_0(a|s)}$ (Eq. (21),(50)) into the conjugate in Eq. (47) or Table 2,

$$\frac{1}{\beta}\Omega^*_{\pi_0,\beta}(\Delta r_\pi) = \frac{1}{\beta}\sum_a \pi_0(a|s)\exp\left\{\beta\,\Delta r_\pi(a,s)\right\} - \frac{1}{\beta} = \frac{1}{\beta}\sum_a \pi_0(a|s)\exp\left\{\beta\frac{1}{\beta}\log\frac{\pi(a|s)}{\pi_0(a|s)}\right\} - \frac{1}{\beta} = \frac{1}{\beta}\sum_a \pi(a|s) - \frac{1}{\beta} = 0.$$

**Proof of Lemma 3** See Husain et al. (2021) Lemma 3.

### D.2 Robust Set for $\alpha$-Divergence under $\mu(a,s)$ Regularization

For state-action occupancy regularization and KL divergence, Lemma 2 holds with $\frac{1}{\beta}\Omega^*_{\mu_0,\beta}(\Delta r_\mu) = 0$ for normalized $\mu(a,s)$ and $\Delta r_\mu(a,s) = \frac{1}{\beta}\log\frac{\mu(a,s)}{\mu_0(a,s)}$. However, the reasoning in App. D no longer holds for $\alpha$-divergence regularization to a reference $\mu_0(a,s)$. Substituting the worst-case reward perturbations (Eq. (24)

or (67)) into the conjugate function (Eq. (66) or Table 2)

$$
\begin{aligned}
\frac{1}{\beta}\Omega^{*(\alpha)}_{\mu_0,\beta}(\Delta r_\mu) &= \frac{1}{\beta}\frac{1}{\alpha}\sum_a \mu_0(a,s)\exp_\alpha\left\{\beta\cdot\Delta r_\mu(a,s)\right\}^\alpha - \frac{1}{\beta}\frac{1}{\alpha} \\
&= \frac{1}{\beta}\frac{1}{\alpha}\sum_a \mu_0(a,s)\Big[1+\beta(\alpha-1)\Big(\frac{1}{\beta}\frac{1}{\alpha-1}\big(\frac{\mu(a,s)}{\mu_0(a,s)}^{\alpha-1}-1\big)\Big)\Big]_+^{\frac{\alpha}{\alpha-1}} - \frac{1}{\beta}\frac{1}{\alpha} \\
&= \frac{1}{\beta}\frac{1}{\alpha}\sum_a \mu_0(a,s)^{1-\alpha}\mu(a,s)^\alpha - \frac{1}{\beta}\frac{1}{\alpha}
\end{aligned}
$$

(94)

whose value is not equal to 0 in general and instead is a function of the given $\mu(a,s)$. This may result in the original environmental reward not being part of the robust set, since substituting $\Delta r(a,s)=0$ into Eq. (94) results in $\frac{1}{\beta}\Omega^{*(\alpha)}_{\mu_0,\beta}(\Delta r)=0$.

### D.3 Plotting the $\alpha$-Divergence Feasible Set

To plot the boundary of the feasible set in the single step case, for the KL divergence regularization in two dimensions, we can simply solve for the $\Delta r(a_2,s)$ which satisfies the constraint $\sum_a \pi_0(a|s)\exp\{\beta\,\Delta r(a|s)\}=1$ for a given $\Delta r(a_1,s)$

$$
\Delta r(a_2,s) = \frac{1}{\beta}\log\frac{1}{\pi_0(a_2|s)}(1-\pi_0(a_1|s)\exp\left\{\beta\cdot\Delta r(a_1,s)\right\}).
$$

(95)

The interior of the feasible set contains $\Delta r(a_1,s)$ and $\Delta r(a_2,s)$ that are greater than or equal to these values.

However, we cannot analytically solve for the feasible set boundary for general $\alpha$-divergences, since the conjugate function $\frac{1}{\beta}\Omega^{*(\alpha)}_{\pi_0,\beta}(\Delta r)$ depends on the normalization constant of $\pi_{\Delta r}(a,s)$. Instead, we perform exhaustive search over a range of $\Delta r(a_1,s)$ and $\Delta r(a_2,s)$ values. For each pair of candidate reward perturbations, we use CVX-PY (Diamond & Boyd, 2016) to solve the conjugate optimization and evaluate $\frac{1}{\beta}\Omega^{*(\alpha)}_{\pi_0,\beta}(\Delta r)$. We terminate our exhaustive search and record the boundary of the feasible set when we find that $\frac{1}{\beta}\Omega^{*(\alpha)}_{\pi_0,\beta}(\Delta r)=0$ within appropriate precision.

## E Value Form Reward Perturbations

### E.1 Proof of Thm. 1 (Husain et al. (2021))

We rewrite the derivations of Husain et al. (2021) for our notation and setting, where $\Omega(\mu)$ represents a convex regularizer. Starting from the regularized objective in Eq. (12),

$$
\max_{\mu\in\mathcal{M}}\mathcal{RL}_{\Omega,\beta}(\mu) = \max_{\mu\in\mathcal{M}}\ \langle\mu,r\rangle - \frac{1}{\beta}\Omega(\mu),
$$

(96)

note that the objective is *concave*, as the sum of a linear term and the concave $-\Omega$. Since the conjugate is an involution for convex functions, we can rewrite $\mathcal{RL}_{\Omega,\beta}(r) = -(-\mathcal{RL}_{\Omega,\beta}(r)) = -((-\mathcal{RL}_{\Omega,\beta})^*)^*$, which

yields

$$
\begin{aligned}
\mathcal{RL}_{\Omega,\beta}(r) &= \sup_{\mu\in\mathcal{M}} -((-\mathcal{RL}_{\Omega,\beta})^*)^* \\
&\stackrel{(1)}{=} \sup_{\mu\in\mathcal{M}} -\left( \sup_{r'\in\mathbb{R}^{\mathcal{A}\times\mathcal{S}}} \langle\mu, r'\rangle - (-\mathcal{RL}_{\Omega,\beta})^*(r') \right) \\
&= \sup_{\mu\in\mathcal{M}} \inf_{r'\in\mathbb{R}^{\mathcal{A}\times\mathcal{S}}} \langle\mu, -r'\rangle + (-\mathcal{RL}_{\Omega,\beta})^*(r') \\
&\stackrel{(2)}{=} \sup_{\mu\in\mathcal{M}} \inf_{r'\in\mathbb{R}^{\mathcal{A}\times\mathcal{S}}} \langle\mu, r'\rangle + (-\mathcal{RL}_{\Omega,\beta})^*(-r') \\
&\stackrel{(3)}{=} \sup_{\mu\in\mathcal{M}} \inf_{r'\in\mathbb{R}^{\mathcal{A}\times\mathcal{S}}} \langle\mu, r'\rangle + \left( \sup_{\mu'} \langle\mu', -r'\rangle + \mathcal{RL}_{\Omega,\beta}(\mu') \right) \\
&\stackrel{(4)}{=} \sup_{\mu\in\mathcal{M}} \inf_{r'\in\mathbb{R}^{\mathcal{A}\times\mathcal{S}}} \langle\mu, r'\rangle + \left( \sup_{\mu'} \langle\mu', -r'\rangle + \langle\mu', r\rangle - \frac{1}{\beta}\Omega(\mu') \right) \\
&= \sup_{\mu\in\mathcal{M}} \inf_{r'\in\mathbb{R}^{\mathcal{A}\times\mathcal{S}}} \langle\mu, r'\rangle + \frac{1}{\beta}\left( \sup_{\mu'} \langle\mu', \beta\cdot(r-r')\rangle - \Omega(\mu') \right) \\
&\stackrel{(5)}{=} \sup_{\mu\in\mathcal{M}} \inf_{r'\in\mathbb{R}^{\mathcal{A}\times\mathcal{S}}} \langle\mu, r'\rangle + \frac{1}{\beta}\Omega^*(\beta\cdot(r-r')) \\
&\stackrel{(6)}{=} \inf_{r'\in\mathbb{R}^{\mathcal{A}\times\mathcal{S}}} \sup_{\mu\in\mathcal{M}} \langle\mu, r'\rangle + \frac{1}{\beta}\Omega^*(\beta\cdot(r-r'))
\end{aligned}
\tag{97}
$$

where (1) applies the definition of the conjugate of $(-\mathcal{RL}_{\Omega,\beta})^*$, (2) reparameterizes the optimization in terms of $r' \to -r'$, (3) is the conjugate for $(-\mathcal{RL}_{\Omega,\beta})$, and (4) uses the definition of the regularized RL objective for occupancy measure $\mu'(a,s)$ with the reward $r(a,s)$. Finally, (5) recognizes the inner maximization as the conjugate function for a modified reward and (6) swaps the order of inf and sup assuming the problem is feasible.

Note that Eq. (97) is a standard unregularized RL problem with modified reward $r'(a,s)$. As in Sec. 2, introducing Lagrange multipliers $V(s)$ to enforce the flow constraints and $\lambda(a,s)$ for the nonnegativity constraint,

$$
\mathcal{RL}_{\Omega,\beta}(r) = \inf_{r'} \inf_{V,\lambda} \sup_{\mu} \left\langle \mu, r' + \gamma\mathbb{E}_{a,s}^{s'}[V] - V + \lambda \right\rangle + \frac{1}{\beta}\Omega^*(\beta\cdot(r-r')) + (1-\gamma)\langle\nu_0, V\rangle
\tag{98}
$$

Now, eliminating $\mu(a,s)$ yields the condition

$$
r'(a,s) + \gamma\mathbb{E}_{a,s}^{s'}[V(s')] - V(s) + \lambda(a,s) = 0 \implies V(s) = r'(a,s) + \gamma\mathbb{E}_{a,s}^{s'}[V(s')] + \lambda(a,s).
\tag{99}
$$

Letting $\Delta r_V(a,s) = r(a,s) - r'(a,s)$, we can consider Eq. (99) as a constraint and rewrite Eq. (98) as

$$
\mathcal{RL}_{\Omega,\beta}(r) = \inf_{\Delta r_V} \inf_{V,\lambda} (1-\gamma)\langle\nu_0, V\rangle + \frac{1}{\beta}\Omega^*(\beta\cdot\Delta r_V)
\tag{100}
$$
$$
\text{subj. to } V(s) = r(a,s) + \gamma\mathbb{E}_{a,s}^{s'}[V(s')] - \Delta r_V(a,s) + \lambda(a,s)
$$

which matches Theorem 1. See Husain et al. (2021) for additional detail.

See App. A.3 for the proof of Prop. 3, which equates the form of $\Delta r_V(a,s)$ and $\Delta r_\pi(a,s)$ at optimality in the regularized MDP.

## E.2 Path Consistency (Comparison with Nachum et al. (2017); Chow et al. (2018))

We have seen in Sec. 3.3 and App. E.2 that the path consistency conditions arise from the KKT conditions. For KL divergence regularization, Nachum et al. (2017) observe the optimal policy $\pi_*(a|s)$ and value $V_*(s)$

satisfy

$$r(a, s) + \gamma \mathbb{E}_{a,s}^{s'}\big[V_*(s')\big] - \frac{1}{\beta} \log \frac{\pi_*(a|s)}{\pi_0(a|s)} = V_*(s) \,, \tag{101}$$

where the Lagrange multiplier $\lambda_*(a, s)$ is not necessary since the $\pi_*(a|s) > 0$ unless $\pi_0(a|s) = 0$ or the rewards or values are infinite. This matches our condition in Eq. (39), where we can also recognize $\Delta r_{\pi_*}(a, s) = \frac{1}{\beta} \log \frac{\pi_*(a|s)}{\pi_0(a|s)} = r(a, s) + \gamma \mathbb{E}_{a,s}^{s'}\big[V_*(s')\big] - V_*(s) - \lambda_*(a, s)$ as the identity from Prop. 3. Nachum et al. (2017) use Eq. (101) to derive a learning objective, with the squared error $\mathbb{E}_{a,s,s'}\big[\big(r(a, s) + \gamma \mathbb{E}_{a,s}^{s'}\big[V(s')\big] - \frac{1}{\beta} \log \frac{\pi(a|s)}{\pi_0(a|s)} - V(s)\big)^2\big]$ used as a loss for learning $\pi(a|s)$ and $V(s)$ (or simply $Q(a, s)$, Nachum et al. (2017) Sec. 5.1) using function approximation.

Similarly, Chow et al. (2018) consider a (scaled) Tsallis entropy regularizer, $\frac{1}{\beta}\Omega(\pi) = \frac{1}{\beta}\frac{1}{\alpha(\alpha-1)}(\sum_a \pi(a|s) - \sum_a \pi(a|s)^\alpha)$. For $\alpha = 2$, the optimal policy and value function satisfy

$$r(a, s) + \gamma \mathbb{E}_{a,s}^{s'}\big[V_*(s')\big] + \lambda(a, s) + \frac{1}{\beta}\frac{1}{\alpha(\alpha-1)} - \frac{1}{\beta}\frac{1}{\alpha-1}\pi(a|s)^{\alpha-1} = V_*(s) + \Lambda(s) \tag{102}$$

where $\Lambda(s)$ is a Lagrange multiplier whose value is learned in Chow et al. (2018). However, inspecting the proof of Theorem 3 in Chow et al. (2018), we see that this multiplier is obtained via the identity $\Lambda(s) := \psi_{Q_*}(s; \beta) - V_*(s) = \psi_{\Delta r_{\pi_*}}(s; \beta)$ (see Eq. (23), App. C.3). Our notation in Eq. (102) differs from Chow et al. (2018) in that we use $\frac{1}{\beta}$ as the regularization strength (compared with their $\alpha$). We have also written Eq. (102) to explicitly include the constant factors appearing in the $\alpha$-divergence.

In generalizing the path consistency equations, we will consider the $\alpha$-divergence, with $\Omega(\pi) = \frac{1}{\alpha(\alpha-1)}((1 - \alpha)\sum_a \pi_0(a|s) + \alpha \sum_a \pi(a|s) - \sum_a \pi_0(a|s)^{1-\alpha}\pi(a|s)^\alpha)$. Note that this includes an additional $\alpha$ factor which multiplies the $\sum_a \pi(a|s)$ term, compared to the Tsallis entropy considered in Chow et al. (2018). In particular, this scaling will change the $\frac{1}{\beta}\frac{1}{\alpha(\alpha-1)}$ additive constant term in Eq. (102), to a term of $\frac{1}{\beta}\frac{1}{\alpha-1}$.

Our expression for $\alpha$-divergence path consistency, derived using the identity $r(a, s) + \gamma \mathbb{E}_{a,s}^{s'}\big[V_*(s')\big] + \lambda_*(a, s) - \frac{1}{\beta} \log_\alpha \frac{\pi_*(a|s)}{\pi_0(a|s)} = V_*(s) + \psi_{\Delta r_{\pi_*}}(s; \beta)$ in Eq. (39), becomes

$$r(a, s) + \gamma \mathbb{E}_{a,s}^{s'}\big[V_*(s')\big] + \lambda(a, s) - \frac{1}{\beta} \log_\alpha \frac{\pi_*(a|s)}{\pi_0(a|s)} = V_*(s) + \psi_{\Delta r_{\pi_*}}(s; \beta) \tag{103}$$

where we have rearranged terms from Eq. (28) in the main text to compare with Eq. (102). Note that we can recognize $\Delta r_\pi(a, s) = \frac{1}{\beta} \log_\alpha \frac{\pi_*(a|s)}{\pi_0(a|s)} + \psi_{\Delta r_{\pi_*}}(s; \beta)$ and we substitute $\Lambda(s) := \psi_{Q_*}(s; \beta) - V_*(s) = \psi_{\Delta r_{\pi_*}}(s; \beta)$ compared to Eq. (102).

### E.3  Indifference Condition

In the single step setting with KL divergence regularization, Ortega & Lee (2014) argue that the perturbed reward for the optimal policy is constant with respect to actions

$$r(a) - \Delta r_{\pi_*}(a) = c \;\; \forall a \in \mathcal{A} \,, \tag{104}$$

when $\Delta r_{\pi_*}(a)$ are obtained using the optimal policy $\pi_*(a|s)$. Ortega & Lee (2014) highlight that this is a well-known property of Nash equilibria in game theory where, for the optimal policy and worst-case adversarial perturbations, the agent obtains equal perturbed reward for each action and thus is indifferent between them.

In the sequential setting, we can consider $Q(a, s) = r(a, s) + \gamma \mathbb{E}_{a,s}^{s'}\big[V(s')\big]$ as the analogue of the single-step reward or utility function. Using our advantage function interpretation for the optimal policy in Prop. 3, we directly obtain an indifference condition for the sequential setting

$$Q_*(a, s) - \Delta r_{\pi_*}(a, s) = V_*(s) \tag{105}$$

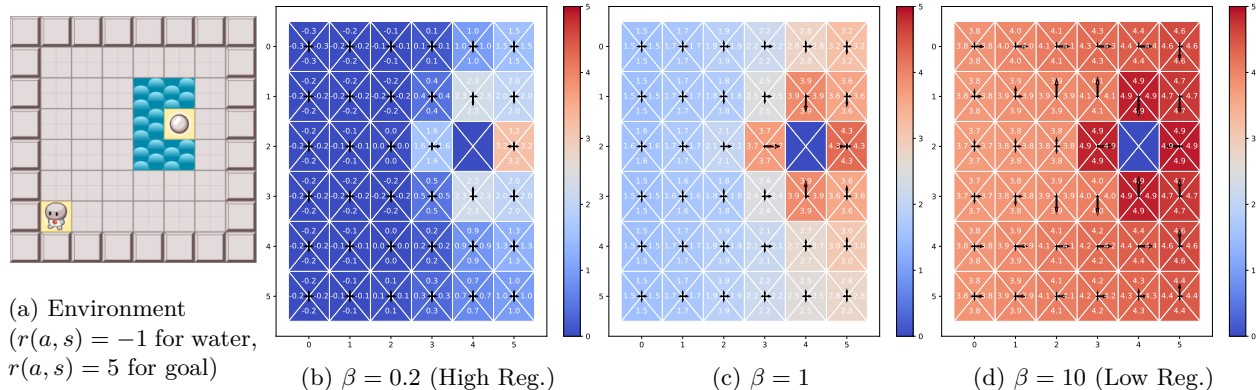

(a) Environment
$(r(a,s) = -1$ for water,
$r(a,s) = 5$ for goal)

(b) $\beta = 0.2$ (High Reg.)

(c) $\beta = 1$

(d) $\beta = 10$ (Low Reg.)

Figure 8: **Confirming Optimality of the Policy.** We show the perturbed rewards $r'(a,s) = Q(a,s) - \Delta r(a,s)$ for policies trained with KL divergence regularization. The indifference condition holds in all cases, with $r'(a,s) = c(s)$ for each state-action pair, with $c(s) = V(s)$ for KL regularization. This confirms that the policy is optimal (Ortega & Lee, 2014; Nachum et al., 2017).

for actions with $\lambda_*(a,s) = 0$ and nonzero probability under $\pi_*(a|s)$. Observe that $V_*(s)$ is a constant with respect to $a \in \mathcal{A}$ for a given state $s \in \mathcal{S}$. Recall that our notation for $V_*(s)$ omits its dependence on $\beta$ and $\alpha$. This indifference condition indeed holds for arbitrary regularization strengths $\beta$ and choices of $\alpha$-divergence, since our proof of the advantage function interpretation in App. A.3 is general. Finally, we emphasize that the indifference condition holds only for the optimal policy with a given reward $r(a,s)$ (see Fig. 3).

### E.4 Confirming Optimality using Path Consistency and Indifference

In Fig. 4, we plotted the regularized policies and worst-case reward perturbations for various regularziation strength $\beta$ and KL divergence regularization. In Fig. 8, we now seek to certify the optimality of each policy using the path consistency or indifference conditions. In particular, we confirm the following equality holds

$$r(a,s) + \gamma \mathbb{E}_{a,s}^{s'}\left[V(s')\right] - \frac{1}{\beta} \log \frac{\pi(a|s)}{\pi_0(a|s)} = V(s) \qquad \forall (a,s) \in \mathcal{A} \times \mathcal{S} \tag{106}$$

which aligns with the path consistency condition in (Nachum et al., 2017). Compared with Eq. (28) or Eq. (103), Eq. (106) uses the fact that $\lambda(a,s) = 0$ and $\psi_{\Delta r}(s;\beta) = 0$ in the case of KL divergence regularization. This equality also confirms the indifference condition since the right hand side $V_*(s)$ is a constant with respect to actions. Finally, we can recognize our advantage function interpretation $Q_*(a,s) - \Delta r_{\pi_*}(a,s) = V_*(s)$ in Eq. (106), by substituting $Q_*(a,s) = r(a,s) + \gamma \mathbb{E}_{a,s}^{s'}\left[V_*(s')\right]$ and $\Delta r_{\pi_*}(a,s) = \frac{1}{\beta} \log \frac{\pi_*(a|s)}{\pi_0(a|s)}$ for KL divergence regularization.

In Fig. 8, we plot $r(a,s) + \gamma \mathbb{E}_{a,s}^{s'}\left[V(s')\right] - \frac{1}{\beta} \log \frac{\pi(a|s)}{\pi_0(a|s)} = Q(a,s) - \Delta r_\pi(a,s)$ for each state-action pair to confirm that it yields a constant value and conclude that the policy and values are optimal. Note that this constant is different across states based on the soft value function $V_*(s)$, which also depends on the regularization strength.

## F Comparing Entropy and Divergence Regularization

In this section, we provide proofs and discussion to support our observations in Section 5.1 on the benefits of divergence regularization over entropy regularization.

### F.1 Tsallis Entropy and $\alpha$-Divergence

To show a relationship between the Tsallis entropy and the $\alpha$-divergence, we first recall the definition of the $q$-exponential function $\log_q$ (Tsallis, 2009). We also define $\log_\alpha(u)$, with $\alpha = 2 - q$, so that our use of $\log_\alpha(u)$

matches Lee et al. (2019) Eq. (5)

$$\log_q(u) = \frac{1}{1-q}\left(u^{1-q} - 1\right) \qquad \log_\alpha(u) := \log_{2-q}(u) = \frac{1}{\alpha-1}\left(u^{\alpha-1} - 1\right) \tag{107}$$

The Tsallis entropy of order $q$ (Tsallis, 2009; Naudts, 2011) can be expressed using either $\log_q$ or $\log_\alpha$

$$H_q^T[\pi(a)] = \frac{1}{q-1}\left(1 - \sum_{a\in\mathcal{A}}\pi(a)^q\right) = \sum_{a\in\mathcal{A}}\pi(a)\log_q\left(\frac{1}{\pi(a)}\right) \tag{108}$$

$$= -\sum_{a\in\mathcal{A}}\pi(a)\cdot\log_{2-q}\left(\pi(a)\right) \tag{109}$$

Eq. (108) and Eq. (109) mirror the two equivalent ways of writing the Shannon entropy for $q = 1$. In particular, we have $q = 2 - q$ and $H_1[\pi(a)] = \sum\pi(a)\log\frac{1}{\pi(a)} = -\sum\pi(a)\log\pi(a)$. See Naudts (2011) Ch. 7 for discussion of these two forms of the deformed logarithm.

To connect the Tsallis entropy and the $\alpha$-divergence in Eq. (7), we can consider a uniform reference measure $\pi_0(a) = 1\ \forall a$. For normalized $\sum_a\pi(a) = 1$,

$$D_\alpha[\pi_0(a):\pi(a)] = \frac{1}{\alpha(1-\alpha)}\left((1-\alpha)\sum_{a\in\mathcal{A}}\pi_0(a) + \alpha\sum_{a\in\mathcal{A}}\pi(a) - \sum_{a\in\mathcal{A}}\pi_0(a)^{1-\alpha}\pi(a|s)^\alpha\right) \tag{110}$$

$$= \frac{1}{\alpha(1-\alpha)}\left(\alpha + (1-\alpha) - \sum_{a\in\mathcal{A}}\pi(a|s)^\alpha\right) + \frac{1}{\alpha(1-\alpha)}\left((1-\alpha)\sum_{a\in\mathcal{A}}\pi_0(a) - (1-\alpha)\right) \tag{111}$$

$$= -\frac{1}{\alpha}H_\alpha^T[\pi(a)] + c \tag{112}$$

which recovers the negative Tsallis entropy of order $\alpha$, up to an multiplicative factor $\frac{1}{\alpha}$ and additive constant. Note that including this constant factor via $\alpha$-divergence regularization allows us to avoid an inconvenient $\frac{1}{\alpha}$ factor in optimal policy solutions (Eq. (27)) compared with Eq. 8 and 10 of Lee et al. (2019).

### F.2 Non-Positivity of $\Delta r_\pi(a, s)$ for Entropy Regularization

We first derive the worst-case reward perturbations for entropy regularization, before analyzing the sign of these reward perturbations for various values of $\alpha$ in Prop. 5. In particular, we have $\Delta r_\pi(a, s) \leq 0$ for entropy regularization with $0 < \alpha \leq 1$, which includes Shannon entropy regularization at $\alpha = 1$. This implies that the modified reward $r'_\pi(a, s) \geq r(a, s)$ for all $(a, s)$.

**Lemma 4.** *The worst-case reward perturbations for Tsallis entropy regularization correspond to*

$$\Delta r_\pi(a, s) = \frac{1}{\beta}\log_\alpha\pi(a|s) + \frac{1}{\beta}\frac{1}{\alpha}\left(1 - \sum_{a\in\mathcal{A}}\pi(a|s)^\alpha\right). \tag{113}$$

*with limiting behavior of $\Delta r_\pi(a, s) = \frac{1}{\beta}\log\pi(a|s)$ for Shannon entropy regularization as $\alpha \to 1$.*

*Proof.* We can write the Tsallis entropy using an additional constant $k$, with $k = \frac{1}{\alpha}$ mirroring the $\alpha$-divergence

$$H_\alpha^T[\pi] = \frac{k}{\alpha-1}\left(\sum_{a\in\mathcal{A}}\pi(a|s) - \sum_{a\in\mathcal{A}}\pi(a|s)^\alpha\right). \tag{114}$$

Note that we use *negative* Tsallis entropy for regularization since the entropy is concave. Thus, the worst case reward perturbations correspond to the condition $\Delta r_\pi(a, s) = \nabla\frac{1}{\beta}\Omega_{\pi_0}^{(H_\alpha)}(\mu) = -\nabla\mathbb{E}_{\mu(s)}\left[H_\alpha^T[\pi]\right]$. Differentiating with respect to $\mu(a, s)$ using similar steps as in App. B.3 Eq. (59)-(60), we obtain

$$\Delta r_\pi(a, s) = k\cdot\frac{1}{\beta}\alpha\,\log_\alpha\pi(a|s) + k\cdot\frac{1}{\beta}(\alpha-1)H_\alpha^T[\pi] \tag{115}$$

For $k = \frac{1}{\alpha}$ and $(\alpha-1)H_\alpha^T[\pi] = (\sum_a\pi(a|s) - \sum_a\pi(a|s)^\alpha)$, we obtain Eq. (113). $\qquad\square$

**Proposition 5.** *For $0 < \alpha \le 1$ and $\beta > 0$, the worst-case reward perturbations with Tsallis entropy regularization from [Lemma 4](#) are non-positive, with $\Delta r_\pi(a, s) \le 0$. This implies that the entropy-regularized policy is robust to only pointwise reward increases for these values of $\alpha$.*

*Proof.* We first show $\log_\alpha \pi(a|s) \le 0$ for $0 < \pi(a|s) \le 1$ and any $\alpha$. Note that we may write

$$\log_\alpha \pi(a|s) = \int_1^{\pi(a|s)} u^{\alpha-2} du = \frac{1}{\alpha-1} u^{\alpha-1} \Big|_1^{\pi(a|s)} = \frac{1}{\alpha-1} (\pi(a|s)^{\alpha-1} - 1). \quad (116)$$

Since $u^{\alpha-2}$ is a non-negative function for $0 \le \pi(a|s) \le 1$, then $\int_1^{\pi(a|s)} u^{\alpha-2} du \le 0$.

To analyze the second term, consider $0 < \alpha \le 1$. We know that $\pi(a|s)^\alpha \ge \pi(a|s)$ for $0 < \pi(a|s) \le 1$, so that $\sum_a \pi(a|s)^\alpha \ge \sum_a \pi(a|s) = 1$. Thus, we have $\sum_a \pi(a|s) - \sum_a \pi(a|s)^\alpha \le 0$ and $\alpha > 0$ implies $\frac{1}{\alpha}(\sum_a \pi(a|s) - \sum_a \pi(a|s)^\alpha) \le 0$. Since both terms are non-positive, we have $\Delta r_\pi(a, s) \le 0$ for $0 < \alpha \le 1$ as desired.

However, for $\alpha > 1$ or $\alpha < 0$, we cannot guarantee the reward perturbations are non-positive. Writing the second term in Eq. (113) as $\frac{\alpha-1}{\alpha} H_\alpha^T[\pi]$, we first observe that that $H_\alpha^T[\pi] \ge 0$. Now, $\alpha > 1$ or $\alpha < 0$ implies that $\frac{\alpha-1}{\alpha} > 0$, so that the second term is non-negative, compared to the first term, which is non-positive. $\quad \square$

### F.3 Bounding the Conjugate Function

**Conjugate for a Fixed $\alpha, \beta$:** We follow similar derivations as [Lee et al. (2019)](#) to bound the value function for general $\alpha$-divergence regularization instead of entropy regularization. We are interested in providing a bound for $\frac{1}{\beta} \Omega_{\pi_0,\beta}^{*(\alpha)}(Q)$ with fixed $\alpha, \beta$. To upper bound the conjugate, consider the optimum over each term separately

$$\frac{1}{\beta} \Omega_{\pi_0,\beta}^{*(\alpha)}(Q) = \max_{\pi \in \Delta^{|\mathcal{A}|}} \langle \pi, Q \rangle - \frac{1}{\beta} D_\alpha[\pi_0 : \pi]$$

$$\le \max_{\pi \in \Delta^{|\mathcal{A}|}} \langle \pi, Q \rangle - \min_\pi \frac{1}{\beta} D_\alpha[\pi_0 : \pi]$$

$$= \max_{\pi \in \Delta^{|\mathcal{A}|}} \langle \pi, Q \rangle - 0$$

$$= Q(a_{\max}, s).$$

where we let $a_{\max} := \arg\max_a Q(a, s)$.

We can also lower bound the conjugate function in terms of $\max_a Q(a, s)$. Noting that any policy $\pi(a|s)$ provides a lower bound on the value of the maximization objective, we consider $\pi_{\max}(a|s) = \delta(a = a_{\max})$. For evaluating $\pi_{\max}(a|s)^\alpha$, we assume $0^\alpha = 0$ for $\alpha > 0$ and undefined otherwise. Thus, we restrict our attention to $\alpha > 0$ to derive the lower bound

$$\frac{1}{\beta} \Omega_{\pi_0,\beta}^{*(\alpha)}(Q) = \max_{\pi \in \Delta^{|\mathcal{A}|}} \langle \pi, Q \rangle - \frac{1}{\beta} D_\alpha[\pi_0 : \pi]$$

$$\ge \langle \pi_{\max}, Q \rangle - \frac{1}{\beta} D_\alpha[\pi_0 : \pi_{\max}]$$

$$\overset{(1)}{=} Q(a_{\max}, s) - \frac{1}{\beta} \left( \frac{1}{\alpha(1-\alpha)} - \frac{1}{\alpha(1-\alpha)} \pi_0(a_{\max}|s)^{1-\alpha} 1^\alpha \right)$$

$$= Q(a_{\max}, s) + \frac{1}{\beta} \frac{1}{\alpha} \frac{1}{1-\alpha} \left( \pi_0(a_{\max}|s)^{1-\alpha} - 1 \right)$$

$$= Q(a_{\max}, s) + \frac{1}{\beta} \frac{1}{\alpha} \log_{2-\alpha} \pi_0(a_{\max}|s).$$

where (1) uses $\pi_{\max}(a|s) = \delta(a = a_{\max})$ and simplifies terms in the $\alpha$-divergence. One can confirm that $\frac{1}{\alpha} \log_{2-\alpha} \pi_0(a_{\max}|s) = \frac{1}{\alpha} \frac{1}{1-\alpha} (\pi_0(a_{\max}|s)^{1-\alpha} - 1) \le 0$ for $\alpha > 0$. Combining these bounds, we can write

$$\text{For } \alpha > 0, \qquad Q(a_{\max}, s) + \frac{1}{\beta} \frac{1}{\alpha} \log_{2-\alpha} \pi_0(a_{\max}|s) \le \frac{1}{\beta} \Omega_{\pi_0,\beta}^{*(\alpha)}(Q) \le Q(a_{\max}, s). \quad (117)$$

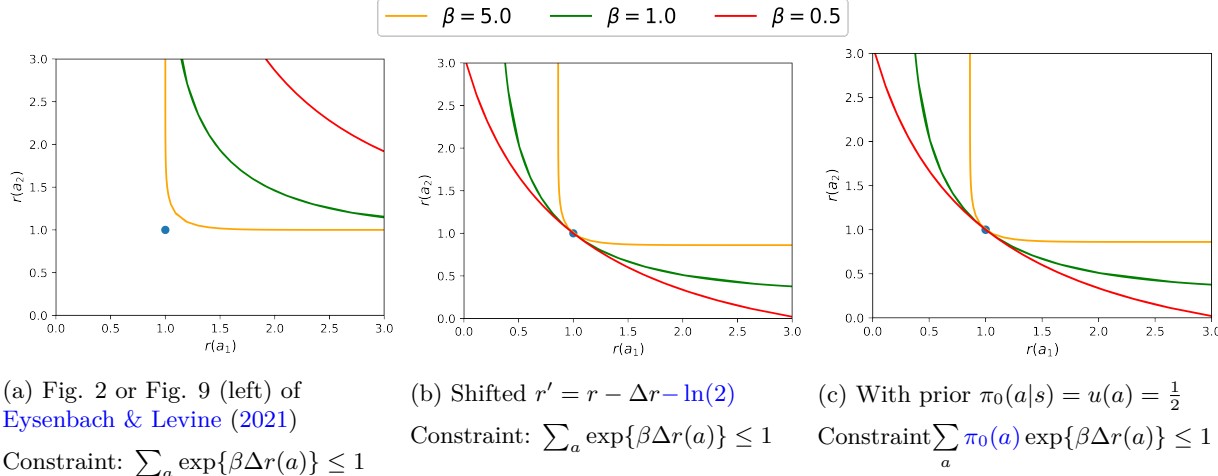

(a) Fig. 2 or Fig. 9 (left) of
Eysenbach & Levine (2021)

Constraint: $\sum_a \exp\{\beta \Delta r(a)\} \leq 1$

(b) Shifted $r' = r - \Delta r - \ln(2)$

Constraint: $\sum_a \exp\{\beta \Delta r(a)\} \leq 1$

(c) With prior $\pi_0(a|s) = u(a) = \frac{1}{2}$

Constraint $\sum_a \pi_0(a) \exp\{\beta \Delta r(a)\} \leq 1$

Figure 9: Analyzing the feasible set for entropy regularization (a) versus divergence regularization (b,c).

**Conjugate as a Function of $\beta$:** Finally, we can analyze the conjugate as a function of $\beta$. As $\beta \to 0$ and $1/\beta \to \infty$, the divergence regularization forces $\pi(a|s) = \pi_0(a|s)$ for any $\alpha$ and the conjugate $\frac{1}{\beta}\Omega_{\pi_0,\beta}^{*(\alpha)}(Q) = \langle \pi(a|s), Q(a,s) \rangle - \frac{1}{\beta}\Omega(\pi) \to \langle \pi(a|s), Q(a,s) \rangle$. As $\beta \to \infty$ and $1/\beta \to 0$, the unregularized objective yields a deterministic optimal policy with $\pi(a|s) = \max_a Q(a,s)$. In this case, the conjugate $\frac{1}{\beta}\Omega_{\pi_0,\beta}^{*(\alpha)}(Q) \to \max_a Q(a,s)$. Thus, treating the conjugate as a function of $\beta$, we obtain

$$\mathbb{E}_{\pi_0(a|s)}[Q(a,s)] \leq \frac{1}{\beta}\Omega_{\pi_0,\beta}^{*(\alpha)}(Q) \leq \max_a Q(a,s) \tag{118}$$

For negative values of $\beta$, the optimization becomes a minimization, with the optimal solution approaching a deterministic policy with $\pi(a|s) = \min_a Q(a,s)$ as $\beta \to -\infty$, $1/\beta \to 0$.

### F.4 Feasible Set for Entropy Regularization

In this section, we compare feasible sets derived from entropy regularization (Fig. 9a) with those derived from the KL divergence (Fig. 2, Fig. 9b, Fig. 9c), and argue that entropy regularization should be analyzed as a special case of the KL divergence to avoid misleading conclusions.

In their App. A8, Eysenbach & Levine (2021) make the surprising conclusion that policies with *higher* entropy regularization are *less* robust, as they lead to smaller feasible or robust sets than for lower regularization. This can be seen in the robust set plot for entropy regularization in Fig. 9a, where higher $\beta$ indicates lower regularization strength $(1/\beta)$. The left panels in Eysenbach & Levine (2021) Fig. 2 or Fig. 9 match our Fig. 9a. See their App. A8 for additional discussion.

To translate from Shannon entropy to KL divergence regularization, we include an additional additive constant of $\frac{1}{\beta}\log|\mathcal{A}|$ corresponding to the (scaled) entropy of the uniform distribution, with $-\frac{1}{\beta}D_{KL}[\pi : \pi_0] = \frac{1}{\beta}H(\pi) - \frac{1}{\beta}\log|\mathcal{A}|$. We obtain Fig. 9b by shifting each curve in Fig. 9a by this scaled constant . This highlights the delicate dependence on the feasible set on the exact form of the objective function, as the constant shifts the robust reward set by $(r'(a_1), r'(a_2)) \leftarrow (r'(a_1) - \frac{1}{\beta}\log 2, r'(a_2) - \frac{1}{\beta}\log 2)$. For KL divergence regularization, the feasible set now includes the original reward function. As expected, we can see that policies with higher regularization strength are *more* robust with *larger* feasible sets.

An alternative approach to correct the constraint set is to include the uniform reference distribution as in Prop. 1 and Eq. (19), so that we calculate $\sum_a \pi_0(a|s) \exp\{\beta \cdot \Delta r\} \leq 1$. Similarly, the constraint in Eysenbach & Levine (2021) can be modified to have $\sum_a \exp\{\beta \cdot \Delta r\} \leq |\mathcal{A}|$ in the case of entropy regularization.

Our modifications to the feasible set constraints clarify interpretations of how changing regularization strength affects robustness. We do not give detailed consideration to the other question briefly posed in Eysenbach & Levine (2021) App. A8, of 'if a reward function $r'$ is included in a robust set, what other reward

functions are included in that robust set?', whose solution is visualized in Eysenbach & Levine (2021) Fig. 9 (right panel) without detailed derivations. However, this plot matches our Fig. 9b and 9c. This suggests that the constraint arising from KL divergence regularization with strength $\beta$, $\sum \pi_0(a|s) \exp\{\beta \cdot \Delta r(a,s)\} = 1$, is sufficient to define both the robust set for a given reward function and to characterize the other reward functions to which an optimal policy is robust. The key observation is that the original reward function is included in the robust set when explicitly including the reference distribution $\pi_0(a|s)$ as in divergence regularization.

## G Worked Example for Deterministic Regularized Policy ($\alpha = 2$, $\beta = 10$)

We consider the single-step example in Sec. 4.1 Fig. 2 or App. H Fig. 10-11, with a two-dimensional action space, optimal state-action value estimates, $Q_*(a,s) = r(a,s) = \{1.1, 0.8\}$, and uniform prior $\pi_0(a|s)$.

The case of policy regularization with $\alpha = 2$ and $\beta = 10$ is particularly interesting, since the optimal policy is deterministic with $\pi_*(a_1|s) = 1$. [6] First, we solve for the optimal policy for $Q_*(a,s) = r(a,s)$ as in App. C.2,

$$\frac{1}{\beta}\Omega^*_{\pi_0,\beta}(Q_*) = \max_{\pi \in \Delta^{|\mathcal{A}|}} \left\langle \pi, Q_* \right\rangle - \frac{1}{\beta}\Omega^{(\alpha)}_{\pi_0}(\pi) - \psi_Q(s;\beta)\left(\sum_a \pi(a|s) - 1\right) + \sum_a \lambda(a,s)$$

$$\implies \pi_*(a|s) = \pi_0(a|s)\left[1 + \beta(\alpha-1)\left(Q_*(a,s) + \lambda_*(a,s)\underbrace{-V_*(s) - \psi_{\Delta r_{\pi_*}}(s;\beta)}_{=\psi_{Q_*}(s;\beta)\ \text{(see App. C.3 )}}\right)\right]^{\frac{1}{\alpha-1}}.$$

where for $\alpha = 2$, we obtain $\pi_*(a|s) = \pi_0(a|s)\left[1 + \beta\left(Q_*(a,s) + \lambda_*(a,s) - V_*(s) - \psi_{\Delta r_{\pi_*}}(s;\beta)\right)\right]$.

Using CVX-PY (Diamond & Boyd, 2016) to solve this optimization for $\alpha = 2$, $\beta = 10$, $\pi_0(a|s) = \frac{1}{2}\,\forall a$, and the given $Q_*(a,s)$, we obtain

$$\begin{array}{lll}
Q_*(a_1,s) = 1.1 & Q_*(a_2,s) = 0.8 & \lambda_*(a_1,s) = 0 \\
\pi_*(a_1|s) = 1 & \pi_*(a_2|s) = 0 & \lambda_*(a_2,s) = 0.1 \\
V_*(s) = 1.05 & \psi_{\Delta r_{\pi_*}}(s;\beta) = -0.05 & \psi_{Q_*}(s;\beta) = 1.0\,.
\end{array} \tag{119}$$

Our first observation is that, although the policy is deterministic with $\pi_*(a_1|s) = 1$, the value function $V_*(s) = 1.05$ is not equal to $\max_a Q_*(a,s) = 1.1$ as it would be in the case of an unregularized policy. Instead, we still need to subtract the $\alpha$-divergence regularization term, which is nonzero. With $\alpha = 2$ and $1 - \alpha = -1$, we have

$$V_*(s) = \langle \pi_*(a|s), Q_*(a,s) \rangle - \underbrace{\frac{1}{\beta}\frac{1}{\alpha}\frac{1}{1-\alpha}\left(1 - \sum_a \pi_0(a|s)^{1-\alpha}\pi_*(a|s)^\alpha\right)}_{\frac{1}{\beta}D_\alpha[\pi_0:\pi_*]} = 1.1 - \frac{1}{10}\frac{1}{2}\frac{1}{-1}\left(1 - .5^{-1}\cdot 1^2 - .5^{-1}\cdot 0^2\right)$$

$$= 1.1 + .05 \cdot (1 - 2) = 1.05$$

Recall that for normalized $\pi_0$, $\pi$, we have $\psi_{\Delta r_{\pi_*}}(s;\beta) = -\frac{1}{\beta}(1-\alpha)D_\alpha[\pi_0(a|s) : \pi_*(a|s)] = -0.05$, so that we can confirm Eq. (119) for $\alpha = 2$.

Finally, we confirm the result of Prop. 3 by calculating the reward perturbations in two different ways. For $a_1$, we have

$$\Delta r_{\pi_*}(a_1,s) = \frac{1}{\beta}\frac{1}{\alpha-1}\left(\frac{\pi_*(a_1|s)}{\pi_0(a_1|s)}^{\alpha-1} - 1\right) + \psi_{\Delta r_{\pi_*}}(s;\beta) = \frac{1}{10}\frac{1}{1}\left(\frac{1}{.5}^1 - 1\right) - .05 = .05$$

$$= Q_*(a_1,s) - V_*(s) + \lambda_*(a_1,s) = 1.1 - 1.05 + 0 = .05,$$

and for $a_2$,

$$\Delta r_{\pi_*}(a_2,s) = \frac{1}{\beta}\frac{1}{\alpha-1}\left(\frac{\pi_*(a_2|s)}{\pi_0(a_2|s)}^{\alpha-1} - 1\right) + \psi_{\Delta r_{\pi_*}}(s;\beta) = \frac{1}{10}\frac{1}{1}\left(\frac{0}{.5}^1 - 1\right) - .05 = -.15$$

$$= Q_*(a_2,s) - V_*(s) + \lambda_*(a_2,s) = 0.8 - 1.05 + 0.1 = -.15$$

---

[6]We use $\alpha = 2$ instead of $\alpha = 3$ in Fig. 2 for simplicity of calculations. See App. H Fig. 10 for $\alpha = 2$ robust set plots.

so that we have $\Delta r_{\pi_*}(a_1, s) = 0.05$ and $\Delta r_{\pi_*}(a_2, s) = -0.15$.

We can observe that the indifference condition *does not* hold, since $Q_*(a_1, s) - \Delta r_{\pi_*}(a_1, s) = 1.1 - 0.05 = 1.05$ does not match $Q_*(a_2, s) - \Delta r_{\pi_*}(a_2, s) = 0.8 - (-0.15) = 0.95$.

However, adding the Lagrange multiplier $\lambda_*(a_2, s) = 0.1$ accounts for the difference in these values. This allows us to confirm the path consistency condition (Eq. (28)),

$$\underbrace{r(a, s) + \gamma \mathbb{E}_{a,s}^{s'}\big[V_*(s')\big]}_{Q_*(a,s)} - \underbrace{\frac{1}{\beta} \log_\alpha \frac{\pi_*(a|s)}{\pi_0(a|s)} - \psi_{\Delta r_{\pi_*}}(s; \beta)}_{\Delta r_{\pi_*}(a,s)} = V_*(s) - \lambda_*(a, s) \qquad \forall(a, s) \in \mathcal{A} \times \mathcal{S} \qquad (120)$$

with $Q_*(a_1, s) - \Delta r_{\pi_*}(a_1, s) - V_*(s) + \lambda_*(a_1, s) = 1.1 - 0.05 - 1.05 + 0 = 0$ and $Q_*(a_2, s) - \Delta r_{\pi_*}(a_2, s) - V_*(s) + \lambda_*(a_2, s) = 0.8 - (-0.15) - 1.05 + 0.1 = 0$.

## H   Additional Feasible Set Plots

In Fig. 10 and 11, we provide additional feasible set plots for the $\alpha$-divergence with $\alpha \in \{-1, 1, 2, 3\}$ and $\beta \in \{0.1, 1.0, 5, 10\}$ with $r(a, s) = [1.1, 0.8]$. As in Fig. 2, we show the feasible set corresponding to the single-step optimal policy for $Q_*(a, s) = r(a, s)$ for various regularization schemes. Since each policy is optimal, we can confirm the indifference condition for the KL divergence, with $Q_*(a, s) - \Delta r_{\pi_*}(a, s) = V_*(s)$ constant across actions and equal to the soft value function, certainty equivalent, or conjugate function $V_*(s) = \frac{1}{\beta} \Omega_{\pi_0,\beta}^{*(\alpha)}(Q)$. When the indifference condition holds, we can obtain the ratio of action probabilities in the regularized policy by taking the slope of the tangent line to the feasible set boundary at $r'_{\pi_*}(a, s)$, as in Sec. 4.1.

However, indifference does not hold in cases where the optimal policy sets $\pi_*(a_2|s) = 0$, which occurs for $(\alpha = 2, \beta = 10)$, $(\alpha = 3, \beta \in \{5, 10\})$ for a uniform reference policy and additionally for $(\alpha = 2, \beta = 5)$ with the nonuniform reference in Fig. 11. In these cases, we cannot ignore the Lagrange multiplier in Eq. (28), $r'_{\pi_*}(a, s) = Q_*(a, s) - \Delta r_{\pi_*}(a, s) = V_*(s) - \lambda_*(a, s)$, and $\lambda_*(a_2, s) > 0$ results in a different perturbed reward $r'_{\pi_*}(a_1, s) \neq r'_{\pi_*}(a_2, s)$.

For $\alpha = -1$ and low regularization strength ($\beta = 10$), we observe a wider feasible set boundary than for KL divergence regularization. For $\alpha = 2$ and $\alpha = 3$, the boundary is more restricted and the worst-case reward perturbations become notably smaller when the policy is deterministic. For example, we can compare $\beta = 5$ versus $\beta = 10$ for $\alpha = 2$. However, as in Fig. 7, we do not observe notable differences in the robust sets at lower regularization strengths based on the choice of $\alpha$-divergence.

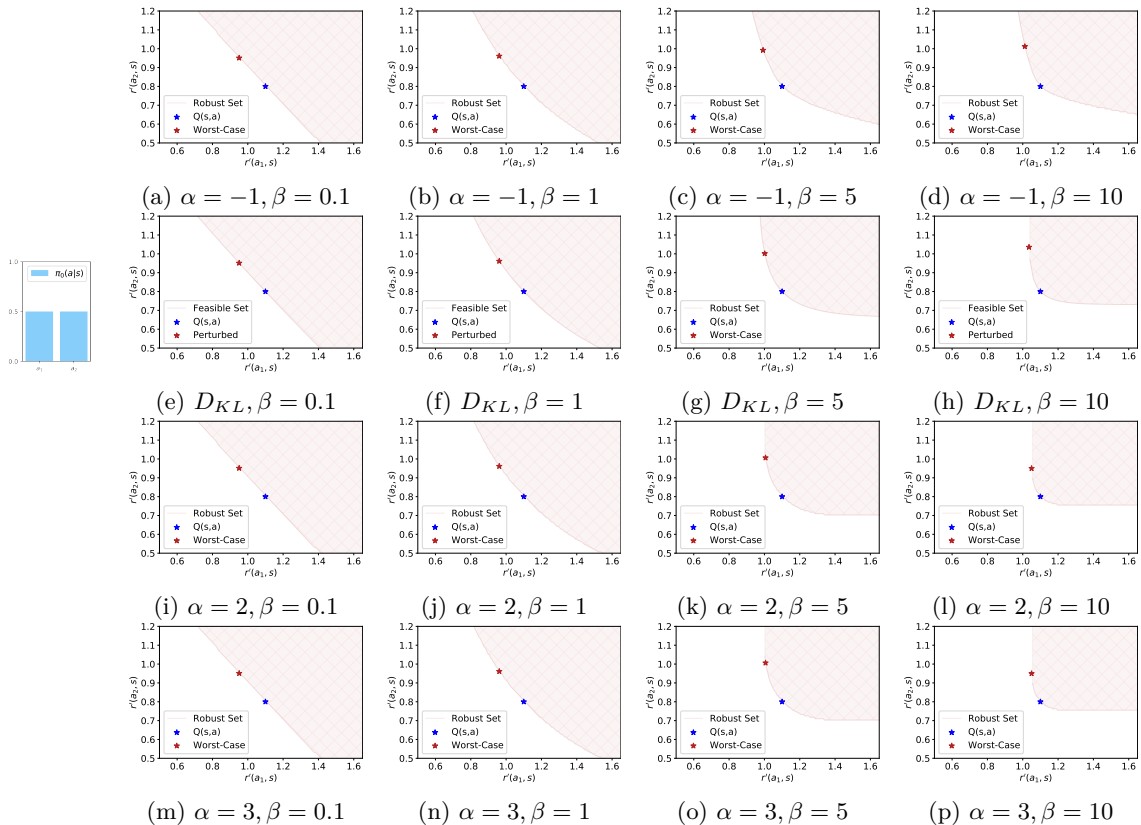

Figure 10: Reference distribution $\pi_0 = (\frac{1}{2}, \frac{1}{2})$. See caption of Fig. 11.

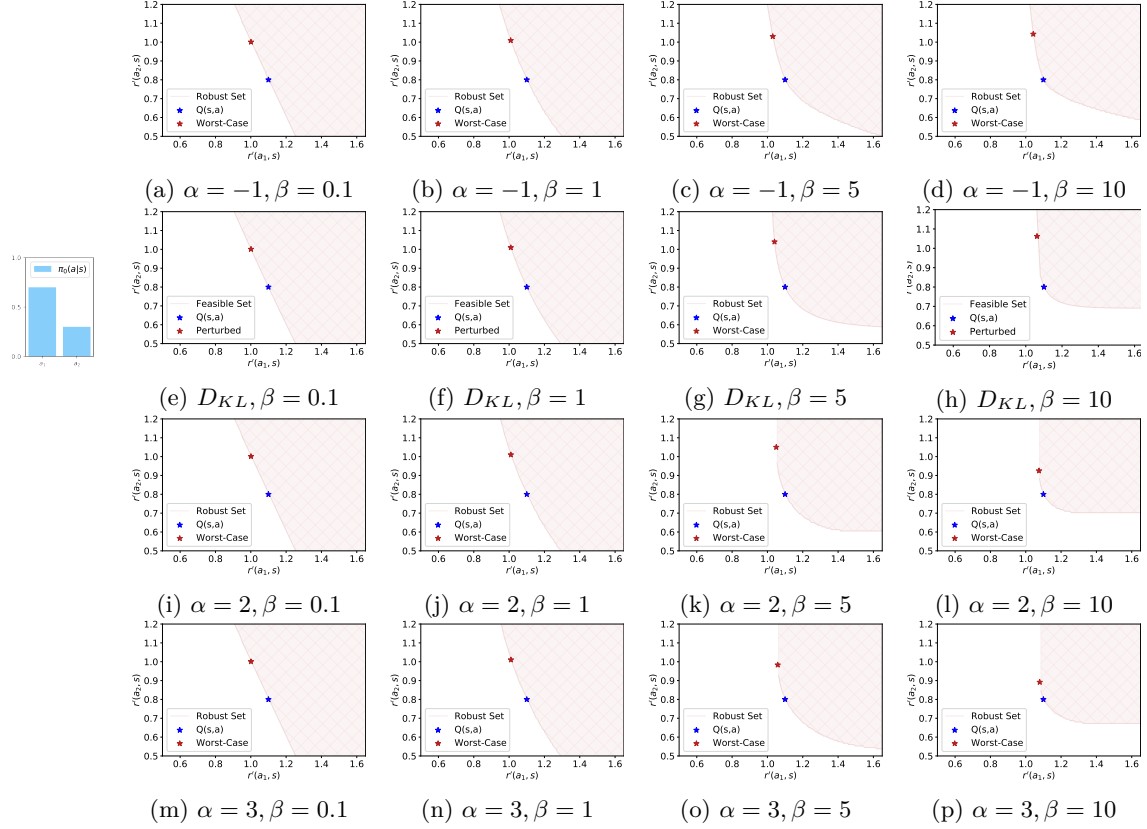

Figure 11: Reference distribution $\pi_0 = (\frac{2}{3}, \frac{1}{3})$. Feasible Set (red region) of perturbed rewards available to the adversary, for KL ($\alpha = 1$) and $\alpha$-divergence ($\alpha = \{-1, 2, 3\}$) regularization, various $\beta$, and fixed $Q_*(a, s) = r(a, s)$ values (blue star). We consider the optimal $\pi_*(a|s)$ with regularization parameters $\alpha, \beta, \pi_0$ and the given $Q$-values. Red star indicates worst-case perturbed reward $r'_{\pi_*} = r - \Delta r_{\pi_*}$ for optimal policy.

