# OpenReview forum: "Your Policy Regularizer is Secretly an Adversary"
_TMLR — Accepted by TMLR_

### Review · Reviewer_1MHa · 2022-04-20

**Summary Of Contributions:**

This paper studies policy regularization algorithms (such as entropy regularization) that are widely used in RL practice. The main focus is an equivalence result showing that the regularized RL problem is equivalent to a robust reward maximization problem with the reward perturbation lying in a certain set depending on the regularizer, which are first established in Eysenbach and Levine (2021) and Husain et al. (2021).

The main contribution of this paper is a generalized and more unified version of this result:

* spells out clearly the underlying reason (convex conjugacy) that enables this result
* covers new regularizers ($\alpha$-divergence)
* a visualization of the worst-case reward perturbation corresponding to the optimal policy with various regularizers, in toy and grid-world environments


**Broader Impact Concerns:**

I don’t see any broader impact or ethical concerns regarding this paper.

**Requested Changes:**

* Perhaps the authors could revise the mathematical statements to make them more self-contained. Adding references to equation numbers (and adding new equations when necessary) may help.

* I also suggest the authors carefully revise the claims in the abstract and introduction to better acknowledge existing work. Maybe consider moving Table 2 to the beginning (e.g. Page 2 / 3) to the paper.

* I find the barplot visualizations in Figure 3 not that easy to digest (also, perhaps not much more informative over Figure 1 / 2). Maybe consider moving it to the Appendix or improving its presentation.


**Strengths And Weaknesses:**

Strengths:
* The topic of the paper, regularized RL, should be of great interest to the RL / ML community in general. It is also widely used in modern RL practice and so practitioners may benefit from the understandings provided in this paper.

* Overall, the paper is well-executed, and may provide a better entry point for this topic compared with the aforementioned two papers:
  - Eysenbach and Levine (2021), Theorem 4.1 first proves this result for the entropy regularizer. However, their paper did not spell out convex conjugacy as the main underlying reason for this result.
  - Husain et al (2021) is more similar to the present paper in that they also use convex conjugacy to obtain this result, and they present a generic result that holds for general convex regularizers. The improvement of the present paper over Husain et al. (2021) is perhaps a bit more marginal. However, given this generic result, the present paper contains more concrete specializations (e.g. $\alpha$-divergence) and discussions, which could make it more digestible.

* I also like the visualization experiments, in particular the grid-world experiment in Figure 4. It is interesting to see how various regularization strengths correspond to qualitatively different worst-case reward perturbations.

* The main claims within this paper are justified in my opinion. I did not go through all the Appendix, but I checked a few main proofs and they appear to be correct.

Weaknesses:
* The paper seems like an “understanding” paper in my opinion, as opposed to either an RL theory paper or an algorithms / empirical RL paper. While that itself may not be a concern, one may argue that the understanding presented in this paper is not new, as the essence is already in the aforementioned two papers. Therefore, even though I like the execution of this paper, I am not sure how to judge the worthiness of such a contribution.

* The mathematical statements in some results / discussions are often not self-contained and could be confusing. A few examples are,
  - The statement of Proposition 1, “the feasible set of adversarial reward perturbations is defined by the constraint xxx”, does not seem to have a subject. I wonder whether the authors mean “The regularized problem (12) is equivalent to the constrained problem (1) with feasible set xxx”.
  - Section 5.1, difference between Shannon entropy (used in Eysenbach & Levine 2021) vs. KL. Why is there a difference? I am confused here because on Page 3 the authors consider the KL divergence to be wrt unnormalized measures, so I think they are equal up to an additive constant. Perhaps the authors could have the formula for Shannon entropy vs. the particular KL they are referring to side-by-side to highlight the difference.

* Related to the first point, I feel like the overall claim of this paper (in particular the title/abstract) may be slightly overclaiming and needs to be tuned. In particular, the abstract sounds like the equivalence result itself is new, which does not properly reflect existing work. The main text suffers less from this issue, e.g. Table 2 seems like a rather fair comparison with prior work.

Given the above, my overall feeling about this paper is weakly positive, with the main advantage being the practical importance of the topic and the good execution, and main concerns being on the worthiness of the contribution.

---

### Review · Reviewer_sL7p · 2022-04-26

**Summary Of Contributions:**

This paper studies the use of regularizers in reinforcement learning. The main objective of the submission is to give a unified explanation that divergence regularization in reinforcement learning / both in the policy and the state visitation settings can be interpreted as finding a solution that is robust to reward perturbations. These results had been alluded to in the existing literature, but there had not been a thorough study and explanation of these phenomena before. The authors provide mathematical derivations of these claims and some experimental results to provide visual explanation of their findings. The main ‘insight’ is that by writing regularized RL as a saddle point problem, one can interpret the solution as a ‘game’ between a learner and an adversary. The adversary’s objective can be written as producing a perturbation of the reward within a feasibility set determined by the nature divergence regularizer.

**Broader Impact Concerns:**

This work does not pose any ethical concerns.

**Requested Changes:**

I believe the work has enough merits to deserve publication.  That being said, I would really like to see addressed the concerns expressed above around the nature of the adversary and how that links with the claims made by the paper.

**Strengths And Weaknesses:**

The paper is very well written, and the text is very easy to follow.  I have some concerns about the relevance of the results presented in this work. Most of the derivations are standard consequences of convex analysis and are to my understanding present in other works in the literature. Regularization in MDPs and the properties of their dual, and primal problems have a long and rich literature. Nonetheless, there is some value in presenting these simple derivations in a single place and with a unified story. As for the robustness explanation advanced by the authors, I am not extremely convinced this angle is more than by noting the Lagrangian can (obviously) be written as a min -max objective. My main concern here is how the constraints on the adversary’s perturbations are a property of the regularizer. This means the adversary here is more of a semantic devise than a true model for potential adversarial behavior.

---

### Review · Reviewer_drrf · 2022-05-02

**Summary Of Contributions:**

The first main contribution of this work is to show robustness of regularization policy optimization, i.e., there exists a robust set of reward perturbation, such that for any perturbed reward in this set, solving the regularized value maximization problem does not harm the regularized optimality (Proposition 1). The authors also did provide results for KL divergence, alpha-divergence, Shannon entropy and Tsallis entropy regularizers.

The second main contribution is the calculation of the worst-case reward perturbations. The results match existing work of path consistency conditions and an indifference condition in game theory literature.

The authors then visualized the robust reward sets under simple cases (Figures 1 and 2), and conducted simulations to verify the theory.





**Broader Impact Concerns:**

This work contains mostly theoretical results and simulations. As far as I can see, I did not see any ethical implications.

**Requested Changes:**

Please clarify the following questions.

(1) In Figure 1, it seems that both $r(s, a)$ and the red star (Prop. 2) are on a frontier (it looks to me all the points on the same frontier are kind of the same in terms of they are on the boundary of being robust). Why is the red star called the worst-case?

(2) The main results give me the following intuition: if we do not want to hurt the performance (which is the maximum regularized value), then only adding reward could achieve that (make a larger $r^\prime$), e.g. as shown in Eq. (19)

However, if we define $r^\prime = r - 100$, then Eq. (19) would not hold, but arguably this reward shift should not change the performance/robustness of the the algorithm (since it does not change the problem's nature). Does the theory support this? Or if this does hurt robustness, how come this would happen?

(3) Following the above (if the case is sometimes even if conditions like Eq. (19) is not satisfied, robustness can still be preserved), I think a more interesting question would be to investigate the robustness for perturbations beyond the robust sets (or show the robust sets in this work is practical enough).

Since a larger $r^\prime$ would always intuitively make the value large, and in practice an environment noise probably would not satisfy this. It would be ideal if the authors could discuss whether the calculated robust perturbation sets in this work could cover e.g. a Gaussian noise, or what would be the robustness result beyond the robust sets.



**Strengths And Weaknesses:**

Strengths:

(1) The understanding of regularizers are ways to create robust reward perturbation is interesting. The main results (Propositions 1 and 2) tell a novel and reasonable understanding of the effect of regularizers in reinforcement learning (RL).

(2) The techniques are not highly novel but are cleverly used.

This work and its techniques are mainly based on regularized policy optimization, which is a relatively well-studied line of work in RL. The authors used existing techniques in a meaningful way. Relations with existing results of path consistency and indifference conditions are well discussed, which I appreciate.

(3) Simulations are conducted to support the results.

Weakness:

(1) The theory seems a bit weak. It gives me a sense that "to not hurt the performance we would just make reward value larger", which is reasonable but not quite surprising. I added some comments in the below question part.

(2) The experiments seem not enough to show that the robust sets in the theory is a very useful result. If there could be some theoretical/empirical results showing that the robust perturbation sets cover a number of practical perturbations/noises, then that would be great.

---

### Decision · Action_Editors · 2022-07-01

**Recommendation:** Accept as is

**Comment:**

The reviewers were all in favor of accepting the paper. During the reviewer discussion (whose contents were summarized to the authors in a direct discussion), there were some concerns raised regarding the validity of the robustness notions considered in the paper, but eventually it was deemed that a detailed study of these issues probably goes way beyond the scope of this work. Thus, given the overall positive evaluation of the work by the reviewers, I am eventually happy to recommend this paper for acceptance at TMLR.